# Channel Simulation and Distributed Compression with Ensemble Rejection Sampling

**Buu Phan**[1]    **Ashish Khisti** [1]

Department of Electrical and Computer Engineering, University of Toronto
`truong.phan@mail.utoronto.ca, akhisti@ece.utoronto.ca`

## Abstract

We study *channel simulation* and *distributed matching*, two fundamental problems with several applications to machine learning, using a recently introduced generalization of the standard rejection sampling (RS) algorithm known as Ensemble Rejection Sampling (ERS). For channel simulation, we propose a new coding scheme based on ERS that achieves a near-optimal coding rate. In this process, we demonstrate that standard RS can also achieve a near-optimal coding rate and generalize the result of Braverman and Garg (2014) to the continuous alphabet setting. Next, as our main contribution, we present a distributed matching lemma for ERS, which serves as the rejection sampling counterpart to the Poisson Matching Lemma (PML) introduced by Li and Anantharam (2021). Our result also generalizes a recent work on importance matching lemma (Phan et al, 2024) and, to our knowledge, is the first result on distributed matching in the family of rejection sampling schemes where the matching probability is close to PML. We demonstrate the practical significance of our approach over prior works by applying it to distributed compression. The effectiveness of our proposed scheme is validated through experiments involving synthetic Gaussian sources and distributed image compression using the MNIST dataset.

## 1   Introduction

One-shot channel simulation is a task of efficiently compressing a finite collection of noisy samples. Specifically, this can be described as a two-party communication problem where the encoder obtains a sample $X \sim P_X$ and wants to transmit its noisy version $Y \sim P_{Y|X}$ to the decoder, with the communication efficiency measured by the coding cost $R$ (bits/sample), see Figure 1 (left). Since the conditional distribution $P_{Y|X}$ can be designed to target different objectives, channel simulation is a generalized version of lossy compression. As a result, it has been widely adopted in various machine learning tasks such as data/model compression [1, 4, 46, 19], differential privacy [37, 42], and federated learning [23]. While much of the prior work has focused on the point-to-point setting described above, recent research has extended channel simulation techniques to more general distributed compression scenarios [27, 35]. These scenarios often follow a canonical setup, shown in Figure 1 (middle, right), in which the encoder (party $A$) and the decoder (party $B$) each aim to generate samples $Y_A$ and $Y_B$, respectively, according to their own target distributions $P_Y^A$ and $P_Y^B$, using a shared source of randomness $W$. Although their sampling goals may differ, the selection processes are coupled through $W$, resulting in a non-negligible probability that both parties select the same output. We refer to this quantity as the *distributed matching probability*, which can be leveraged to reduce communication overhead in distributed coding schemes. For example, in the Wyner-Ziv setup [45], where the decoder has access to side information unavailable to the encoder, this framework enables the design of efficient one-shot compression protocols [35].

39th Conference on Neural Information Processing Systems (NeurIPS 2025).

Currently, Poisson Monte Carlo (PMC) [32] and importance sampling (IS) are the two main Monte Carlo methods being applied across both scenarios [29]. Particularly, the Poisson Functional Representation Lemma (PFRL) [28] provides a near-optimal coding cost for channel simulation. The Poisson Matching Lemma (PML) [27] was later developed for distributed matching scenarios, enabling the analysis of achievable error rates in various compression settings. However, PMC requires an infinite number of proposals, which can cause certain issues involving termination of samples in a practical scenario when the density functions, typically $P_Y^B$, are estimated via machine learning. IS-based approaches, including the importance matching lemma (IML) for distributed compression [35], bypass this issue by limiting the number of proposals in $W$ to be finite. Yet, the output distribution from IS is biased [19, 41], and thus not favorable in certain applications. It is hence interesting to see whether a new Monte Carlo scheme and coding method can be developed to handle both scenarios without compromising sample quality or termination guarantees.

This work studies rejection sampling schemes and its applicability to these two scenarios. We begin by revisiting and improving the coding efficiency of standard rejection sampling (RS) in channel simulation [40, 41]. In particular, we introduce a new coding scheme based on sorting that attains a near-optimal coding cost, extending the prior achievability result by Braverman and Garg [5] for discrete distributions to broader settings, while employing a distinct mechanism. However, our analysis also suggests that the distributed matching probabilities of both RS and its adaptive variant, namely greedy rejection sampling (GRS)[14], are lower than that of the PML, making them less suitable for distributed compression. Interestingly, we find that by combining RS with IS—a technique known as *Ensemble Rejection Sampling* (ERS) [9]—one can improve the distributed matching probability without degrading the sample quality. We demonstrate, with provable guarantees, that ERS retains efficient coding performance in channel simulation and can be naturally extended to distributed compression settings where the target distribution $P_Y^B$ must be learned using machine learning, typically encountered in high-dimensional data scenarios.

In summary, our contribution is as follows:

1. We propose a new compression method for RS that achieves a coding cost near the theoretical optimum. However we also argue that RS and its variant GRS do not achieve competitive performance in distributed matching.

2. We analyze ERS and show that it achieves competitive performance in distributed matching compared to PML and IML, while maintaining a coding cost close to the theoretical optimum in channel simulation.

3. We propose a practical distributed compression scheme based on ERS, supported by theoretical guarantees. We demonstrate the benefits of our approach through experiments on synthetic Gaussian sources and the MNIST image dataset.

Finally, we note that the term *distributed matching* in this paper encompasses settings that differ across communities. Without communication, it aligns with the problem of *correlated sampling* [3] in theoretical computer science, while classical *coupling* does not capture scenarios with limited communication. To stay consistent with the setups in Li and Anantharam [27], we use *distributed matching* to refer to these scenarios, to be discussed in Section 3.2.

## 2 Related Work

*Channel Simulation.* Our work introduces a novel channel simulation algorithm based on standard RS and ERS [9]. Our results enhance the coding efficiency compared to prior works [17, 41, 40] for standard RS and extend the best-known results for RS [5] to continuous settings. A related and more widely studied scheme in channel simulation is greedy rejection sampling (GRS), which can achieve a near-optimal coding cost. However, GRS is also more computationally intensive when applied to continuous distributions [14, 18, 16] as it requires iteratively evaluating a complex and potentially intractable integral. Our work studies ERS, the generalized version of standard RS, and shows a new coding scheme to achieve a near-optimal bound for a continuous alphabet. The ERS-based algorithm can be considered as an extension of the IS-based method for exact sampling setting [35, 41] and serves as a complementary approach to existing exact algorithms, such as the PFRL [28] and its faster variants [12, 15, 21]. Finally, there exist other channel simulation methods, though these are restricted to specific distribution classes [1, 24, 39].

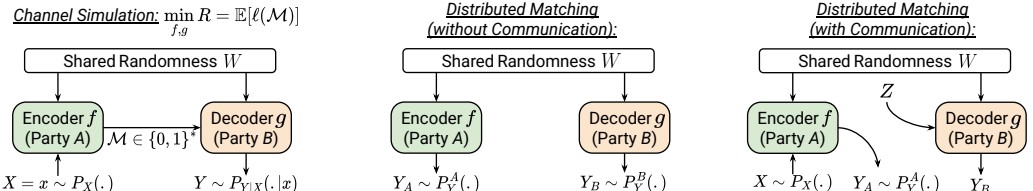

Figure 1: *Left*: Channel simulation setup. *Middle*: Distributed matching without communication. *Right*: Distributed matching with communication where the decoder's input $Z \sim P_{Z|X,Y_A}$ represents side information and/or messages from the encoder.

*Distributed Compression.* In distributed compression, one requires a generalized form of channel simulation, i.e. distributed matching, to reduce the coding cost, with current approaches include PML [27] and IML [35], as discussed earlier. Prior work has examined the matching probability of standard RS in various settings, primarily for discrete alphabets [8, 36]. Our method builds on ERS, a new RS-based scheme, and shows that its performance in distributed matching is comparable to PML, enabling practical applications in distributed compression. Other information-theoretic [30, 38, 43] and quantization-based approaches [31, 47] for this problem are generally impractical for implementation. Meanwhile, recent work has explored neural networks-based solutions [33, 44], with some provides empirical evidence that neural networks can learn to perform binning [34].

## 3 Problem Setup

### 3.1 Channel Simulation

Let $(X, Y) \in \mathcal{X} \times \mathcal{Y}$ be a pair of random variables with joint distribution $P_{X,Y}$, with $P_X$ and $P_Y$ are their respective marginal distributions. In this setup, see Figure 1 (left), the encoder observes $X = x \sim P_X(.)$ and wants to communicate a sample $Y \sim P_{Y|X}(.|x)$ to the decoder, with the coding cost of $R$ (bits/sample). Given that both parties share the source of common randomness $W \in \mathcal{W}$ independent of $X$, we define $f$ and $g$ to be the encoder and decoder mapping as follow:

$$f : \mathcal{X} \times \mathcal{W} \to \mathcal{M}; \qquad g : \mathcal{M} \times \mathcal{W} \to \mathcal{Y},$$

where the encoder message $M \in \mathcal{M} = \{0, 1\}^*$ is a binary string with length $\ell(M)$ and $R = \mathbb{E}[\ell(M)]$. Here, we require that the decoder's output follows $P_{Y|X}(.|x)$, i.e., $Y = g(f(x, W), W) \sim P_{Y|X}(.|x)$. Depending on the encoding and decoding function $f$ and $g$, the specification of what $\mathcal{W}$ includes varies. A general requirement for a channel simulation scheme to be efficient is that $R$ satisfies:

$$R \leq I(X;Y) + c_1 \log(I(X;Y) + c_2) + c_3, \tag{1}$$

where $I(X;Y)$ is the mutual information between $X$ and $Y$ and the theoretical optimal solution attainable in the asymptotic (i.e., infinite blocklength) setting [7]. Different techniques may produce slightly different coding costs, characterized by the positive constants $c_1, c_2$, and $c_3$[18, 26], but any approach that fails to achieve the leading term $I(X;Y)$ is generally considered inefficient.

### 3.2 Distributed Matching

We describe the two setups, with and without communication. Both setups consider two parties: $A$ (the encoder) and $B$ (the decoder) sharing a source of common randomness $W \in \mathcal{W}$.

#### 3.2.1 Distributed Matching Without Communication

In this setup, visualized in Figure 1 (middle), each party $A$ and $B$ aim to generate samples $Y_A$ and $Y_B$ from their respective distributions $P_Y^A$ and $P_Y^B$, which are locally available to each party, by selecting values from $W$. Each party constructs their respective mapping $f$ and $g$ as follows:

$$f : \mathcal{W} \to \mathcal{Y}, \quad g : \mathcal{W} \to \mathcal{Y},$$

with the requirement that $Y_A = f(W) \sim P_Y^A$ and $Y_B = g(W) \sim P_Y^B$. Following prior work on PML [27], we are interested in the lower bound of the conditional probability that both parties select

the same value, given that $Y_A = y$, with the following form:

$$\Pr(Y_A = Y_B \mid Y_A = y) \geq \Gamma(P_Y^A(y), P_Y^B(y)), \tag{2}$$

where in the case of PML, we have $\Gamma(P_Y^A(y), P_Y^B(y)) = (1 + P_Y^A(y)/P_Y^B(y))^{-1}$. For IML, $\Gamma(P_Y^A(y), P_Y^B(y)) = (1 + (1+\epsilon)P_Y^A(y)/P_Y^B(y))^{-1}$ where $\epsilon \to 0$ as the number of proposals increases.

### 3.2.2 Distributed Matching With Communication

In practice, communication from the encoder to the decoder is allowed to improve the matching probability. Also, the target distributions at each end may depend on their respective local inputs. Specifically, let $(X, Y, Z) \in \mathcal{X} \times \mathcal{Y} \times \mathcal{Z}$ be a triplet of random variables with joint distribution $P_{X,Y,Z}$. We first define the following mappings, also see Figure 1 (right):

$$f : \mathcal{X} \times \mathcal{W} \to \mathcal{Y}, \quad g : \mathcal{W} \times \mathcal{Z} \to \mathcal{Y},$$

where the protocol is as follows:

1. Encoder (party $A$): given $X = x \sim P_X$ independent of $W$, the encoder sets its target function $P_Y^A = P_{Y|X}(.|x)$ and selects a sample $Y_A = f(x, W) \sim P_Y^A$.

2. Given $X = x, Y_A = y$, we generate $Z = z \sim P_{Z|X,Y}(.|x,y)$, which can be thought as some noisy version of $(X, Y_A)$. Note that the Markov chain $Z - (X, Y_A) - W$ holds.

3. Decoder (party $B$): having access to $Z = z$, sets its target distribution to $P_Y^B(\cdot) = \tilde{P}_{Y|Z}(\cdot \mid z)$, where $\tilde{P}_{Y|Z}$ can be arbitrary. It then queries a sample $Y_B = g(W, z)$ from the source $W$.

The constraint $Y_B \sim P_Y^B$ is not necessarily satisfied, but this is not required in this setting [27], where the goal is to ensure the decoder selects the same value as the encoder with high probability. As in the case without communication, we are interested in establishing the bound with the following form:

$$\Pr(Y_A = Y_B \mid Y_A = y, Z = z, X = x) \geq \Gamma(P_Y^A(y), P_Y^B(y)), \tag{3}$$

where for PML and IML, $\Gamma(P_Y^A(y), P_Y^B(y))$ also follows the form discussed in Section 3.2.1.

**Remark 3.1.** *Since $Z - (X, Y_A) - W$ forms a Markov chain and $Z$ is input to the decoder, the communication in this setting happens by designing $P_{Z|X,Y}(\cdot \mid x, y)$ to include the encoder message. Finally, this setup generalizes the no-communication one by setting $(X, Z)$ to fixed constants.*

### 3.3 Bounding Condition

In this work, we often consider the ratio $P_Y(y)/Q_Y(y)$ to be bounded for all $y$, where $P_Y, Q_Y$ are the target and proposal distribution, respectively. We formalize this in Definition 3.2.

**Definition 3.2.** A pair of distributions $(P_Y, Q_Y)$ is said to satisfy a *bounding condition with constant* $\omega \geq 1$ if $\max_y P_Y(y)/Q_Y(y) \leq \omega$. Furthermore, let $(X, Y) \sim P_{X,Y}$, a triplet $(P_X, P_{Y|X}, Q_Y)$ satisfies an *extended bounding condition with constant* $\omega \geq 1$ if $\max_{x,y} P_{Y|X}(y|x)/Q_Y(y) \leq \omega$.

We note that the extended condition is practically satisfied when $(P_{Y|X=x}, Q_Y)$ satisfies the bounding condition for every $x$, and $P_X$ has bounded support.

## 4 Rejection Sampling

We review the existing coding scheme of standard RS and introduce a new technique that achieves a bound comparable to (1). We then discuss results on matching probability bounds for RS and GRS.

**Sample Selection.** We define the common randomness $W = \{(U_1, Y_1), (U_2, Y_2), \dots\}$, where each $U_i \overset{\text{i.i.d.}}{\sim} \mathcal{U}(0, 1)$ and each $Y_i \overset{\text{i.i.d.}}{\sim} P_Y$, and require that the triplet $(P_X, P_{Y|X}, P_Y)$ satisfies the extended bounding condition in Definition 3.2 with $\omega$. Given $X = x$, the encoder picks the first index $K$ where $U_K \leq \frac{P_{Y|X}(Y_K|x)}{\omega P_Y(Y_K)}$, obtaining $Y_K \sim P_{Y|X}(.|x)$.

**Runtime-Based Coding.** This approach encodes the sample following the entropy of $H(K)$. Since each individual sample $Y_i$ has the acceptance probability $\Pr(\text{Accept}) = \omega^{-1}$, we can compress $K$

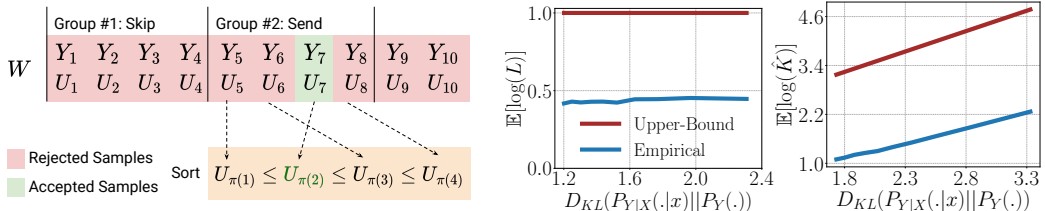

Figure 2: *Left*: Visualization of our Sorting Method for Standard RS. *Right*: Empirical results comparing $\mathbb{E}[\log(L)]$ and $\mathbb{E}[\log(\hat{K})]$ with their associated theoretical upper-bound across different target distribution. We use $P_Y(.) = \mathcal{N}(0, 1.0)$ and $P_{Y|X}(.|x) = \mathcal{N}(1.0, \sigma^2)$ where $\sigma^2 \in [0.01, 0.1]$.

with a coding cost of $R \leq H[K] + 1 \leq \log(\omega) + 2$, which is inefficient compared to $I(X; Y)$. For this reason, GRS is often preferred, but with practical limitations as discussed in Section 2.

**Our Approach.** Unlike the previous method, where the coding of $K$ is independent of $W$, we aim to design a scheme that leverages the availability of $W$ at both parties, thereby reducing the coding cost $R$ through the conditional entropy $H[K \mid W]$. Our *Sorting Method* operates on this idea, where instead of sending $K$, we send the rank of $U_K$ within a subset in $W$. Assume that the encoder and decoder agree on the value of $\omega$ prior to communication, we first collect every $\lfloor \omega \rfloor$ proposals into one group, ( $\lfloor . \rfloor, \lceil . \rceil$ are floor and ceil functions respectively). We encode two messages: one for the group index $L$ and one for the rank $\hat{K}$ of the selected $U_K$ within that group, in particular:

1. *Encoding $L$*: The encoder sends the ceiling $L = \lceil \frac{K}{\lfloor \omega \rfloor} \rceil$, i.e. $L = 2$ in Figure 2 (left). The decoder then knows $(L-1)\lfloor \omega \rfloor + 1 \leq K \leq L\lfloor \omega \rfloor$, i.e. $K$ is in group $L$.

2. *Encoding $\hat{K}$*: The encoder and decoder sort the list of $U_i$ for $(L-1)\lfloor \omega \rfloor + 1 \leq i \leq L\lfloor \omega \rfloor$:

$$U_{\pi(1)} \leq U_{\pi(2)} \leq ... \leq U_{\pi(\lfloor \omega \rfloor)}$$

where $\pi(.)$ maps the sorted indices with the original ones. The encoder sends the rank of $U_K$ within this list, i.e. sends the value $\hat{K}$ such that $K = \pi(\hat{K})$, which the decoder uses to retrieve $Y_K$ accordingly. This corresponds to $\hat{K} = 2$ in Figure 2 (left).

**Coding Cost.** In terms of the coding cost at each step, i.e., $\mathbb{E}[\log L]$ and $\mathbb{E}[\log \hat{K}]$, we have:

$$\mathbb{E}[\log L] \leq 1 \text{ bit} , \quad \mathbb{E}[\log \hat{K}] \leq D_{KL}(P_{Y|X}(.|x)||P_Y(.)) + \log(e) \text{ bits}, \tag{4}$$

where Figure 2 (right) shows the empirical results verifying the bounds. The proof for these bounds are shown in Appendix B.2. We then perform entropy coding for each message separately using Zipf's distribution and prefix-free coding. Proposition 4.1 shows their overall coding cost:

**Proposition 4.1.** *Given $(X, Y) \sim P_{X,Y}$ and $K$ defined as above. Then we have:*

$$R \leq I(X; Y) + \log(I(X; Y) + 1) + 9, \tag{5}$$

*Proof: See Appendix B.4.*

Note that the approach of Braverman and Garg [5] for discrete distributions can be extended to the continuous case, included in Appendix B.1 for completeness. Our sorting mechanism is fundamentally different and can be extended to the more general ERS framework, where incorporating the method of Braverman and Garg [5] is nontrivial.[1]

**Distributed Matching.** In distributed matching setups in Section 3.2 where both parties use standard RS to select samples from their respective distributions, we show in Appendix C.2 that RS performance is not as strong compared to PML and IML. For GRS, we provide an analysis via a non-trivial example in Appendix D.2, where we managed to construct target and proposal distributions such that $\Pr(Y_A = Y_B \mid Y_A = y) \to 0.0$, even when $P_Y^A(y) = P_Y^B(y)$. In contrast, this probability is greater than $1/2$ for PML, thus concluding that RS and GRS are less efficient compared to PML and IML.

---

[1]At the time of acceptance, we became aware of a concurrent work by [13] proposing a similar approach; however, their formulation requires communicating additional information, resulting in a slightly suboptimal coding cost compared to ours.

# 5 Ensemble Rejection Sampling

We show that ERS[9], an exact sampling scheme that combine RS with IS, can improve the matching probability and maintain a coding cost close to the theoretical optimum in channel simulation.

## 5.1 Background

**Setup and Definitions.** We begin by defining the common randomness $W$, which includes a set of exponential random variables to employ the Gumbel-Max trick for IS [35, 41], i.e.:

$$W = \{(B_1, U_1), (B_2, U_2), ...\}, \text{ where } U_i \sim \mathcal{U}(0,1) \tag{6}$$

$$B_i = \{(Y_{i1}, S_{i1}), (Y_{i2}, S_{i2}), ..., (Y_{iN}, S_{iN})\}, \text{ where } Y_{ij} \sim P_Y(.), S_{ij} \sim \text{Exp}(1), \tag{7}$$

where we refer to each $B_i$ as a batch. A selected sample $Y_K$ from $W$ is defined by two indices: the *batch index* $K_1$ and the *local index* in $B_{K_1}$, denoted as $K_2$. Its *global index* within $W$ is $K$, where $K = (N-1)K_1 + K_2$ and we write $Y_K \triangleq Y_{K_1, K_2}$.

**Sample Selection.** Consider the target distribution $P_{Y|X}(.|x)$, for each batch $B_i \in W$, the ERS algorithm selects a candidate index $K_i^{\text{cand}}$ via Gumbel-max IS and decides to accept/reject $Y_{K_i^{\text{cand}}}$ based on $U_i$. This process ensures that the accepted $Y_K \sim P_{Y|X}(\cdot|x)$ and is denoted for simplicity as:

$$K = \text{ERS}(W; P_{Y|X=x}, P_Y), \tag{8}$$

where the procedure is shown in Figure 3 (top, left) and Algorithm 1 in Appendix E.1. This procedure assumes the bounding condition holds for $(P_{Y|X}(y|x), P_Y(y))$ with $\omega$. The target and proposal distributions can be any, e.g., replacing $P_Y$ with $Q_Y$, as long as the bounding condition holds.

**Remark 5.1.** *Since the accept/reject operation happens on the whole batch $B_i$, we define the batch average acceptance probability as $\Delta$ (see Appendix E.1) where $\Delta \to 1.0$ as $N \to \infty$ and $N^* = N\Delta^{-1}$ as the average number of proposals (or runtime) required for ERS.*

## 5.2 Channel Simulation with ERS

For $N = 1$, ERS becomes the standard RS and thus achieves the coding cost shown in Proposition 4.1. When $N \to \infty$, we have the batch acceptance probability $\Delta \to 1.0$, meaning that we mostly accept the first batch and thus achieve the coding cost of Gumbel-max IS schemes [35, 41], which follows (1). This section presents the result for general $N$, which is more challenging to establish as discussed below. We assume the extended bounding condition in Definition 3.2 holds for $(P_X, P_{Y|X}, P_Y)$.

**Encoding Scheme.** We view the selection of $K_1$ as a rejection sampling process on a whole batch (see Appendix E.1) and apply the Sorting Method to encode $K_1$. Specifically, we collect every $\lfloor \Delta^{-1} \rfloor$ batches into one *group of batches*, send the group index and the rank of $U_{K_1}$ within this group. For the local index $K_2$, we use the Gumbel-Max Coding approach [35]. This process is visualized in Figure 3 (middle), detailed as follow:

- Encoding $K_1$: we represent $K_1$ by two messages $L$ and $\hat{K}_1$. Here, $L$ is the group of batches index $K_1$ belongs to and $\hat{K}_1$ is the rank of $U_{K_1}$ within this $L^{\text{th}}$ group, i.e., we sort the list: $U_{\phi(1)} \leq U_{\phi(2)} \leq ... \leq U_{\phi(\lfloor \Delta^{-1} \rfloor)}$ and send the rank $\hat{K}_1$ of $U_{K_1}$, i.e. $\phi(\hat{K}_1) = K_1$.
- Encoding $K_2$: We first sort the exponential random variables within the selected batch $K_1$, i.e. $S_{\pi(1)} \leq S_{\pi(2)} \leq ... \leq S_{\pi(N)}$ and send the rank $\hat{K}_2$ of $S_{K_2}$, i.e. $\pi(\hat{K}_2) = K_2$.

**Coding Cost.** We outline the main results for the coding costs, details in Appendix E. Specifically:

$$\mathbb{E}[\log L] \leq 1 \text{ bit}, \mathcal{K} = \mathbb{E}[\log \hat{K}_1] + \mathbb{E}[\log \hat{K}_2] \leq D_{KL}(P_{Y|X}(.|x)||P_Y(.)) + 2\log(e) + 3 \text{ bits}, \tag{9}$$

where the second bound is one of the core technical contributions of this work. We empirically validate the bound on $\mathcal{K}$ in Figure 3(right). Proposition 5.2 shows the overall coding cost for $K$:

**Proposition 5.2.** *Given $(X, Y) \sim P_{X,Y}$ and $K$ defined as above. For any batch size $N$, we have:*

$$R \leq I(X;Y) + 2\log(I(X;Y) + 8) + 12, \tag{10}$$

*Proof. See Appendix E.4.*

**Remark 5.3.** *The upper-bound in (10) is expected to be conservative, as evidenced by the evaluation of actual rates in Figure 3 (right). We further demonstrate the improvements in our proposed method over the baselines in the distributed compression application, to be elaborated upon in the subsequent discussion.*

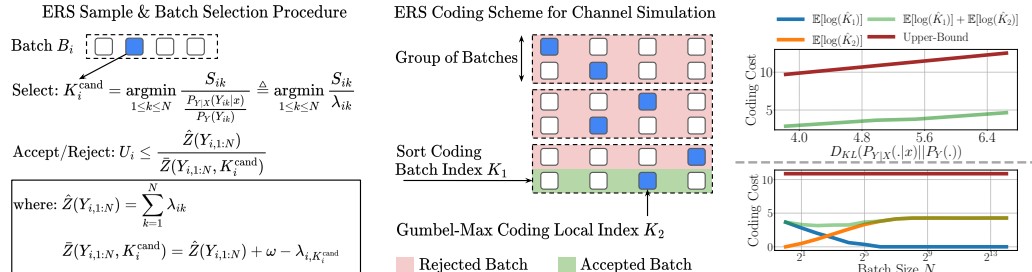

Figure 3: *Left*: Illustration of ERS Selection Method. *Middle:* Coding scheme for channel simulation. *Right*: Empirical results on the coding cost of $\hat{K}_1$, $\hat{K}_2$ and their theoretical upper-bound (in bits). Both figures use $P_Y(.)=\mathcal{N}(0, 1.0)$, where the first figure sets $N = 32$ and varies $P_{Y|X}(.|x)=\mathcal{N}(1.0, \sigma^2)$ with $\sigma^2 \in [0.1, 5] \times 10^{-3}$. The second one fixes $\sigma^2 = 10^{-3}$ while varying $N$.

## 5.3 Distributed Matching Probabilities

We consider the communication setup described in Section 3.2.2, which generalizes the no-communication one in Section 3.2.1, see Remark 3.1. We use subscripts to distinguish the indices selected by each party, e.g., $K_A$ and $K_B$ denote the global indices chosen by the encoder (party A) and decoder (party B), respectively. Recall that the encoder observes $X=x\sim P_X$ and sets $P_Y^A=P_{Y|X}(\cdot \mid x)$, while the decoder observes $Z=z$ and sets $P_Y^B=\tilde{P}_{Y|Z}(\cdot \mid z)$, not necessarily follow $P_{Y|Z}(\cdot \mid z)$. The target distributions $P_Y^A$, $P_Y^B$, and the proposal distribution $Q_Y$ in $W$ must satisfy the bounding conditions outlined in Section 3.3 for the ratio pairs $(P_Y^A, Q_Y)$ and $(P_Y^B, Q_Y)$. Each party then uses ERS to select their indices:

$$K_A = \mathrm{ERS}(W; P_Y^A, Q_Y), \quad K_B = \mathrm{ERS}(W; P_Y^B, Q_Y), \tag{11}$$

where the function $\mathrm{ERS}(.)$ is defined in (8) and we set $Y_A=Y_{K_A}$ and $Y_B=Y_{K_B}$ as the values reported by each party. Proposition 5.4 shows a bound on the matching probability in this setting. The bound for the no-communication case naturally follows with appropriate modification, see Appendix F.2.

**Proposition 5.4.** *Let $K_A, K_B, P_Y^A$ and $P_Y^B$ defined as above. For $N \geq 2$, we have:*

$$\Pr(Y_A = Y_B | Y_A = y, X = x, Z = z) \geq \left(1 + \mu_1'(N) + \frac{P_Y^A(y)}{P_Y^B(y)}\left(1 + \mu_2'(N)\right)\right)^{-1}, \tag{12}$$

*where $\mu_1'(N)$ and $\mu_2'(N)$ defined in Appendix F.5 are decay coefficients depending on the distributions where $\mu_1'(N), \mu_2'(N) \to 0$ as $N \to \infty$ with rate $N^{-1}$ under mild assumptions on the distributions $P_Y^A(.), P_Y^B(.)$ and $Q_Y(.)$.*

*Proof: See Appendix F.6.*

**ERS with Batch Index Communication.** In practice, $P_Y^B(y)$ is often learned via deep learning, making it difficult to obtain the upper bound for $P_Y^B(y)/Q_Y(y)$, thus preventing a well-defined select condition. A practical workaround is for the encoder to transmit the selected batch index $K_{1,A}$ to the decoder, limiting the search space to a finite subset. This aligns with Section 3.2.2 by incorporating $K_{1,A}$ into the construction of $Y_A$, $Z$, and $Y_B$. Its matching bound, see Appendix G, is similar to Proposition 5.4, but with different decaying coefficients.

**Remark 5.5.** *Since the decay coefficients $\mu_1'(N), \mu_2'(N) \to 0.0$ with the rate $N^{-1}$, for any small $\epsilon$ one can choose $N > N_0(\epsilon)$ such that $\mu_1'(N), \mu_2'(N) \leq \epsilon$.*

**Empirical Results.** Figure 4 (left, middle) validates and compares ERS matching probability (with and without batch communication) with PML and IML, where we see both ERS approaches converge to PML performance. For the same average number of proposals $N^*$, Figure 4 (middle) demonstrates that ERS (with batch index communication) achieves consistently higher matching probabilities than IS, while maintaining an unbiased sample distribution. For completeness, Figure 4 (right) shows the bias of IS can remain high even when the number of proposals is sufficiently large, i.e. $4\omega$. We discuss the overhead of the batch index in Section 5.3.1 on application to distributed compression.

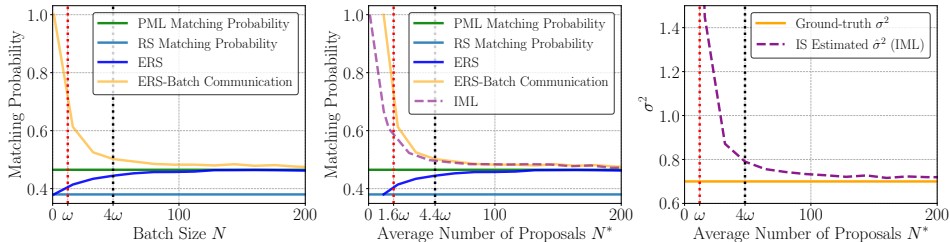

Figure 4: (Best viewed in color) We set $Q_Y = \mathcal{N}(0, 100)$, $P_Y^A = \mathcal{N}(0.5, 0.7)$ and $P_Y^B = \mathcal{N}(-0.5, 0.7)$. *Left*: Matching probabilities versus the batch size $N$. *Middle*: Matching probabilities versus the average number of proposals where the **red** and **black** dotted lines correspond to the batch sizes $\omega$ and $4\omega$ shown in the left figure. *Right*: Sample quality of IS, measured by the estimated variance $\hat{\sigma}^2$.

### 5.3.1 Lossy Compression with Side Information

We apply our matching result with batch index communication to the Wyner-Ziv distributed compression setting [45], where the encoder observes $X = x \sim P_X$ and the decoder has access to correlated side information $X' \sim P_{X'|X}(\cdot|x)$ unavailable to the encoder. Let $P_{Y'|X}(\cdot|x)$ denote the target distribution that the encoder aims to simulate, which, together with $X'$, induces the joint distribution $P_{X,X',Y'}$. For any integer $\mathcal{V} > 0$ and $U_i \sim \mathcal{U}(0,1)$, we set $Y_{ij} = (Y'_{ij}, V_{ij})$ in batch $B_i$ within $W$ where:

$$Y'_{ij} \sim Q_{Y'}(\cdot) \text{ (i.e., the ideal output)}, \quad V_{ij} \sim \text{Unif}[1{:}\mathcal{V}] \text{ (i.e., the hash value for index } j)$$

The main idea is, after selecting the index $K_A$ where $Y_{K_A} \sim P_{Y'|X=x}$, the encoder sends its hash $V_{K_A}$ along with the batch index $K_{1,A}$ to the decoder. The decoder, on the other hand, aims to infer $K_A$ by using the posterior $P_{Y'|X'=x'}$. The message $(V_{K_A}, K_{1,A})$ from the encoder will further reduce the decoder's search space within $W$ and improves the matching probability (details in Appendix H). Proposition 5.6 provides a bound on the probability the decoder outputs a wrong index:

**Proposition 5.6.** *Fix any $\epsilon > 0$ and let $(P_X, P_{Y'|X}, Q_{Y'})$ satisfies the extended bounding condition with $\omega$, for $N \geq \max(N_0(\epsilon), \omega)$ where $N_0(\epsilon)$ is defined in Remark 5.5, we have:*

$$\Pr(Y'_{K_A} \neq Y'_{K_B}) \leq \mathbb{E}_{X,Y',X'}\left[1 - \left(1 + \epsilon + (1+\epsilon)\mathcal{V}^{-1}2^{i(Y';X) - i(Y';X')}\right)^{-1}\right] \quad (13)$$

*where $i_{Y';X}(y';x) = \log P_{Y'|X}(y'|x) - \log P_{Y'}(y')$ is the information density. The coding cost is $\log(\mathcal{V}) + r$ where $r$ is the coding cost of sending the selected batch index $K_{1,A}$ and $r \leq 4$ bits.*

*Proof: See Appendix H*

**Remark 5.7.** *We can reduce the overhead $r$ in Proposition 5.6 by jointly compressing $n$ i.i.d. samples, i.e., to $4/n$ per sample. This also improves the matching probability in practice (see Appendix H).*

## 6 Experiments

We study the performance of ERS in the Wyner-Ziv distributed compression setting on synthetic Gaussian sources and MNIST dataset. All experiments are conducted on a single NVIDIA RTX A-4500. We use the batch communication version of ERS and encode the index with unary coding. Finally, we use the following formula from IS literature [6] as a starting point for choosing the batch size: $N = 2^{\mathbb{E}_X[D_{KL}(P_{Y|X}(\cdot|x)||Q_Y(\cdot)] + t}$ where $t \geq 4$ often gives $\Delta \geq 0.5$, resulting in a small overhead $r$.

### 6.1 Synthetic Gaussian Sources

We study and compare the performance of ERS, IML and PML in the Gaussian setting. Let $X \sim \mathcal{N}(0, \sigma_X^2)$ with $\sigma_X^2 = 1$ and is truncated within the range $[-2, 2]$ and the side information $X' = X + \zeta$ where $\zeta \sim \mathcal{N}(0, \sigma_{X'|X}^2)$ and $\sigma_{X'|X}^2 = 0.01$. The proposal and target distributions are $Q_{Y'}(.) = \mathcal{N}(0, \sigma_{Y'}^2)$, $P_{Y'|X}(.|x) = \mathcal{N}(x, \sigma_{Y'|X}^2)$, $P_{Y'|X'}(.|x') = \mathcal{N}\left(x'\sigma_X^2/\sigma_{X'}^2, \sigma_{Y'}^2 - \sigma_X^4/\sigma_{X'}^2\right)$ where $\sigma_{Y'}^2 = \sigma_X^2 + \sigma_{Y'|X}^2$, $\sigma_{X'}^2 = \sigma_X^2 + \sigma_{X'|X}^2$, and $\sigma_{Y'|X}^2$ is a fixed variance corresponding to the desired distortion level set by the encoder. The expression for $P_{Y'|X'}(\cdot|x')$ is an approximation derived from

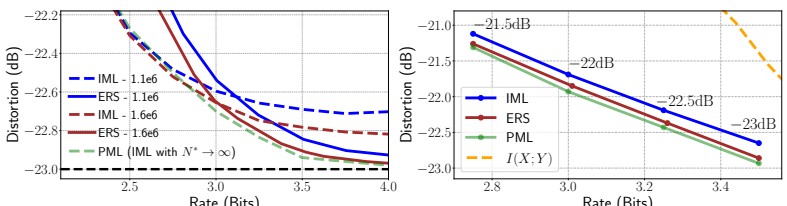

Figure 5: *Left*: Comparison of RD performance between different matching results for the Gaussian setting when targeting $-23$dB distortion (**black dotted line**), with the average number of proposals $N^* \in \{1.1\text{e}6, 1.6\text{e}6\}$. *Right*: RD curves of different methods. Each group targets the same distortion levels and uses the same average number of proposals $N^*$ for ERS and IML, shown in the right table.

the posterior distribution assuming $X$ is unbounded (i.e., not truncated). We jointly compress 4 i.i.d. samples to improve rate-distortion (RD) performance and average the result over $10^6$ runs.

Figure 5 (left) investigates the RD tradeoff between ERS and IML with similar number of proposals (on average) $N^*$ while targeting a distortion level of $-23$dB, i.e. $\sigma^2_{Y'|X}=5\text{e}^{-3}$. We observe that ERS outperforms IML in distortion regimes close to the target level, i.e. below $-22.6$dB as the rate increases, since IML samples are inherently biased. This bias also causes IML, with $N^* = 1.6\text{e}^6$, to be less effective than ERS, with $N^* = 1.1\text{e}^6$, for a distortion regime lower than $-22.8$dB, despite having more samples. Also, the batch index conveys information that helps improve the matching probability, similar to Figure 4 (middle), compensating for the overhead. Overall, for appropriately chosen $N^*$, ERS is more effective than IML on achieving low distortion levels while remaining competitive compared to PML, which is unbiased and requires no extra overhead.

In Figure 5 (right), we plot the RD tradeoff at different target distortion levels. We compare the distortion achieved by different methods at the rate where ERS reaches distortion within approximately $0.2$ dB of the target. Again, for appropriately chosen batch size $N$ and rate, ERS outperforms IML due to the inherent bias in importance sampling, and achieves performance close to that of PML. Note that PML does not generalize to practical setting when $P_Y^B$ is estimated via machine learning as the decoder cannot determine the number of samples upfront. In general, all three approaches outperform the asymptotic baseline $I(X;Y)$ in which there is no side information. Finally, standard RS achieves $-17$ dB at 10 bits when targeting $-23$ dB, falling outside the plotted range.

## 6.2 Distributed Image Compression

We apply our method in the task of distributed image compression [44, 33] with the MNIST dataset[25]. Following the setup in [35], the side information is the cropped bottom-left quadrant of the image and the source is the remaining. To reduce the complexity caused by high dimensionality, we use an encoder neural network to project the data into a 3D embedding space. This vector and the side information are input into a decoder network to output the reconstruction $\hat{X}$, and the process is trained end-to-end under the $\beta$-VAE framework. For each input $X = x$, we set the target distribution $P_{Y'|X}(\cdot|x) = \mathcal{N}(\mu(x), \sigma^2(x))$ where $\mu(\cdot), \sigma(\cdot)$ are the outputs of the $\beta$-VAE network. Since $P_{Y'|X'}$ is unknown, we employ a neural contrastive estimator [22] to learn the ratio between $P_{Y'|X'}(y'|x')/Q_{Y'}(y')$ from data, where $Q_{Y'}=\mathcal{N}(0, 1)$. Since the upperbound of this ratio is unknown, PML cannot be applied [41]. Models and training details are in Appendix J and K.

Extending the scope of the previous experiment, we study the interaction between matching schemes and feedback mechanisms for error correction, introduced in previous IML work [35]. Here, the decoder returns its retrieved index to the encoder, which then confirms or corrects it with the cost of 1 plus $\log(N/\mathcal{V})$ for ERS and $\log(N^*_{\text{IML}}/\mathcal{V})$ for IML, see Appendix I. This is relevant when aiming to mitigate mismatching errors or to generate samples that closely follow the encoder's target distribution, as in applications such as differential privacy. Since IML produces biased samples, we reduce this bias by setting the number of proposals to the maximum feasible value in our simulation system, i.e., $N^*_{\text{IML}} = 2^{26}$, ensuring it exceeds the ones used by ERS, denoted $N^*_{\text{ERS}}$, in this experiment.

We train four models, each targeting a different pixel distortion level, and compare their performance in Figure 6, where two samples are compressed jointly. With feedback, ERS consistently outperforms IML in both embedding and pixel domains. This is because the feedback scheme in IML incurs a

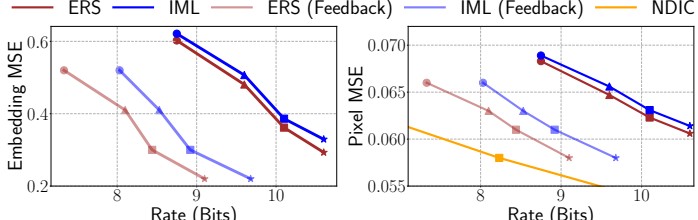

Figure 6: MNIST Rate-distortion comparison for pixels, i.e. $||X - \hat{X}||_2^2$ and embeddings domain, i.e. $||\mu(X) - Y'||_2^2$, between ERS and IML. Identical markers (from top to bottom) indicate the same target models, with the target distortion levels corresponding to those achieved using feedback.

higher return message cost due to the large $N_{\text{IML}}^*$, while still introducing slight bias in its output samples. In contrast, ERS operates with a smaller batch size $N$, significantly reducing the correction message size without compromising the sample quality. Without feedback, under a distortion regime close to the target level, ERS outperforms IML for reasons discussed in the Gaussian experiment, though the performance gap is smaller. We include NDIC results [33]—a specialized deep learning approach that targets optimal RD performance. On the other hand, our method operates on a probabilistic matching nature and can accommodate scenarios with distributional constraints.

## 7 Conclusion

This work explores the use of the RS-based family for channel simulation and distributed compression. We focus on ERS where we develop a new efficient coding scheme for channel simulation and derive a performance bound for distributed compression that is comparable to PML [27]. We validate our theoretical results on both synthetic and image datasets, showing their advantages and adaptability across various setups, including feedback-based error correction schemes. From these results, possible future directions include improving the current runtime efficiency—which is $O(\omega)$—by incorporating acceleration techniques such as space partitioning [21] or importance sampling methods like Multiple IS [11], as well as extending the distributed compression setup to incorporate differential privacy.

## Acknowledgment

Resources used in preparing this research were provided, in part, by the Province of Ontario, the Government of Canada through CIFAR, and companies sponsoring the Vector Institute www.vectorinstitute.ai/partnerships/.

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

## A  Runtime of ERS.

We provide an analysis of ERS runtime. Let $\omega = \max_{x,y} P_{Y|X}(y|x)/Q(y)$ and $\omega_x = \max_y P_{Y|X}(y|x)/Q(y)$, where $P_{Y|X=x}$ is the target distribution and $Q_Y$ is the proposal distribution. For the batch size $N$ and input $x$, we have the following bound on the average batch acceptance probability $\Delta_x$, which we will show in Appendix E.1:

$$\Delta_x \geq \frac{N}{N-1+\omega_x} \geq \frac{N}{N-1+\omega}, \tag{14}$$

Thus, the expected number of batches in ERS is:

$$\text{Expected Number of Batches} = \frac{1}{\Delta_x} \leq \frac{N-1+\omega}{N}, \tag{15}$$

which leads to the runtime, i.e. the expected number of proposals as:

$$\text{Expected Runtime} = \frac{N}{\Delta_x} \leq N-1+\omega. \tag{16}$$

In practice, since we typically choose $N = O(\omega)$, the expected runtime is also $O(\omega)$.

## B  Coding Cost of Standard Rejection Sampling

For the proof, we generalize and use $P(.)$ and $Q(.)$ as the target and proposal distributions. This allows shorthand the notations while also generalizing the results for arbitrary distributions.

### B.1  Extension of Braverman and Garg [5]'s Method for Continuous Setting

This method is an extension of the work by Braverman and Garg [5] to the continuous setting. The core idea is to divide the acceptance region into smaller bins, visualized in Figure 7. Specifically, for each pair $(U_i, Y_i)$ from $W$, we denote $\tilde{U}_i = \omega U_i Q(Y_i)$. The encoder selects the index $K$ according to rejection sampling rule, which is 7 in Figure 7. It then sends the bin index of the first accepted sample, where the bin corresponds to the smallest scaled region that $\tilde{U}_K$ belongs to. In Figure 7, this corresponds to the orange region and the content of the message is 3. Then the encoder sends another message which indicates the rank of the selected sample within that bin, which is 1. The decoder then $K$ accordingly. Formally, the two steps are as follow:

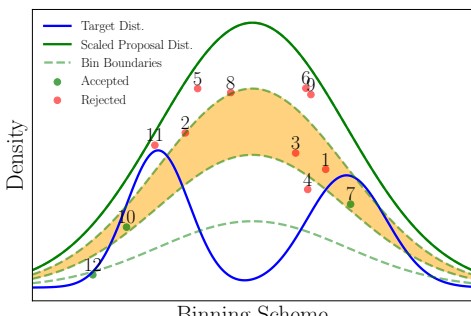

Figure 7: Binning Method for RS.

- *Binning*: The encoder sends to the decoder the ceiling $T = \lceil \frac{\tilde{U}_K}{Q(Y_K)} \rceil$. Upon receiving $T$, the decoder collects the set:

$$\mathcal{S}_T = \{i | (T-1)Q(Y_i) \leq \tilde{U}_i \leq TQ(Y_i)\}, \tag{17}$$

- *Index Selection*: The encoder locates the original chosen index $K$ within $\mathcal{S}_T$, says $G$, and send $G$ to the receiver. We have $\mathbb{E}[\log G] \leq 1$.

*Binning Step.* We will show the $\mathbb{E}[\log T] \leq D_{KL}(P||Q) + \log(e)$., adapting the proof for the discrete case presented in [36]. First, we note that:

$$Y_K \sim P(.), \quad U_K|Y_K \sim \mathcal{U}\left(0, \frac{P(Y_K)}{\omega Q(Y_K)}\right) \tag{18}$$

We then have:

$$\mathbb{E}[\log T] = \mathbb{E}\left[\log\left(\left\lceil \frac{\tilde{U}_K}{Q(Y_K)}\right\rceil\right)\right] \tag{19}$$

$$\leq \mathbb{E}\left[\log\left(1 + \frac{\tilde{U}_K}{Q(Y_K)}\right)\right] \tag{20}$$

$$= \mathbb{E}\left[\log\left(1 + \omega U_K\right)\right] \tag{21}$$

$$= \mathbb{E}\left[\mathbb{E}\left[\log\left(1 + \omega U_K\right)|Y_K\right]\right] \tag{22}$$

$$= \int_{-\infty}^{+\infty} P(y)\left[\frac{\omega Q(y)}{P(y)} \int_0^{\omega^{-1}P(y)/Q(y)} \log\left(1 + \omega u\right) du\right] dy \quad \text{(Due to (18))} \tag{23}$$

$$\leq \int_{-\infty}^{+\infty} P(y)\left[\frac{\omega Q(y)}{P(y)} \int_0^{\omega^{-1}P(y)/Q(y)} \log\left(1 + \frac{P(y)}{Q(y)}\right) du\right] dy \tag{24}$$

$$\leq \int_{-\infty}^{+\infty} P(y)\left[\frac{\omega Q(y)}{P(y)} \int_0^{\omega^{-1}P(y)/Q(y)} \log\left(\frac{P(y)}{Q(y)}\right) + \frac{Q(y)\log(e)}{P(y)} du\right] dy \tag{25}$$

$$= \int_{-\infty}^{+\infty} P(y)\log\left(\frac{P(y)}{Q(y)}\right) dy + \int_{-\infty}^{+\infty} Q(y)\log(e) dy \tag{26}$$

$$= D_{KL}(P||Q) + \log(e), \tag{27}$$

where we use the following results for the last inequality:

$$\log(1 + x) \leq \log(x) + \frac{\log(e)}{x} \quad \text{(for all } x > -1\text{).} \tag{28}$$

*Index Selection Step.* We first show that $\mathbb{E}[G] \leq 2$ by using recursion. We define $\mathcal{A}$ as an event where the first samples is accepted, i.e. $U_1 \leq \frac{P(Y_1)}{\omega Q(Y_1)}$. Then, if $\mathcal{A}$ happens then we have $G = 1$, i.e. $\mathbb{E}[G|\mathcal{A}] = 1$, since it is also the first sample in $\mathcal{S}_T$.

Before proceeding to the case where $\mathcal{A}$ does not happen, i.e. $\bar{\mathcal{A}}$, we define the following random variable $M = \mathbb{1}[1 \in \mathcal{S}_T]$, i.e. $M = 1$ if the first proposed sample from $W$ stays within the ceiling $(T - 1)Q(Y_1) \leq \tilde{U}_1 \leq TQ(Y_1)$ and $M = 0$ otherwise.

Then we have the two following recursion identities:

$$\begin{cases} \mathbb{E}[G|\bar{\mathcal{A}}, M = 0] = \mathbb{E}[G] \\ \mathbb{E}[G|\bar{\mathcal{A}}, M = 1] = 1 + \mathbb{E}[G] \end{cases} \tag{29}$$

For the first equality, given that the first sample $U_1, Y_1$ does not stay within $\mathcal{S}_T$ does not implies any information about $G$, since all the samples are i.i.d. For the second equality takes into account the fact that we now accept the first sample $(U_1, Y_1)$ and repeat the counting process. Hence, we have:

$$\mathbb{E}[G|\bar{\mathcal{A}}] = \Pr(M = 0|\bar{\mathcal{A}})\mathbb{E}[G|\bar{\mathcal{A}}, M = 0] + \Pr(M = 1|\bar{\mathcal{A}})\mathbb{E}[G|\bar{\mathcal{A}}, M = 1] \tag{30}$$

$$= \mathbb{E}[G] + \Pr(M = 1|\bar{\mathcal{A}}) \tag{31}$$

We now express $\mathbb{E}[G]$ as follows:

$$\mathbb{E}[G] = \Pr(A)\mathbb{E}[G|A] + \Pr(\bar{A})\mathbb{E}[G|\bar{A}] \tag{32}$$

$$= \Pr(A) + \Pr(\bar{A})(\mathbb{E}[G] + \Pr(M = 1|\bar{A})) \tag{33}$$

Rearranging the terms, we obtain:

$$\mathbb{E}[G] = 1 + \frac{\Pr(M = 1, \bar{A})}{\Pr(A)} \tag{34}$$

We have $\Pr(A) = \int_{-\infty}^{\infty} \omega^{-1} P(y) Q^{-1}(y) Q(y) dy = \omega^{-1}$. For $\Pr(M = 1, \bar{A})$, we have:

$$\Pr(M = 1, \bar{A}) \leq \Pr(M = 1) \tag{35}$$

$$= \sum_{t=0}^{\infty} \Pr((T - 1)Q(Y_1) \leq \tilde{U}_1 \leq (T - 1)Q(y), T = t) \tag{36}$$

$$= \sum_{t=0}^{\infty} \Pr((t - 1)Q(Y_1) \leq \tilde{U}_1 \leq (t - 1)Q(y)) \Pr(T = t) \tag{37}$$

$$= \sum_{t=0}^{\infty} \omega^{-1} \Pr(T = t) \tag{38}$$

$$= \omega^{-1} \tag{39}$$

Thus, we obtain $\mathbb{E}[G] \leq 2$ and hence $\mathbb{E}[\log G] \leq 1$.

## B.2  The Sorting Method

The encoding process is as follows:

- *Grouping*: the encoder sends the ceiling $L = \lceil \frac{K}{\lfloor \omega \rfloor} \rceil$ to the decoder. The decoder then knows $(L - 1)\omega + 1 \leq K \leq L\omega$, i.e. $K$ is in range $L$. We have $\mathbb{E}[\log L] = 1$ bit.
- *Sorting*: The encoder and decoder both sort the uniform random variables $U_i$ within the selected range $(L - 1)\lfloor \omega \rfloor + 1 \leq i \leq L\lfloor \omega \rfloor$. Let the sorted list be $U_{\pi(1)} \leq U_{\pi(2)} \leq ... \leq U_{\pi(\lfloor \omega \rfloor)}$ where $\pi(.)$ is the mapping between the sorted index and the original unsorted one. The encoder sends the rank of $U_K$ within this list, i.e. sends the value $\hat{K}$ such that $K = \pi(\hat{K})$. The decoder receive $\hat{K}$ and retrieve $Y_K$ accordingly. The coding cost for this step is $D_{KL}(P||Q) + \log(e)$.

We provide detail analysis for each step below:

*Grouping Step.* Since each proposal is accepted with probability $\omega^{-1}$, this means:

$$\Pr(K > \ell \lfloor \omega \rfloor) = \left(1 - \omega^{-1}\right)^{\ell \lfloor \omega \rfloor} < \left(\frac{1}{2}\right)^{\ell}, \tag{40}$$

where we will prove the RHS inequality in Appendix B.3. Hence, we have $\Pr(L > \ell) < \left(\frac{1}{2}\right)^{-\ell}$ and:

$$\mathbb{E}[L] = \sum_{\ell=0}^{\infty} \Pr(L > \ell) < 1 + 0.5^{-1} + 0.5^{-2} + ... = 2. \tag{41}$$

Finaly, using Jensen's inequality, we have:

$$\mathbb{E}[\log L] \leq \log(\mathbb{E}[L]) = 1. \tag{42}$$

*Sorting Step.* To bound the coding cost in step 2, we first express $\mathbb{E}[\log \hat{K}]$ with the rule of conditional expectation as follows:

$$\mathbb{E}[\log \hat{K}] = \int_{-\infty}^{\infty} P(y) \mathbb{E}[\log \hat{K}|Y_K = y] dy \tag{43}$$

$$= \int_{-\infty}^{\infty} P(y) \left( \int_{-\infty}^{\infty} \mathbb{E}[\log \hat{K}|Y_K = y, U_K = u] P(U_K = u|Y_K = y) du \right) dy \tag{44}$$

$$= \int_{-\infty}^{\infty} P(y) \left( \int_{0}^{\frac{P(y)}{\omega Q(y)}} \mathbb{E}[\log \hat{K}|Y_K = y, U_K = u] \frac{\omega Q(y)}{P(y)} du \right) dy, \tag{45}$$

where the last step, $P(U_K = u|Y_K = y) = \frac{\omega Q(y)}{P(y)}$ for $0 \leq u \leq \frac{P(y)}{\omega Q(y)}$ is due to the acceptance condition in rejection sampling. We will show in Section B.3.1 that:

$$\mathbb{E}[\log \hat{K}|Y_K = y, U_K = u] \leq \log(\omega u + 1) \tag{46}$$

Then, combining this with Equation (45), we obtain:

$$\mathbb{E}[\log \hat{K}] \leq \int_{-\infty}^{\infty} P(y) \left( \int_0^{\frac{P(y)}{\omega Q(y)}} \frac{\omega Q(y)}{P(y)} \log(\omega u + 1) du \right) dy \tag{47}$$

$$\leq \int_{-\infty}^{\infty} P(y) \left( \int_0^{\frac{P(y)}{\omega Q(y)}} \frac{\omega Q(y)}{P(y)} \log \left( \frac{P(y)}{Q(y)} + 1 \right) du \right) dy \tag{48}$$

$$= \int_{-\infty}^{\infty} P(y) \left[ \frac{P(y)}{\omega Q(y)} \frac{\omega Q(y)}{P(y)} \log \left( \frac{P(y)}{Q(y)} + 1 \right) \right] dy \tag{49}$$

$$= \int_{-\infty}^{\infty} P(y) \log \left( \frac{P(y)}{Q(y)} + 1 \right) dy \tag{50}$$

$$\leq \int_{-\infty}^{\infty} P(y) \left[ \log \left( \frac{P(y)}{Q(y)} \right) + \frac{\log(e) Q(y)}{P(y)} \right] dy \tag{51}$$

$$= D_{KL}(P||Q) + \log(e) \tag{52}$$

Hence, we have $\mathbb{E}[\log \hat{K}] \leq D_{KL}(P||Q) + \log(e)$ on average.

### B.3 Proof for Inequality (40)

The proof for this inequality is self-contained. We want to prove that for any $\omega \geq 1$, we have:

$$f(\omega) = (1 - \omega^{-1})^{\lfloor \omega \rfloor} \leq \frac{1}{2}. \tag{53}$$

Consider the behavior of $f(\omega)$ at every interval $[n, n+1)$ where $n \in \mathbb{Z}^+, n \geq 1$. Since $\omega \geq 1$, the function $f_n(\omega) = \left( 1 - \omega^{-1} \right)^n$ is increasing and hence:

$$\sup_{\omega} f_n(\omega) = \left( 1 - \frac{1}{n+1} \right)^n = \left( \frac{n}{n+1} \right)^n$$

for every interval $[n, n+1)$. We will show that $\sup_{\omega} f_n(\omega)$ is decreasing for $n \geq 1$ and thus we have $\sup_{\omega} f(\omega) = \sup_{\omega} f_1(\omega) = \frac{1}{2}$.

Consider the function $g(x) = \left( \frac{x}{x+1} \right)^n$ for $x \geq 1, x \in \mathbb{R}$. Let $h(x) = \ln(g(x)) = x \ln(\frac{x}{x+1})$, then we simply need to show $h(x)$ is decreasing. Consider its first derivative:

$$h'(x) = \ln \left( \frac{x}{x+1} \right) + \frac{1}{x+1} \leq 0, \tag{54}$$

since:

$$\ln \left( \frac{x}{x+1} \right) = \ln \left( 1 - \frac{1}{x+1} \right) \leq -\frac{1}{x+1} \tag{55}$$

due to the inquality $\ln(1 + y) < y$ for all $y$.

### B.3.1 Proof for Inequality (46)

We begin by applying Jensen's inequality for concave function $\log(x)$:

$$\mathbb{E}[\log \hat{K} | Y_K = y, U_K = u] \leq \log \mathbb{E}[\hat{K} | Y_K = y, U_K = u] \quad \text{(by Jensen's Inequality)} \tag{56}$$

$$= \log \mathbb{E}_L[\mathbb{E}[\hat{K} | Y_K = y, U_K = u, L = \ell]] \tag{57}$$

Given $K$ is within the range $L = \ell$ and $U_K = u$, we can express $\hat{K}$ as follows:

$$\hat{K} = |\{U_i < u, (\ell - 1)\lfloor \omega \rfloor + 1 \leq i \leq \ell \lfloor \omega \rfloor\}| + 1, \tag{58}$$

$$= \Omega(u, \ell) + 1 \tag{59}$$

i.e. the number of $U_i$ (plus 1 for the ranking) within the range $L$ that has value lesser than $u$.

We can see that the the index $i$ within the range $L$ satisfying $U_i < u$ are from the index that are either (1) rejected, i.e. index $i < K$ or (2) not examined by the algorithm, i.e. index $i > K$. The rest of this proof will show the following upperbound:

$$\mathbb{E}[\Omega(u, \ell)|Y_K = y, U_K = u, L = \ell] \le \omega u, \text{ for any } \ell \tag{60}$$

For readability, we split the proof into different proof steps.

**Proof Step 1:** We condition on the mapped index of $\pi(\hat{K})$ on the original array:

$$\mathbb{E}[\hat{K}|Y_K = y, U_K = u, L = \ell] \tag{61}$$

$$= \mathbb{E}_{\pi(\hat{K})} \left[ \mathbb{E}[\hat{K} \mid Y_K = y, U_K = u, L = \ell, \pi(\hat{K}) = k] \right] \tag{62}$$

$$= \mathbb{E}_{\pi(\hat{K})} \left[ \mathbb{E}[\Omega(u, \ell) + 1 \mid Y_K = y, U_K = u, L = \ell, \pi(\hat{K}) = k] \right] \tag{63}$$

$$= \mathbb{E}_{\pi(\hat{K})} \left[ \mathbb{E}[\Omega(u, \ell) \mid Y_K = y, U_K = u, L = \ell, \pi(\hat{K}) = k] \right] + 1 \tag{64}$$

$$= \mathbb{E}_{\pi(\hat{K})} \left[ \mathbb{E}[\Omega_1(u, \ell, k) + \Omega_2(u, \ell, k) \mid Y_K = y, U_K = u, L = \ell, \pi(\hat{K}) = k] \right] + 1, \tag{65}$$

where $\Omega_1(u, \ell, k), \Omega_2(u, \ell, k)$ are the number of $U_i < u$ within the range $L = \ell$ that occurs before and after the selected index $k$ respectively. Specifically:

$$\Omega_1(u, \ell, k) = |\{U_i < u, (\ell - 1)\lfloor\omega\rfloor + 1 \le i < (\ell - 1)\lfloor\omega\rfloor + k\}| \tag{66}$$
$$\Omega_2(u, \ell, k) = |\{U_i < u, (\ell - 1)\lfloor\omega\rfloor + k + 1 \le i \le \ell\lfloor\omega\rfloor\}|, \tag{67}$$

which also naturally gives $\Omega(u, \ell) = \Omega_1(u, \ell, k) + \Omega_2(u, \ell, k)$.

**Proof Step 2:** Consider $\Omega_2(u, \ell, k)$, since each proposal $(Y_i, U_i)$ is i.i.d distributed and the fact that $k$ is the index of the accepted sample, for every $i > K$, we have:

$$\Pr(U_i < u \mid Y_K = y, U_K = u, L = \ell, \pi(\hat{K}) = k) = \Pr(U_i < u)$$

This gives us:

$$\mathbb{E}[\Omega_2(u, \ell, k) \mid Y_K = y, U_K = u, L = \ell, \pi(\hat{K}) = k] = (\lfloor\omega\rfloor - k)\Pr(U < u) \tag{68}$$
$$= (\lfloor\omega\rfloor - k)u \tag{69}$$
$$\le \frac{(\lfloor\omega\rfloor - k)u}{\Pr(\text{reject a sample})} \tag{70}$$
$$\le \frac{(\lfloor\omega\rfloor - k)u}{1 - \omega^{-1}} \tag{71}$$

**Proof Step 3:** For $\Omega_1(u, \ell, k)$, we do not have such independent property since for every sample with index $i < K$, we know that they are rejected samples, and hence for $i < k$:

$$\Pr(U_i < u \mid Y_K = y, U_K = u, L = \ell, \pi(\hat{K}) = k) = \Pr(U_i < u|Y_i \text{ is rejected}) \tag{72}$$
$$= \frac{\Pr(U_i < u, Y_i \text{ is rejected})}{\Pr(Y_i \text{ is rejected})} \tag{73}$$
$$\le \frac{\Pr(U_i < u)}{\Pr(Y_i \text{ is rejected})} \tag{74}$$
$$= \frac{u}{1 - \omega^{-1}}, \tag{75}$$

which gives us:

$$\mathbb{E}[\Omega_2(u, \ell, k) \mid Y_K = y, U_K = u, L = \ell, \pi(\hat{K}) = k] \le \frac{(k - 1)u}{1 - \omega^{-1}} \tag{76}$$

To prove Equation (72), note that the following events are equivalent:

$$\{Y_K = y, U_K = u, L = \ell, \pi(\hat{K}) = k\} = \{Y_k = y, U_k = u, Y_{1\ldots k-1} \text{ are rejected}\} \tag{77}$$
$$\triangleq \Lambda(u, y, k) \tag{78}$$

Here, we note that $Y_k, U_k$ denote the value at index $k$ within $W$, which is different from $Y_K, U_K$, the value selected by the rejection sampler. Hence:

$$\Pr(U_i < u | \Lambda(u, y, k)) = \frac{\Pr(U_i < u, Y_{1\ldots k-1} \text{ are rejected} | Y_k = y, U_k = u)}{\Pr(Y_{1\ldots k-1} \text{ are rejected} | Y_k = y, U_k = u)} \tag{79}$$

$$= \frac{\Pr(U_i < u, \ Y_{1\ldots k-1} \text{ are rejected})}{\Pr(Y_{1\ldots k-1} \text{ are rejected})} \quad \text{(Since } (Y_i, U_i) \text{ are i.i.d)} \tag{80}$$

$$= \Pr(U_i < u | Y_i \text{ is rejected}), \tag{81}$$

**Proof Step 4:** From the above result from Step 2 and 3, we have $\Omega(u, \ell) = \Omega_1(u, \ell, k) + \Omega_2(u, \ell, k) \le \omega u$ and as a result:

$$\mathbb{E}[K | Y_K = y, U_K = u, L = \ell] \le \frac{(\lfloor \omega \rfloor - 1)u}{1 - \omega^{-1}} + 1 \tag{82}$$

$$\le \frac{(\omega - 1)u}{1 - \omega^{-1}} + 1 \quad \text{(Since} \lfloor \omega \rfloor \le \omega) \tag{83}$$

$$= \omega u + 1 \tag{84}$$

which completes the proof.

## B.4 Overall Coding Cost.

We now provide the upperbound on $H[K]$ for our *Sorting Method*. Since the message in the *Binning Method* also consists of two parts, the results are the same. For each part of the message, namely $L$ and $K$, we encode it with a prefix-code from Zipf distribution [28]. For $H[L]$, we have:

$$H[L] \le \mathrm{E}_X[\mathrm{E}[\log L | X = x]] + \log(\mathrm{E}_X[\mathrm{E}[\log L | X = x]] + 1) + 1 \tag{85}$$

$$= 3 \text{ bits} \tag{86}$$

Hence, the rate for the first message is $R_1 \le H[L] + 1 = 4$ bits.

Similarly, for $H[\hat{K}]$:

$$H[\hat{K}] \le \mathbb{E}_X[\mathbb{E}[\log \hat{K} | X = x] + \log(\mathbb{E}_X[\mathbb{E}[\log \hat{K} | X = x] + 1) + 1 \tag{87}$$

$$= I(X; Y) + \log(e) + \log(I(X; Y) + \log(e) + 1) + 1 \tag{88}$$

$$\le I(X; Y) + \log(I(X; Y) + 1) + 2\log(e) + 1 \tag{89}$$

Hence, the rate for the second message is $R_2 \le H[\hat{K}] + 1 = I(X; Y) + \log(I(X; Y) + 1) + 2\log(e) + 2$bits . Also note that:

$$H[K | W] = H[L, \hat{K} | W] \quad \text{(Given } W, K \text{ and } (L, \hat{K}) \text{ are bijective )} \tag{90}$$

$$\le H[L | W] + H[\hat{K} | W] \tag{91}$$

$$\le H[L] + H[\hat{K}] \tag{92}$$

$$\le I(X; Y) + \log(I(X; Y) + 1) + 7 \text{ (bits)} \tag{93}$$

Since we are compressing two messages separately, we have: $R \le R_1 + R_2 = I(X; Y) + \log(I(X; Y) + 1) + 9$ (bits)

# C Matching Probability of Rejection Sampling

## C.1 Distributed Matching Probabilities of RS

Follow the setup in Section 3.2.1, each party independently performs RS using the proposal distribution $Q_Y(\cdot)$ to select indices $K_A$ and $K_B$ and set $(Y_A, Y_B) = (Y_{K_A}, Y_{K_B})$. We assume the bounding condition holds for both parties, i.e. $\max_y \left( P_Y^A(y)Q_Y^{-1}(y), P_Y^B(y)Q_Y^{-1}(y) \right) \le \omega$, Proposition C.1 shows the probability that they select the same index, given that $Y_{K_A} = y$.

**Proposition C.1.** *Let $W, Q(.), P_Y^A(.)$ and $P_Y^B(.)$ defined as above. Then we have:*

$$\Pr(Y_A = Y_B | Y_A = y) = \frac{\min(1, P_Y^B(y)/P_Y^A(y))}{1 + \mathrm{TV}(P_Y^A, P_Y^B)} \geq \frac{1}{2\left(1 + P_Y^A(y)/P_Y^B(y)\right)} \tag{94}$$

*Furthermore, we have:*

$$\Pr(Y_{K_A} = Y_{K_B}) = \frac{1 - \mathrm{TV}(P_Y^A, P_Y^B)}{1 + \mathrm{TV}(P_Y^A, P_Y^B)}. \tag{95}$$

*where $\mathrm{TV}(P_Y^A, P_Y^B)$ is the total variation distance between two distribution $P_Y^A$ and $P_Y^B$.*

This matching probability is not as strong, compared to PML as well as IML, details in Appendix C.2[2]. In the case of GRS, we provide an analysis via a non-trivial example in Appendix D.2, where we demonstrate that it is possible to construct target and proposal distributions such that $\Pr(K_A = K_B \mid Y_{K_A} = y) \to 0.0$, even when $P_Y^A(y) = P_Y^B(y)$. In contrast, this probability is greater than $1/4$ for standard RS. In summary, while GRS and RS can achieve a coding cost in (1), its matching probability remains lower than that attainable by PML and IML.

### C.1.1  Proof.

We denote by $K_A, K_B$ the index selected by parties $A$ and $B$, respectively. We first note that the event $\{K_A = K_B = i, Y_i = y\}$ is equivalent to the event $\{K_A = K_B = i, Y_{K_A} = y\}$, thus:

$$\Pr(K_A = K_B = i | Y_{K_A} = y) = \frac{\Pr(K_A = K_B = i | Y_i = y) Q_Y(y)}{P_Y^A(y)}, \tag{96}$$

where the denominator is due to $Y_{K_A} \sim P_Y^A(.)$. Since:

$$\Pr(K_A = K_B | Y_{K_A} = y) = \sum_{i=1}^{\infty} \Pr(K_A = K_B = i | Y_{K_A} = y) \tag{97}$$

$$= \frac{Q_Y(y)}{P_Y^A(y)} \sum_{i=1}^{\infty} P(K_A = K_B = i | Y_i = y) \tag{98}$$

We will later show that:

$$\Pr(K_A = K_B = i | Y_i = y) = \frac{\min(P_Y^A(y), P_Y^B(y))}{\omega Q_Y(y)} \left[ 1 - \frac{1}{\omega} \int \max(P_Y^A(y), P_Y^B(y)) dy \right]^{i-1}, \tag{99}$$

which gives us:

$$\Pr(K_A = K_B | Y_{K_A} = y) \tag{100}$$

$$= \frac{Q_Y(y)}{P_Y^A(y)} \cdot \frac{\min(P_Y^A(y), P_Y^B(y))}{\omega Q_Y(y)} \sum_{i=1}^{\infty} \left[ 1 - \frac{1}{\omega} \int \max(P_Y^A(y), P_Y^B(y)) dy \right]^{i-1} \tag{101}$$

$$= \frac{\min(P_Y^A(y), P_Y^B(y))}{\omega P_Y^A(y)} \sum_{i=0}^{\infty} \left[ 1 - \frac{1}{\omega} \int \max(P_Y^A(y), P_Y^B(y)) dy \right]^{i} \tag{102}$$

$$= \frac{\min(P_Y^A(y), P_Y^B(y))}{\omega P_Y^A(y)} \frac{\omega}{\int \max(P_Y^A(y), P_Y^B(y)) dy} \tag{103}$$

$$= \frac{\min(1, P_Y^B(y)/P_Y^A(y))}{\int \max(P_Y^A(y), P_Y^B(y)) dy} \tag{104}$$

$$= \frac{\min(1, P_Y^B(y)/P_Y^A(y))}{1 + \mathrm{TV}(P_Y^A, P_Y^B)}, \tag{105}$$

where $TV(P_Y^A, P_Y^B)$ is the total variation distance between $P_Y^A(.)$ and $P_Y^B(.)$. Using the inequality $\min(u, v) \geq \frac{uv}{u+v}$ and the fact that $TV(P_Y^A, P_Y^B) \leq 1$ gives us the latter inequality.

---

[2]Daliri et al. [8] also arrives to a similar conclusion but for discrete case, targeting a different problem.

To show (99), we first compute the following probabilities where $A$ and $B$ both accept/terminate a given sample $Y = y$:

$$\gamma(y) = \Pr(A \text{ and } B \text{ accepts } Y | Y = y) \tag{106}$$

$$= \Pr(U \leq \min(P_Y^A(y), P_Y^B(y))) | Y = y) \tag{107}$$

$$= \frac{\min(P_Y^A(y), P_Y^B(y))}{\omega Q_Y(y)} \tag{108}$$

and,

$$\hat{\gamma}(y) = \Pr(A \text{ and } B \text{ rejects } Y | Y = y) \tag{109}$$

$$= \Pr(U > \max(P_Y^A(y), P_Y^B(y))) | Y = y) \tag{110}$$

$$= 1 - \frac{\max(P_Y^A(y), P_Y^B(y))}{\omega Q_Y(y)} \tag{111}$$

Then we have:

$$\Pr(K_A = K_B = i | Y_i = y_i) \tag{112}$$

$$= \int \Pr(K_A = K_B = i | Y_{1:i} = y_{1:i}) Q_Y(Y_{1:i-1} = y_{1:i-1} | Y_i = y) dy_{1:i-1} \tag{113}$$

$$= \int \Pr(K_A = K_B = i | Y_{1:i} = y_{1:i}) Q_Y(Y_{1:i-1} = y_{1:i-1}) dy_{1:i-1} \tag{114}$$

$$= \int \Pr(K_A = K_B = i | Y_{1:i} = y_{1:i}) Q_Y(Y_{1:i-1} = y_{1:i-1}) dy_{1:i-1} \tag{115}$$

$$= \gamma(y_i) \int \prod_{j=1}^{i-1} \hat{\gamma}(y_j) Q_Y(y_j) dy_{1:i-1} \tag{116}$$

$$= \gamma(y_i) \prod_{j=1}^{i-1} \int \hat{\gamma}(y) Q_Y(y) dy \tag{117}$$

$$= \frac{\min(P_Y^A(y), P_Y^B(y))}{\omega Q_Y(y)} \left[ \int \left( 1 - \frac{\max(P_Y^A(y), P_Y^B(y))}{\omega Q_Y(y)} \right) Q_Y(y) dy \right]^{i-1} \tag{118}$$

$$= \frac{\min(P_Y^A(y), P_Y^B(y))}{\omega Q_Y(y)} \left[ 1 - \frac{1}{\omega} \int \max(P_Y^A(y), P_Y^B(y) dy \right]^{i-1} \tag{119}$$

Finally, we note that:

$$\Pr(B \text{ outputs } y | A \text{ outputs } y) \tag{120}$$

$$= \Pr(K_B = K_A | Y_{K_{P_A}=y}) + \Pr(\text{party } B \text{ outputs } y, K_B \neq K_A | Y_{K_A=y}) \tag{121}$$

Finally, note that in the case where $P_A(.), P_B(.)$ are continuous distribution, we have:

$$\Pr(\text{party } B \text{ outputs } y, K_{P_B} \neq K_{P_A} | Y_{K_{P_A}=y}) = 0.0 \tag{122}$$

This completes the proof.

## C.2 Comparision with Poisson Matching Lemma

We will compare the average matching probability $\Pr(K_A = K_B)$ between RS and PML in the continuous case. Starting from equation (30) in [27] and assume $P_Y^A(y) \leq P_Y^B(y)$, we have:

$$P(Y_A = Y_B = y) \tag{123}$$

$$= \Pr(K_A = K_B | Y_A = y) P(Y_A = y) \tag{124}$$

$$= \frac{1}{\int_{-\infty}^{\infty} \max \left\{ \frac{P_Y^A(v)}{P_Y^A(y)}, \frac{P_Y^B(v)}{P_Y^B(y)} \right\} dv} \tag{125}$$

$$= \frac{P_Y^A(y)}{\int_{-\infty}^{\infty} \max \left\{ P_Y^A(v), \frac{P_Y^B(v)}{P_Y^B(y)} P_Y^A(y) \right\} dv} \tag{126}$$

$$\geq \frac{P_Y^A(y)}{\int_{-\infty}^{\infty} \max \left\{ P_Y^A(v), P_Y^B(v) \right\} dv} \quad \text{(Since we assume } P_Y^A(y) \leq P_Y^B(y)) \tag{127}$$

$$= \frac{P_Y^A(y)}{1 + \mathrm{TV}(P_Y^A, P_Y^B)} \tag{128}$$

Repeating the same step for $P_Y^A(y) \geq P_Y^B(y)$, we have:

$$P(Y_A = Y_B = y) \geq \frac{\min(P_Y^A(y), P_Y^B(y))}{1 + \mathrm{TV}(P_Y^A, P_Y^B)} \tag{129}$$

Taking the integral with respect to $y$ for both sides gives us the desired inequality where the RHS expression is the average matching probability of RS. Finally, the same conclusion holds for IML since the matching probability of IML converges to that of PML.

## D Greedy Rejection Sampling.

### D.1 Coding Cost

Compared to the standard RS approach described above, GRS is a more well-known tool for channel simulation [14, 18], as its runtime entropy, i.e., $H[K]$, is significantly lower than that of standard RS. Unlike standard RS, where the acceptance probability remains the same on average at each step, GRS greedily accepts samples from high-density regions as early as possible (see [14] for more details). Using these properties, Flamich and Theis [14] provide the following upper bound on $H[K]$, which generalizes the discrete version established by Harsha et al. [18]:

$$H[K] \leq I[X; Y] + \log(I[X; Y] + 1) + 4, \tag{130}$$

which has a smaller constant compared to the bound for standard RS. We conclude with a note on the coding cost of GRS, highlighting that, unlike standard RS, which is relatively easy to implement in practice, GRS can be more challenging to deploy as it requires repeatedly computing a complex and potentially intractable integral.

### D.2 Matching Probability in Greedy Rejection Sampling

**Setup.** Let the proposal distribution $Q_Y$ be a discrete uniform $\mathrm{Unif}[1, n]$, i.e. $Q_Y(y) = q = 1/n$ and $U \sim \mathcal{U}(0, 1)$ as in standard RS. Then, we define $W$ as follow:

$$W = \{(Y_1, U_1), (Y_2, U_2), ...\} \tag{131}$$

Our goal is to show that, for this proposal distribution $Q_Y$, there exists the target distributions $P_Y^A(.)$ and $P_Y^B(.)$ such that the GRS matching probability $\Pr(Y_A = Y_B | Y_A = y) \to 0.0$ even when $P_Y^A(y) = P_Y^B(y)$. Let $n = 2k + 1$, we construct the following $P_Y^A$ and $P_Y^B$:

$$P_Y^A(Y = 1) = \frac{k+1}{2k+1}, \quad P_Y^A(Y = i) = \begin{cases} \frac{1}{2k+1}, & \text{for } 1 < i \leq k+1 \\ 0.0, & \text{for } i > k+1 \end{cases}, \tag{132}$$

$$P_Y^B(Y = 1) = \frac{k+1}{2k+1}, \quad P_Y^B(Y = i) = \begin{cases} \frac{1}{2k+1}, & \text{for } i > k+1 \\ 0.0, & \text{for } 1 < i \leq k+1 \end{cases}, \tag{133}$$

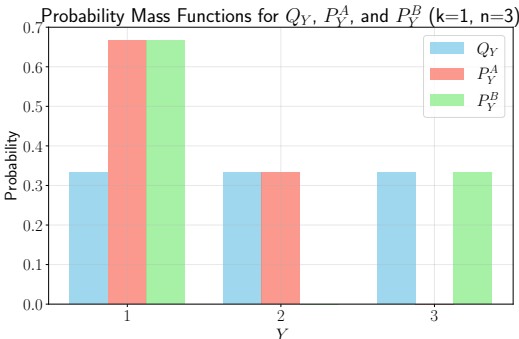

Figure 8: Visualization of example distributions in Section D.2 for $k = 1$.

where we visualize this in Figure 8.

**GRS Matching Probability.** Given party $A$ has target distribution $P_Y^A(.)$ and party $B$ has target distribution $P_Y^B(.)$, with each running the GRS procedure to obtain their samples $Y_A, Y_B$ respectively. We want to characterize the probability that party $A$ and party $B$ outputs the same value, give party $A$'s output. We denote $K_A$ and $K_B$ as the index within $W$ that party $A$ and party $B$ select respectively, i.e., $Y_{K_A} = Y_A$ and $Y_{K_B} = Y_B$.

Consider the event $Y_A = 1$, with the construction above, we have the following properties:

- If party $A$ and party $B$ both see the first proposal $Y_1 = 1$, they will greedily accept it, since $P_Y^A(Y = 1) = P_Y^B(Y = 1) \geq Q_Y(Y = 1)$. So in this case:

$$\Pr(K_A = K_B = 1, Y_A = 1) = Q_Y(Y = 1) = \frac{1}{2k+1}$$

- On the other hand, if the first proposal $Y_1 \neq 1$ then either party $A$ or $B$ must accept and output $Y_1 \neq 1$ since for $y \neq 1$, the probability distribution complement each other and equal to $Q_Y(y) = \frac{1}{2k+1}$. For example, for $n = 3$ and $Y_2 = 2$, then party $A$ will accept it while party $B$ must reject it. Therefore, we have:

$$\Pr(K_A = K_B > 1, Y_A = 1) = 0.0.$$

- Finally, from the previous analysis, for any positive integers $i \neq j$, we have

$$\Pr(K_A = i, K_B = j, Y_A = 1, Y_B = 1) = 0.0,$$

Indeed, consider $i = 1$ then $\Pr(K_A = 1, K_B = j, Y_A = 1, Y_B = 1) = 0.0$ since both of them must accept the first proposal $Y_1 = 1$. On the other hand, if $i > 1$ then we must have $j = 1$ since we know that $Y_1 \neq 1$ in this case and thus one of the party must stop. Since $i > 1$, it has to be party $B$ and in this case, $Y_B \neq 1$.

For this reason, we have:

$$\Pr(Y_A = Y_B = 1) \tag{134}$$
$$= \Pr(K_A = K_B, Y_A = 1, Y_B = 1) + \Pr(K_A \neq K_B, Y_A = 1, Y_B = 1) \tag{135}$$
$$= \Pr(K_A = K_B, Y_A = 1) + \sum_{i \neq j} \Pr(K_A = i, K_B = j, Y_A = 1, Y_B = 1) \tag{136}$$
$$= \Pr(K_A = K_B, Y_A = 1) \tag{137}$$
$$= \Pr(K_A = K_B = 1, Y_A = 1) + \Pr(K_A = K_B > 1, Y_A = 1) \tag{138}$$
$$= Q_Y(Y = 1) \tag{139}$$
$$= \frac{1}{2k+1} \tag{140}$$

and hence:

$$\Pr(Y_A = Y_B | Y_A = 1) = \frac{1}{k+1} \tag{141}$$

which approaches $0.0$ as $n \to \infty$. Overall, due to its greedy selection approach, GRS may yield lower matching probabilities compared to other methods such as PML which we provide the analysis below.

**Matching Probability of PML.** In PML, the matching probability is $\Pr(Y_A = Y_B \mid Y_A = 1) = 1$. This results from PML's more global selection process compared to GRS, as it evaluates all candidates comprehensively. In particular, let $W = (S_1, Y_1), ..., (S_n, Y_n)$ where $S_i \sim \text{Exp}(1)$ and let $K_A, K_B$ be the value within $W$ that each party respectively select in this case. Note that the construction of $W$ in the discrete case for PML does not require $Q_Y$. The selection process according to PML is as follows:

$$K_A = \arg \min_{1 \leq i \leq n} \frac{S_i}{P_Y^A(Y_i)} \quad K_B = \arg \min_{1 \leq i \leq n} \frac{S_i}{P_Y^B(Y_i)}, \tag{142}$$

and each party outputs $Y_A = Y_{K_A}, Y_B = Y_{K_B}$. We see that if $K_A = 1$, then we must have $K_B = 1$. This is because for any $i > 1$, we have $P_Y^A(Y = 1) = P_Y^B(Y = 1) > P_Y^B(Y = i)$ and $P_Y^A(Y = i) = P_Y^B(Y = i + 1 + k)$. Thus, this gives $\Pr(Y_A = Y_B \mid Y_A = 1) = 1$.

---

**Algorithm 1:** Ensemble Rejection Sampling - $\text{ERS}(W; P_Y, Q_Y, \omega = \max_y \frac{P_Y(y)}{Q_Y(y)}, \text{scale} = 1)$

---

**Input:** Target distribution $P_Y$, Proposal distribution $Q_Y$, and the source of randomness $W$ (see
Section 5.1). Default value $\omega = \max_y \frac{P_Y(y)}{Q_Y(y)}$ unless override by some value $> \omega$.
Default scaling factor $\text{scale} = 1$ unless override by some value within $(0, 1]$.
**Output:** Selected Index $K$ and sample $Y_K \sim P_Y$
1. Observe batch $\{B_i, U_i\}$
2. Select candidate index $K_i^{\text{cand}}$:

$$K_i^{\text{cand}} = \operatorname*{argmin}_{1 \leq k \leq N} \frac{S_{ik}}{\lambda_{ik}}, \quad \text{where:} \quad \lambda_{ik} = \frac{P_Y(Y_{ik})}{Q_Y(Y_{ik})}$$

3. Compute:

$$\hat{Z}(Y_{i,1:N}) = \sum_{k=1}^{N} \lambda_{ik}, \quad \bar{Z}(Y_{i,1:N}, K_i^{\text{cand}}) = \hat{Z}(Y_{i,1:N}) + \omega - \lambda_{i,K_i^{\text{cand}}}$$

4. Set $K_1 = i$, $K_2 = K_i^{\text{cand}}$, $K = (N-1)1i + K_i^{\text{cand}}$ and return $Y_K$ if:

$$U_i \leq \frac{\hat{Z}(Y_{i,1:N})}{\bar{Z}(Y_{i,1:N}, K_i^{\text{cand}})} \cdot \text{scale},$$

else repeat Step 1 with $B_{i+1}$.

---

# E  ERS Coding Scheme

## E.1  Prelimaries

We show the standard ERS algorithm in Algorithm 1, following the original version introduced by
Deligiannidis et al. [9] with a slight generalization in terms of the scaling factor ($0 < \text{scale} \leq 1$)
that we will use for channel simulation purpose. This section begins by establishing some detailed
quantities that will be used repeatedly. For simplicity, we use $P_x(.)$ for the target distribution $P_{Y|X=x}$
and $Q(.)$ for the proposal distribution. Let $\omega_x = \max_y P_x(y)/Q(y)$, we define the quantities:

$$\hat{Z}_x(y_{1:N}) = \sum_{j=1}^{N} \frac{P_x(y_j)}{Q(y_j)}, \quad \bar{Z}_x(y_{1:N}, k) = \hat{Z}_x(y_{1:N}) - \frac{P_x(y_k)}{Q(y_k)} + \omega_x \tag{143}$$

and denote the following constants:

$$\Delta_x = \mathbb{E}_{Y_{1:N} \sim Q}\left[\frac{N}{\bar{Z}_x(Y_{1:N}, 1)}\right], \quad \Delta = \frac{N}{N - 1 + \omega}, \tag{144}$$

where we recall that $\omega = \max_{x,y} P_x(y)/Q(y)$, by Jensen's inequality we have the following:

$$\Delta_x \geq \frac{N}{N - 1 + \omega_x} \geq \frac{N}{N - 1 + \omega} = \Delta \text{ for every } x. \tag{145}$$

From this, we can see that as $N \to \infty$, we achieve $\Delta \to 1.0$. This value $\Delta$ turns out to be the average
batch acceptance probability when we set $\text{scale} = \frac{\Delta}{\Delta_x}$, which we elaborate on below.

**Scaled Acceptance Probability.** For the channel simulation setting in this section, we slightly modify
the acceptance probability in Algorithm 1 (step 4) with a scaling factor $\text{scale} = \frac{\Delta}{\Delta_x} \leq 1$ such that
the average batch acceptance probability is the same, regardless of the target distribution $P_x$ [3]. In
particular, the encoder selects the index according to:

$$K = \text{ERS}(W; P_x, Q, \text{scale} = \frac{\Delta}{\Delta_x}), \tag{146}$$

---

[3]This is similar to the case of standard RS where we accept/reject based on the global ratio bound $\omega$ instead
of $\omega_x$.

which means for a batch $i$ containing samples $Y_{i,1:N} = y_{1:N}$, we accept it within step 4 if:

$$\text{Accept if } U_i \leq \frac{\hat{Z}_x(y_{1:N})}{\bar{Z}_x(y_{1:N}, k)} \frac{\Delta}{\Delta_x}, \tag{147}$$

where we modify the scaling scale $= \frac{\Delta}{\Delta_x} \leq 1$ in Algorithm 1, which is a constant and does not affect the resulting output distribution. The value of $k$ is determined via the Gumbel-Max selection procedure in Step 2. The intuition is, within every accepted batch without scaling, we randomly reject $(1 - \text{scale})$ of them. Formally, first consider the following **ERS proposal distribution**:

$$\bar{Q}_{Y_{1:N}, K}(y_{1:N}, k; x) = \left( \frac{P_x(y_k)/Q(y_k)}{\sum_{j=1}^{N} P_x(y_j)/Q(y_j)} \right) \prod_{j=1}^{N} Q(y_j) \tag{148}$$

$$= \left( \frac{P_x(y_k)/Q(y_k)}{\hat{Z}_x(y_{1:N})} \right) \prod_{j=1}^{N} Q(y_j), \tag{149}$$

where the first product in the RHS is the likelihood we obtain the samples $y_{1:N}$ from the original proposal distribution $Q_Y(.)$ and the ratio is due to the IS process. Now, the **ERS target distribution** is $\bar{P}_{Y_{1:N}, K}(y_{1:N}, k; x)$ where

$$\bar{P}_{Y_{1:N}, K}(y_{1:N}, k; x) = \frac{1}{\alpha} \left( \frac{P_x(y_k)/Q(y_k)}{\hat{Z}_x(y_{1:N})} \frac{\hat{Z}_x(y_{1:N})}{\bar{Z}_x(y_{1:N}, k)} \frac{\Delta}{\Delta_x} \right) \prod_{j=1}^{N} Q(y_j) \tag{150}$$

$$= \left( \frac{P_x(y_k)/Q(y_k)}{\Delta_x \bar{Z}_x(y_{1:N}, k)} \right) \prod_{j=1}^{N} Q(y_j), \tag{151}$$

which is the batch target distribution that yields $Y \sim P_x$ when no scaling occur (see [9], Section 2.2), since the normalization factor $\alpha$ is:

$$\alpha = \sum_{k=1}^{N} \int_{-\infty}^{\infty} \left( \frac{P_x(y_k)/Q(y_k)}{\bar{Z}_x(y_{1:N}, k)} \right) \frac{\Delta}{\Delta_x} \left( \prod_{j=1}^{N} Q(y_j) \right) dy_{1:N} \tag{152}$$

$$= N \frac{\Delta}{\Delta_x} \int_{-\infty}^{\infty} \left( \frac{P_x(y_k)/Q(y_k)}{\bar{Z}_x(y_{1:N}, 1)} \right) \left( \prod_{j=1}^{N} Q(y_j) \right) dy_{1:N} \quad \text{(Due to symmetry)} \tag{153}$$

$$= N \frac{\Delta}{\Delta_x} \int_{-\infty}^{\infty} \frac{1}{\bar{Z}_x(y_{1:N}, 1)} \left( \prod_{j=2}^{N} Q(y_j) \right) dy_2^N \tag{154}$$

$$= \Delta \tag{155}$$

It turns out that $\Delta$ is also the **batch acceptance probability** since:

$$\Pr(\text{Accept batch } B) = \mathbb{E}_{(Y_{1:N}, K) \sim \bar{Q}} \left[ \frac{\Delta}{\Delta_x} \frac{\hat{Z}_x(y_{1:N})}{\bar{Z}_x(y_{1:N}, k)} \right] \tag{156}$$

$$= \frac{\Delta}{\Delta_x} \sum_{k=1}^{N} \int_{-\infty}^{\infty} \left( \frac{P_x(y_k)/Q(y_k)}{\bar{Z}_x(y_{1:N}, k)} \right) \tag{157}$$

$$= \frac{\Delta}{\Delta_x} N \int_{-\infty}^{\infty} \left( \frac{P_x(y_1)/Q(y_1)}{\bar{Z}_x(y_{1:N}, 1)} \right) \left( \prod_{j=1}^{N} Q(y_j) \right) dy_{1:N} \tag{158}$$

$$= \Delta, \tag{159}$$

and it can be observed that, without the scaling factor $\frac{\Delta}{\Delta_x}$, the batch acceptance probability is $\Delta_x$. Finally, we can view the ERS as a standard RS procedure with proposal distribution $\bar{Q}_{Y_1^N, K}$ and target distribution $\bar{P}_{Y_1^N, K}$.

**Harris-FKG/Chebyshev Inequality.** We introduce the following inequality (Harris-FKG/Chebyshev), which will be used in the proof:

**Proposition E.1.** *For function $f, g$ on $Y \sim P(.)$ where $f$ is non-increasing and $g$ is non-decreasing, we have:*

$$\mathbb{E}[f(Y)g(Y)] \leq \mathbb{E}[f(Y)]\mathbb{E}[g(Y)]$$

*Proof.* Let $Y_1, Y_2 \sim P(.)$ and they are independent. Then we have:

$$[f(Y_1) - f(Y_2)][g(Y_1) - g(Y_2)] \leq 0 \tag{160}$$

Hence:

$$\mathbb{E}\{[f(Y_1) - f(Y_2)][g(Y_1) - g(Y_2)]\} \leq 0 \tag{161}$$

This gives us:

$$\mathbb{E}[f(Y_1)g(Y_1)] + \mathbb{E}[f(Y_2)g(Y_2)] \leq \mathbb{E}[f(Y_1)]\mathbb{E}[g(Y_2)] + \mathbb{E}[f(Y_2)]\mathbb{E}[g(Y_1)], \tag{162}$$

which completes the proof. $\square$

## E.2 Encoding $K_1$.

We encode $K_1$ the same way as the scheme for standard RS. Similar to standard RS, we encode $K_1$ into two messages. Specifically:

- Step 1: the encoder sends the ceiling $L = \lceil \frac{K_1}{\lfloor \Delta^{-1} \rfloor} \rceil$ to the decoder. The decoder then knows $(L-1)\lfloor \Delta^{-1} \rfloor^{-1} + 1 \leq L \leq L\lfloor \Delta^{-1} \rfloor^{-1}$, i.e. $K_1$ is in chunk $L$ that consists of $\lfloor \Delta^{-1} \rfloor^{-1}$ batches. We have $\mathbb{E}[\log(L)] \leq 1$ bit.

- Step 2: The encoder and decoder both sort the uniform random variables $U_i$ within the selected chunk $(L-1)\lfloor \Delta^{-1} \rfloor^{-1} + 1 \leq i \leq L\lfloor \Delta^{-1} \rfloor^{-1}$. Let the sorted list be $U_{\pi(1)} \leq U_{\pi(2)} \leq ... \leq U_{\pi(\lfloor \Delta^{-1} \rfloor)}$ where $\pi(.)$ is the mapping between the sorted index and the original unsorted one. The encoder sends the rank of $U_{K_1}$ within this list, i.e. sends the value $T$ such that $K_1 = \pi(\hat{K}_1)$. The decoder receive $\hat{K}_1$ and retrieve $B_{K_1}$ accordingly. Section E.2.2 shows the coding cost for this step.

We provide the detail analysis in Section E.2.1 and E.2.2. Notice that the role $\Delta$ plays here is similar to that of $\omega$ in standard RS.

### E.2.1 Coding Cost of $L$

Similar to RS, since each batch is accepted with probability $\Delta$ (see (159)), this means:

$$\Pr(K_1 > \ell\Delta^{-1}) = (1 - \Delta)^{\ell\lfloor \Delta^{-1} \rfloor} < 0.5^{-\ell},$$

which is equivalent to $\Pr(L > \ell) < 0.5^{-\ell}$. Note that we reuse the inequality in Appendix B.3. We have:

$$\mathbb{E}[L] = \sum_{\ell=0}^{\infty} \Pr(L > \ell) < 1 + 0.5^{-1} + 0.5^{-2} + ... = 2, \tag{163}$$

implying $\mathbb{E}[\log L] \leq 1$.

### E.2.2 Coding Cost of $\hat{K}_1$

We will show that:

$$\mathbb{E}[\log \hat{K}_1] \leq \frac{N}{\Delta_x} \mathbb{E}_{Y_{1:N} \sim Q} \left[ \frac{P_x(Y_1)/Q(Y_1)}{\bar{Z}_x(Y_{1:N}, 1)} \log \left( \frac{\hat{Z}_x(Y_{1:N})}{\Delta_x \bar{Z}_x(Y_{1:N}, 1)} \right) \right], \tag{164}$$

where we provide the result of (164) in Section E.4.2.

### E.3 Encoding $K_2$.

Given an accepted batch $\{(Y_i, S_i)\}_{i=1}^N$ , we have:

$$K_2 = \arg \min_{1 \le i \le N} \frac{S_i}{\lambda_i}; \qquad \Theta_P = \min_{1 \le i \le N} \frac{S_i}{\lambda_i}, \tag{165}$$

where we have the weights $\lambda_i$ defined as:

$$\lambda_i = \frac{P(Y_i)}{Q(Y_i)} \tag{166}$$

After communicating the selected batch index $K_1$, the encoder and decoder sort the exponential random variables $\{S_{K_1,i}\}_{i=1}^N$, i.e.

$$S_{K_2,\pi(1)} \le S_{K_2,\pi(2)} \le \dots \le S_{K_2,\pi(N)}, \tag{167}$$

and send the sorted index $\hat{K}_2$ of $K_2$, i.e. $\pi(\hat{K}_2) = K_2$. The decoder also performs the sorting operation and retrieve $K_2$ accordingly. Since $K_2$ are obtained from the batch selected by ERS, we analyze $\mathbb{E}[\log \hat{K}_2' | Y_{1:N}$ are selected$]$, where $\hat{K}_2'$ and $K_2'$ are defined the same as $\hat{K}_2$ and $K_2$ (follows the same Gumbel-Max procedure) but for arbitrary $N$ i.i.d. proposals $Y_{1:N} \sim Q(.)$. In this case:

$$\mathbb{E}[\log \hat{K}_2] = \mathbb{E}[\log \hat{K}_2' | Y_{1:N} \text{ are selected}]$$

Notice the following identity:

$$\bar{P}(y_{1:N}, k_2; x) = P(Y_{1:N} = y_{1:N} | Y_{1:N} \text{ are selected}, K_2' = k_2) \Pr(K_2' = k_2 | Y_{1:N} \text{ are selected})$$

where $\bar{P}(y_{1:N}, k_2; x)$ is the ERS target distribution described previously in Appendix E.1. Then, we obtain the following likelihood:

$$P(Y_{1:N} = y_{1:N} | Y_{1:N} \text{ are selected}, K_2' = 1) \tag{168}$$

$$= \frac{\bar{P}(y_{1:N}, j; x)}{\Pr(K_2' = 1 | Y_{1:N} \text{ are selected})} \tag{169}$$

$$= N \frac{P_x(y_1)/Q(y_1)}{\Delta_x \bar{Z}_x(y_{1:N}, 1)} \prod_{i=1}^N Q(y_i) \tag{170}$$

With this, we now bound the expectation term of interest $\mathbb{E}[\log \hat{K}_2]$ as follows:

$$\mathbb{E}[\log \hat{K}_2] \tag{171}$$

$$= \mathbb{E}[\log \hat{K}_2' | Y_{1:N} \text{ are selected}] \tag{172}$$

$$= \mathbb{E}[\log \hat{K}_2' | Y_{1:N} \text{ are selected}, K_2' = 1] \quad (\text{ Due to Symmetry}) \tag{173}$$

$$= \mathbb{E}_{Y_{1:N}}[\mathbb{E}[\log \hat{K}_2' | Y_{1:N} \text{ are selected}, K_2' = 1, Y_{1:N} = y_{1:N}]] \tag{174}$$

$$= N \int_{-\infty}^{\infty} \frac{P_x(y_1)/Q(y_1)}{\Delta_x \bar{Z}_x(y_{1:N}, 1)} \mathbb{E}[\log \hat{K}_2' | Y_{1:N} \text{ are selected}, Y_{1:N} = y_{1:N}, K_2' = 1] \left( \prod_{i=1}^N Q(y_i) \right) dy_{1:N} \tag{175}$$

$$= N \int_{-\infty}^{\infty} \frac{P_x(y_1)/Q(y_1)}{\Delta_x \bar{Z}_x(y_{1:N}, 1)} \mathbb{E}[\log \hat{K}_2' | Y_{1:N} = y_{1:N}, K_2' = 1] \left( \prod_{i=1}^N Q(y_i) \right) dy_{1:N}, \tag{176}$$

where the last equality is because, given $\{Y_{1:N} = y_{1:N}, K_2' = 1\}$, the event $\{Y_{1:N}$ are selected$\}$ and the random variable $\hat{K}_2'$ are independent. In particular, the decision whether to accept a batch or not does not depends on the rank of $S_{K_2'}$, that is:

$$\Pr(Y_{1:N} \text{ are selected} | Y_{1:N} = y_{1:N}, K_2' = 1, \hat{K}_2' = k_2) \tag{177}$$

$$= \Pr(Y_{1:N} \text{ are selected} | Y_{1:N} = y_{1:N}, K_2' = 1) \tag{178}$$

$$= \frac{\hat{Z}_x(y_{1:N})}{\bar{Z}_x(y_{1:N}, 1)} \frac{\Delta}{\Delta_x} \tag{179}$$

We then have:

$$\mathbb{E}[\log \hat{K}_2' | Y_{1:N} \text{ are selected}] \tag{180}$$

$$= N \int_{-\infty}^{\infty} \prod_{i=1}^{N} Q(y_i) \frac{P_x(y_1)/Q(y_1)}{\Delta_x \bar{Z}_x(y_{1:N}, 1)} \left( \int_0^{\infty} e^{-\theta} \mathbb{E}[\log \hat{K}_2' | Y_{1:N}{=}y_{1:N}, K_2'{=}1, \Theta_P{=}\theta] d\theta \right) dy_{1:N}, \tag{181}$$

since, given $Y_{1:N}$, $\Theta_P$ is independent of $K_2'$ and $\Theta_P{\sim}\text{Exp}(1)$ (see [35], Appendix 18). We now provide an upperbound of $\mathbb{E}[\log \hat{K}_2 | Y_{1:N}{=}y_{1:N}, K_2{=}1, \Theta_P{=}\theta]$, which follows the argument presented in [35], and is repeated here. Applying Jensen's inequality, we have:

$$\mathbb{E}[\log \hat{K}_2' | Y_{1:N} = y_{1:N}, K_2' = 1, \Theta_P = \theta] \leq \log \mathbb{E}[\hat{K}_2' | Y_{1:N} = y_{1:N}, K_2' = 1, \Theta_P = \theta], \tag{182}$$

We then rewrite $\hat{K}_2'$ as the following:

$$\hat{K}_2' = |\{S_i < S_{K_2'}\}| + 1, \tag{183}$$

which gives us:

$$\mathbb{E}[\hat{K}_2' | Y_{1:N} = y_{1:N}, K_2' = 1, \Theta_P = \theta] \tag{184}$$

$$= 1 + \mathbb{E}[|\{S_i < S_{K_2'}\}| | Y_{1:N} = y_{1:N}, K_2' = 1, \Theta_P = \theta] \tag{185}$$

$$= 1 + \mathbb{E}\left[ \left| \left\{ S_i < \theta \frac{P_x(Y_{K_2'})/Q(Y_{K_2'})}{\hat{Z}_x(Y_{1:N})} \right\} \right| \middle| Y_{1:N} = y_{1:N}, K_2' = 1, \Theta_P = \theta \right] \tag{186}$$

$$= 1 + \sum_{i=2}^{N} \Pr\left( S_i < \theta \frac{P_x(Y_{K_2'})/Q(Y_{K_2'})}{\hat{Z}_x(Y_{1:N})} \middle| Y_{1:N} = y_{1:N}, K_2' = 1, \Theta_P = \theta \right) \tag{187}$$

$$= 1 + \sum_{i=2}^{N} \Pr\left( S_i < \theta \frac{P_x(Y_1)/Q(Y_1)}{\hat{Z}_x(Y_{1:N})} \middle| Y_{1:N} = y_{1:N}, \frac{S_j}{\frac{P_x(y_j)/Q(y_j)}{\hat{Z}_x(y_{1:N})}} {\geq} \theta \text{ for } j{\neq}1, \frac{S_1}{\frac{P_x(y_1)/Q(y_1)}{\hat{Z}_x(y_{1:N})}}{=}\theta \right) \tag{188}$$

$$= 1 + \sum_{i=2}^{N} \Pr\left( S_i < \theta \frac{P_x(Y_1)/Q(Y_1)}{\hat{Z}_x(Y_{1:N})} \middle| Y_{1:N}{=}y_{1:N}, \frac{S_j}{\frac{P_x(y_j)/Q(y_j)}{\hat{Z}_x(y_{1:N})}}{\geq}\theta \text{ for } j{\neq}1, \frac{S_1}{\frac{P_x(y_1)/Q(y_1)}{\hat{Z}_x(y_{1:N})}}{=}\theta \right) \tag{189}$$

$$= 1 + \sum_{i=2}^{N} \Pr\left( S_i < \theta \frac{P_x(Y_1)/Q(Y_1)}{\hat{Z}_x(Y_{1:N})} \middle| Y_{1:N} = y_{1:N}, \frac{S_i}{\frac{P_x(y_i)/Q(y_i)}{\hat{Z}_x(y_{1:N})}} \geq \theta \right) \tag{190}$$

Note that:

$$\Pr\left( S_i < \theta \frac{P_x(Y_1)/Q(Y_1)}{\hat{Z}_x(y_{1:N})} \middle| Y_{1:N} = y_{1:N}, \frac{S_i}{\frac{P_x(y_i)/Q(y_i)}{\hat{Z}_x(y_{1:N})}} \geq \theta \right) \tag{191}$$

$$= \mathbf{1}\left\{ \theta \frac{P_x(Y_1)/Q(Y_1)}{\hat{Z}_x(y_{1:N})} \geq \theta \frac{P_x(Y_i)/Q(Y_i)}{\hat{Z}_x(y_{1:N})} \right\} \left[ 1 - \exp\left( -\theta \frac{P_x(y_1)/Q(y_1) - P_x(y_i)/Q(y_i)}{\hat{Z}_x(y_{1:N})} \right) \right] \tag{192}$$

$$\leq 1 - \exp\left( -\theta \frac{P_x(y_1)/Q(y_1) - P_x(y_i)/Q(y_i)}{\hat{Z}_x(y_{1:N})} \right) \tag{193}$$

$$\leq \frac{\theta[P_x(y_1)/Q(y_1) - P_x(y_i)/Q(y_i)]}{\hat{Z}_x(y_{1:N})} \tag{194}$$

$$\leq \frac{\theta P_x(y_1)/Q(y_1)}{\hat{Z}_x(y_{1:N})} \tag{195}$$

As such:

$$\mathbb{E}[\hat{K}_2'|Y_{1:N} = y_{1:N}, K_2' = 1, \Theta_P = \theta] \leq 1 + \sum_{i=2}^{N} \frac{\theta P_x(y_1)/Q(y_1)}{\hat{Z}_x(y_{1:N})} \tag{196}$$

$$\leq 1 + \frac{N\theta P_x(y_1)/Q(y_1)}{\hat{Z}_x(y_{1:N})} \tag{197}$$

and thus:

$$\int_0^\infty e^{-\theta} \mathbb{E}[\log K|Y_{1:N} = y_{1:N}, K_2 = 1, \Theta_P = \theta]d\theta \tag{198}$$

$$\leq \int_0^\infty e^{-\theta} \log\left(1 + \frac{N\theta P_x(y_1)/Q(y_1)}{\hat{Z}_x(y_{1:N})}\right) d\theta \tag{199}$$

$$\leq \log\left(\frac{N P_x(y_1)/Q(y_1)}{\hat{Z}_x(y_{1:N})} + 1\right), \tag{200}$$

which is due to Jensen's inequality for concave function $\log(.)$. Finally, we have:

$$\mathbb{E}[\log \hat{K}_2] \tag{201}$$

$$\leq \mathbb{E}_{Y_{1:N} \sim Q(.)}\left[\frac{N P_x(Y_1)/Q(Y_1)}{\Delta_x \bar{Z}_x(Y_{1:N}, 1)} \log\left(\frac{N P_x(Y_1)/Q(Y_1)}{\hat{Z}_x(Y_{1:N})} + 1\right)\right] \tag{202}$$

$$\leq \mathbb{E}_{Y_{1:N} \sim Q(.)}\left[\frac{N P_x(Y_1)/Q(Y_1)}{\Delta_x \bar{Z}_x(Y_{1:N}, 1)} \log\left(\frac{N P_x(Y_1)/Q(Y_1)}{\hat{Z}_x(Y_{1:N})}\right) + \frac{\log(e)\hat{Z}_x(Y_{1:N})}{N P_x(Y_1)/Q(Y_1)} \frac{N P_x(Y_1)/Q(Y_1)}{\Delta_x \bar{Z}_x(Y_{1:N}, 1)}\right] \tag{203}$$

$$= \mathbb{E}_{Y_{1:N} \sim Q(.)}\left[\frac{N P_x(Y_1)/Q(Y_1)}{\Delta_x \bar{Z}_x(Y_{1:N}, 1)} \log\left(\frac{N P_x(Y_1)/Q(Y_1)}{\hat{Z}_x(Y_{1:N})}\right)\right] + \log(e) \tag{204}$$

The last inequality is due to the FKG inequality:

$$\mathbb{E}_{Y_{1:N} \sim Q(.)}\left[\frac{\log(e)\hat{Z}_x(Y_{1:N})}{\Delta_x \bar{Z}_x(Y_{1:N}, 1)}\right] \tag{205}$$

$$= \log(e)\mathbb{E}_{Y_2^N \sim Q(.)}\left[\left(1 + \sum_{i=2}^{N} \frac{P_x(Y_i)}{Q(Y_i)}\right)\left(\frac{1}{\Delta_x \bar{Z}_x(Y_{1:N}, 1)}\right)\right] \tag{206}$$

$$\leq \log(e)\mathbb{E}_{Y_2^N \sim Q(.)}\left[1 + \sum_{i=2}^{N} \frac{P_x(Y_i)}{Q(Y_i)}\right]\mathbb{E}_{Y_2^N \sim Q(.)}\left[\frac{1}{\Delta_x \bar{Z}_x(Y_{1:N}, 1)}\right] \tag{207}$$

$$= \log(e) \tag{208}$$

So we have the bound on $\mathbb{E}[\log(\hat{K}_2)]$ as:

$$\mathbb{E}[\log(\hat{K}_2)] \leq \mathbb{E}_{Y_{1:N} \sim Q(.)}\left[\frac{N P_x(Y_1)/Q(Y_1)}{\Delta_x \bar{Z}_x(Y_{1:N}, 1)} \log\left(\frac{N P_x(Y_1)/Q(Y_1)}{\hat{Z}_x(Y_{1:N})}\right)\right] + \log(e) \tag{209}$$

### E.4  Total Coding Cost of $K$

We now provide an upperbound on the total coding cost of $K$. We have:

$$H(K|W) = H(L, \hat{K}_1, \hat{K}_2|W) \tag{210}$$

$$\leq H(L|W) + H(\hat{K}_1|W) + H(\hat{K}_2|W) \tag{211}$$

$$\leq H(L) + H(\hat{K}_1) + H(\hat{K}_2) \tag{212}$$

For each of the message, we encode using Zipf distribution. Since $\mathbb{E}[\log(L)] \leq 1$, then:
$$H(L) \leq 3$$

For $H(\hat{K}_1)$, we have:
$$H(\hat{K}_1) \leq \mathbb{E}_X[\mathbb{E}[\log(\hat{K}_1)]] + \log(\mathbb{E}_X[\mathbb{E}[\log(\hat{K}_1)]] + 1) + 1 \tag{213}$$

and $H(\hat{K}_2)$, we have:
$$H(\hat{K}_2) \leq \mathbb{E}_X[\mathbb{E}[\log(\hat{K}_2)]] + \log(\mathbb{E}_X[\mathbb{E}[\log(\hat{K}_2)]] + 1) + 1 \tag{214}$$

and thus we have:
$$H(K|W) \tag{215}$$
$$\leq (\mathbb{E}_X[\mathbb{E}[\log(\hat{K}_1)] + \mathbb{E}[\log(\hat{K}_2)]]) + \log((\mathbb{E}_X[\mathbb{E}[\log(\hat{K}_1)]] + 1)(\mathbb{E}_X[\mathbb{E}[\log(\hat{K}_2)]] + 1)) + 5 \tag{216}$$

By AM-GM inequality, we have:
$$\log((\mathbb{E}_X[\mathbb{E}[\log(\hat{K}_1)]] + 1)(\mathbb{E}_X[\mathbb{E}[\log(\hat{K}_2)]] + 1)) \tag{217}$$
$$\leq \log(\frac{1}{4}(\mathbb{E}_X[\mathbb{E}[\log(\hat{K}_1)]] + 1 + \mathbb{E}_X[\mathbb{E}[\log(\hat{K}_2)]] + 1)^2) \tag{218}$$
$$= 2\log(\mathbb{E}_X[\mathbb{E}[\log(\hat{K}_1)]] + \mathbb{E}_X[\mathbb{E}[\log(\hat{K}_2)]] + 2) - 2 \tag{219}$$

We will show $\mathbb{E}[\log(\hat{K}_1)] + \mathbb{E}[\log(\hat{K}_2)] \leq D_{KL}(P_x||Q) + 3 + 2\log(e)$ at the end of this section. Given this, we have:
$$H(K|W) \leq I(X;Y) + 3 + 2\log(e) + 2\log(I(X;Y) + 5 + 2\log(e)) - 2 + 5 \tag{220}$$
$$\leq I(X;Y) + 2\log(I(X;Y) + 8) + 9. \tag{221}$$

Since we are encoding 3 messages separately, we add 1 bit overhead for each message and thus arrive to the constant 12 as in the original result.

The rest is to bound $\mathbb{E}[\log(\hat{K}_1)] + \mathbb{E}[\log(\hat{K}_2)]$, note that:

$$\mathbb{E}[\log(\hat{K}_1)] + \mathbb{E}[\log(\hat{K}_2)] \tag{222}$$
$$\leq 2\log(e) + \mathbb{E}_{Y_{1:N} \sim Q(.)}\left[\frac{NP_x(Y_1)/Q(Y_1)}{\Delta_x \bar{Z}_x(Y_{1:N}, 1)}\left(\log\left(\frac{\hat{Z}_x(Y_{1:N})}{\Delta_x \bar{Z}_x(Y_{1:N}, 1)}\right) + \log\left(\frac{NP_x(Y_1)/Q(Y_1)}{\hat{Z}_x(Y_{1:N})}\right)\right)\right] \tag{223}$$
$$= 2\log(e) + \mathbb{E}_{Y_{1:N} \sim Q(.)}\left[\frac{NP_x(Y_1)/Q(Y_1)}{\Delta_x \bar{Z}_x(Y_{1:N}, 1)}\log\left(\frac{NP_x(Y_1)/Q(Y_1)}{\Delta_x \bar{Z}_x(Y_{1:N}, 1)}\right)\right] \tag{224}$$
$$= 2\log(e) + \mathbb{E}_{Y_{1:N} \sim Q(.)}\left[\frac{NP_x(Y_1)/Q(Y_1)}{\Delta_x \bar{Z}_x(Y_{1:N}, 1)}\left(\log\frac{P_x(Y_1)}{Q(Y_1)} + \log\left(\frac{N}{\Delta_x \bar{Z}_x(Y_{1:N}, 1)}\right)\right)\right] \tag{225}$$
$$= 2\log(e) + D_{KL}(P_x||Q) + \mathbb{E}_{Y_{1:N} \sim Q(.)}\left[\frac{N}{\Delta_x \bar{Z}_x(Y_{1:N}, 1)}\log\left(\frac{N}{\Delta_x \bar{Z}_x(Y_{1:N}, 1)}\right)\right] \tag{226}$$
$$= 2\log(e) + D_{KL}(P_x||Q) + E_1 \tag{227}$$

where:
$$E_1 = \mathbb{E}_{Y_{1:N} \sim Q(.)}\left[\frac{N}{\Delta_x \bar{Z}_x(Y_{1:N}, 1)}\log\left(\frac{N}{\Delta_x \bar{Z}_x(Y_{1:N}, 1)}\right)\right] \tag{228}$$

We will show in Appendix E.4.1 that:
$$E_1 \leq 3 \tag{229}$$

and thus:
$$\mathbb{E}[\log(\hat{K}_1)] + \mathbb{E}[\log(\hat{K}_2)] \leq 2\log(e) + 3 + D_{KL}(P_x||Q) \tag{230}$$

### E.4.1 Bound on $E_1$

We consider two cases, when the batch size $N \leq 7\omega_x$ and when $N > 7\omega_x$.

*Case 1*: $N \leq 7\omega_x$

Recall that $\bar{Z}_x(Y_{1:N}, 1) > \omega_x$ and $\Delta_x \geq \frac{N}{N-1+\omega_x}$, we have:

$$\frac{N}{\Delta_x \bar{Z}_x(Y_{1:N}, 1)} \leq \frac{N-1+\omega_x}{\omega_x} \tag{231}$$

$$< \frac{8\omega_x - 1}{\omega_x} \quad (\text{ Since } N \leq 7\omega) \tag{232}$$

$$< 8 \tag{233}$$

Thus, we have:

$$E_1 = \mathbb{E}_{Y_{1:N} \sim Q(.)} \left[ \frac{N}{\Delta_x \bar{Z}_x(Y_{1:N}, 1)} \log \left( \frac{N}{\Delta_x \bar{Z}_x(Y_{1:N}, 1)} \right) \right] \tag{234}$$

$$\leq \mathbb{E}_{Y_{1:N} \sim Q(.)} \left[ \frac{N}{\Delta_x \bar{Z}_x(Y_{1:N}, 1)} \log(8) \right] \tag{235}$$

$$= 3 \quad (\text{Since } \Delta_x = \mathbb{E}_{Y_{1:N} \sim Q(.)} \left[ \frac{N}{\bar{Z}_x(Y_{1:N}, 1)} \right]), \tag{236}$$

and hence $E_1 \leq 3$ bit.

*Case 2*: $N > 7\omega$

To upper-bound $E_2$ in this regime, we first note that:

$$\Delta_x = \mathbb{E}_{Y_{1:N} \sim Q(.)} \left[ \frac{N}{\bar{Z}_x(Y_{1:N}, 1)} \right] = \Pr(\text{Accept batch } B) \leq 1 \tag{237}$$

Another way to see this is through the following arguments:

$$\mathbb{E}_{Y_{1:N} \sim Q(.)} \left[ \frac{N}{\bar{Z}_x(Y_{1:N}, 1)} \right] \tag{238}$$

$$= \mathbb{E}_{Y_{1:N} \sim Q(.)} \left[ \frac{N P_x(Y_1)/Q(Y_1)}{\bar{Z}_x(Y_{1:N}, 1)} \right] \tag{239}$$

$$= \mathbb{E}_{Y_{1:N} \sim Q(.)} \left[ \frac{N P_x(Y_1)/Q(Y_1)}{\hat{Z}_x(Y_{1:N})} \frac{\hat{Z}_x(Y_{1:N})}{\bar{Z}_x(Y_{1:N}, 1)} \right] \tag{240}$$

$$\leq \mathbb{E}_{Y_{1:N} \sim Q(.)} \left[ \frac{N P_x(Y_1)/Q(Y_1)}{\hat{Z}_x(Y_{1:N})} \right] \quad \left( \text{Since } \frac{\hat{Z}_x(Y_{1:N})}{\bar{Z}_x(Y_{1:N}, 1)} \leq 1 \right) \tag{241}$$

$$= \sum_{i=1}^{N} \mathbb{E}_{Y_{1:N} \sim Q(.)} \left[ \frac{P_x(Y_i)/Q(Y_i)}{\hat{Z}_x(Y_{1:N})} \right] \quad (\text{Due to symmetry}) \tag{242}$$

$$= 1, \tag{243}$$

and as a consequence (which we will be using later), we have:

$$\mathbb{E}_{Y_{1:N} \sim Q(.)} \left[ \frac{N+1}{\omega_x + \hat{Z}_x(Y_{1:N})} \right] \tag{244}$$

$$= \mathbb{E}_{Y_1^{N+1} \sim Q(.)} \left[ \frac{N+1}{\omega_x + \hat{Z}_x(Y_{1:N})} \right] \tag{245}$$

$$\leq 1. \tag{246}$$

Then, observe that:

$$E_1 = \mathbb{E}_{Y_{1:N} \sim Q(.)} \left[ \frac{N}{\Delta_x \bar{Z}_x(Y_{1:N}, 1)} \log \left( \frac{N}{\Delta_x \bar{Z}_x(Y_{1:N}, 1)} \right) \right] \tag{247}$$

$$= \frac{1}{\Delta_x} \mathbb{E}_{Y_{1:N} \sim Q(.)} \left[ \frac{N}{\bar{Z}_x(Y_{1:N}, 1)} \log \left( \frac{N}{\bar{Z}_x(Y_{1:N}, 1)} \right) \right] + \log \frac{1}{\Delta_x} \tag{248}$$

$$\leq 3 \text{ bits} \tag{249}$$

where, to show the inequality at the end, we will prove the following two inequalities:

$$\log \frac{1}{\Delta_x} \leq \log \left( \frac{8}{7} \right) \tag{250}$$

$$\frac{1}{\Delta_x} \mathbb{E}_{Y_{1:N} \sim Q(.)} \left[ \frac{N}{\bar{Z}_x(Y_{1:N}, 1)} \log \left( \frac{N}{\bar{Z}_x(Y_{1:N}, 1)} \right) \right] \leq \frac{16}{7}, \tag{251}$$

and hence $E_2 \leq 3$ (bits). For the first inequality, we have:

$$\Delta_x = \mathbb{E}_{Y_{1:N} \sim Q(.)} \left[ \frac{N}{\bar{Z}_x(Y_{1:N}, 1)} \right] \tag{252}$$

$$\geq \frac{N}{\mathbb{E}_{Y_{1:N} \sim Q(.)}[\bar{Z}_x(Y_{1:N}, 1)]} \quad (\text{ Jensen's Inequality }) \tag{253}$$

$$= \frac{N}{N - 1 + \omega_x} \tag{254}$$

$$\geq \frac{N}{N - 1 + N/7} \quad (\text{ Since } N > 7\omega_x) \tag{255}$$

$$\geq \frac{7}{8}, \tag{256}$$

hence, we have:

$$\frac{1}{\Delta_x} \leq 8/7, \tag{257}$$

which yields the first inequality after taking the $\log(.)$ in both sides.

For the second inequality, we begin by establishing the following key inequality:

$$\frac{N}{\bar{Z}_x(Y_{1:N}, 1)} \leq \frac{2N}{\hat{Z}_x(Y_{1:N}) + \omega_x}, \tag{258}$$

which is due to:

$$\frac{N}{\bar{Z}_x(Y_{1:N}, 1)} = \frac{N}{\omega_x + \sum_{i=2}^{N} \frac{P_x(Y_i)}{Q(Y_i)}} \tag{259}$$

$$\leq \frac{N}{\omega_x + \frac{1}{2} \sum_{i=2}^{N} \frac{P_x(Y_i)}{Q(Y_i)}} \quad (\text{Since } \frac{P_x(Y_i)}{Q(Y_i)} \geq 0 \text{ for all } i) \tag{260}$$

$$= \frac{2N}{2\omega_x + \sum_{i=2}^{N} \frac{P_x(Y_i)}{Q(Y_i)}} \tag{261}$$

$$\leq \frac{2N}{\omega_x + \sum_{i=1}^{N} \frac{P_x(Y_i)}{Q(Y_i)}} \quad (\text{Since } \frac{P_x(Y_i)}{Q(Y_i)} \leq \omega \text{ for all } i) \tag{262}$$

$$= \frac{2N}{\hat{Z}_x(Y_{1:N}) + \omega_x}, \tag{263}$$

Then, we have:

$$\frac{1}{\Delta_x}\mathbb{E}_{Y_{1:N}\sim Q(.)}\left[\frac{N}{\bar{Z}_x(Y_{1:N},1)}\log\left(\frac{N}{\bar{Z}_x(Y_{1:N},1)}\right)\right] \tag{264}$$

$$\leq\frac{8}{7}\mathbb{E}_{Y_{1:N}\sim Q(.)}\left[\frac{N}{\bar{Z}_x(Y_{1:N},1)}\log\left(\frac{N}{\bar{Z}_x(Y_{1:N},1)}\right)\right] \quad \left(\text{Since }\Delta_x\geq\frac{7}{8}\text{ from (256)}\right) \tag{265}$$

$$=\frac{8}{7}\mathbb{E}_{Y_{1:N}\sim Q(.)}\left[\frac{NP_x(Y_1)/Q(Y_1)}{\bar{Z}_x(Y_{1:N},1)}\log\left(\frac{N}{\bar{Z}_x(Y_{1:N},1)}\right)\right] \tag{266}$$

$$\leq\frac{8}{7}\mathbb{E}_{Y_{1:N}\sim Q(.)}\left[\frac{NP_x(Y_1)/Q(Y_1)}{\bar{Z}_x(Y_{1:N},1)}\log\left(\frac{N}{\bar{Z}_x(Y_{1:N},1)}+1\right)\right] \tag{267}$$

$$\leq\frac{8}{7}\mathbb{E}_{Y_{1:N}\sim Q(.)}\left[\frac{NP_x(Y_1)/Q(Y_1)}{\bar{Z}_x(Y_{1:N},1)}\log\left(\frac{2N}{\hat{Z}_x(Y_{1:N})+\omega_x}+1\right)\right] \quad \text{(Due to Inequality (258) )} \tag{268}$$

$$\leq\frac{8}{7}\mathbb{E}_{Y_{1:N}\sim Q(.)}\left[\frac{NP_x(Y_1)/Q(Y_1)}{\hat{Z}_x(Y_{1:N})}\log\left(\frac{2N}{\hat{Z}_x(Y_{1:N})+\omega_x}+1\right)\right] \quad \text{(Since }\hat{Z}_x(Y_{1:N})\leq\bar{Z}_x(Y_{1:N},1))\tag{269}$$

$$=\frac{8}{7}\sum_{i=1}^{N}\mathbb{E}_{Y_{1:N}\sim Q(.)}\left[\frac{P_x(Y_i)/Q(Y_i)}{\hat{Z}_x(Y_{1:N})}\log\left(\frac{2N}{\hat{Z}_x(Y_{1:N})+\omega_x}+1\right)\right] \quad \text{(Due to symmetry)} \tag{270}$$

$$=\frac{8}{7}\mathbb{E}_{Y_{1:N}\sim Q(.)}\left[\frac{\sum_{i=1}^{N}P_x(Y_i)/Q(Y_i)}{\hat{Z}_x(Y_{1:N})}\log\left(\frac{2N}{\hat{Z}_x(Y_{1:N})+\omega_x}+1\right)\right] \tag{271}$$

$$=\frac{8}{7}\mathbb{E}_{Y_{1:N}\sim Q(.)}\left[\log\left(\frac{2N}{\hat{Z}_x(Y_{1:N})+\omega_x}+1\right)\right] \tag{272}$$

$$\leq\frac{8}{7}\log\left(\mathbb{E}_{Y_{1:N}\sim Q(.)}\left[\frac{2N}{\hat{Z}_x(Y_{1:N})+\omega_x}+1\right]\right) \quad \text{(Jensen's Inequality)} \tag{273}$$

$$=\frac{8}{7}\log\left(1+\frac{2N}{N+1}\mathbb{E}_{Y_{1:N}\sim Q(.)}\left[\frac{N+1}{\hat{Z}_x(Y_{1:N})+\omega_x}\right]\right) \tag{274}$$

$$\leq\frac{8}{7}\log\left(1+\frac{2N}{N+1}\right) \quad \left(\text{Since }\mathbb{E}_{Y_{1:N}\sim Q(.)}\left[\frac{N+1}{\hat{Z}_x(Y_{1:N})+\omega_x}\right]<1\text{ due to Inequality (246)}\right) \tag{275}$$

$$\leq\frac{8}{7}\log(4) \tag{276}$$

$$=\frac{16}{7}\text{(bits)} \tag{277}$$

which completes the proof for this part.

### E.4.2 Proof of Inequality (164)

We first express the quantity $\mathbb{E}[\log\hat{K}_1]$ with conditional expectation. The accepted batch and selected local index $K_2$ are distributed according to $Y_{K_1,1:N},K_2\sim\bar{P}_{Y_{1:N},K;x}$, then:

$$\mathbb{E}[\log\hat{K}_1] \tag{278}$$

$$=\mathbb{E}[\mathbb{E}[\log\hat{K}_1|Y_{K_1,1:N}=y_{1:N},K_2=k_2]] \tag{279}$$

$$=\sum_{k_2=1}^{N}\int_{-\infty}^{\infty}\left(\prod_{j=1,j\neq k_2}^{N}Q(y_j)\right)\frac{P_x(y_k)}{\bar{Z}_x(y_{1:N},k)\Delta_x}\mathbb{E}[\log\hat{K}_1|Y_{K_1,1:N}=y_{1:N},K_2=k_2]dy_{1:N} \tag{280}$$

$$=N\int_{-\infty}^{\infty}\left(\prod_{j=2}^{N}Q(y_j)\right)\frac{P_x(y_1)}{\bar{Z}_x(y_{1:N},1)\Delta_x}\mathbb{E}[\log\hat{K}_1|Y_{K_1,1:N}=y_{1:N},K_2=1]dy_{1:N} \tag{281}$$

Notice that, since we accept a batch $i$ when $U_i \leq \frac{\hat{Z}_x(y_{1:N})}{\bar{Z}_x(y_{1:N},1)} \frac{\Delta}{\Delta_x}$, we have that:

$$P(U_{K_1} = u | Y_{K_1,1:N} = y_{1:N}, K_2 = 1) = \frac{\bar{Z}_x(y_{1:N},1)}{\hat{Z}_x(y_{1:N})} \frac{\Delta_x}{\Delta},$$

then conditioning on $U_{K_1}$ for the last expectation term above:

$$\mathbb{E}[\log \hat{K}_1 | Y_{K_1,1:N} = y_{1:N}, K_2 = k] \tag{282}$$

$$= \int_{-\infty}^{\infty} \mathbb{E}[\log \hat{K}_1 | Y_{K_1,1:N} = y_{1:N}, K_2 = 1, U_{K_1} = u] P(U_{K_1} = u | Y_{K_1,1:N} = y_{1:N}, K_2 = 1) du \tag{283}$$

$$= \int_0^{\frac{\hat{Z}_x(y_{1:N})}{\bar{Z}_x(y_{1:N},1)} \frac{\Delta}{\Delta_x}} \mathbb{E}[\log \hat{K}_1 | Y_{K_1,1:N} = y_{1:N}, K_2 = 1, U_{K_1} = u] \frac{\bar{Z}_x(y_{1:N},1)}{\hat{Z}_x(y_{1:N})} \frac{\Delta_x}{\Delta} du \tag{284}$$

$$\leq \frac{\Delta_x}{\Delta} \int_0^{\frac{\hat{Z}_x(y_{1:N})}{\bar{Z}_x(y_{1:N},1)} \frac{\Delta}{\Delta_x}} \frac{\bar{Z}_x(y_{1:N},1)}{\hat{Z}_x(y_{1:N})} \log\left[1 + \frac{u}{\Delta}\right] du \quad \text{(See the Sort Coding bound below.)} \tag{285}$$

$$\leq \frac{\Delta_x}{\Delta} \int_0^{\frac{\hat{Z}_x(y_{1:N})}{\bar{Z}_x(y_{1:N},1)} \frac{\Delta}{\Delta_x}} \frac{\bar{Z}_x(y_{1:N},1)}{\hat{Z}_x(y_{1:N})} \log\left[1 + \frac{\hat{Z}_x(y_{1:N})}{\Delta_x \bar{Z}_x(y_{1:N},1)}\right] du \tag{286}$$

$$= \log\left[1 + \frac{\hat{Z}_x(y_{1:N})}{\Delta_x \bar{Z}_x(y_{1:N},1)}\right], \tag{287}$$

Finally, we have:

$$\mathbb{E}[\log \hat{K}_1] \tag{288}$$

$$= N \int_{-\infty}^{\infty} \left(\prod_{j=2}^N Q(y_j)\right) \frac{P_x(y_1)}{\bar{Z}_x(y_{1:N},1)\Delta_x} \log\left[1 + \frac{\hat{Z}_x(y_{1:N})}{\Delta_x \bar{Z}_x(y_{1:N},1)}\right] dy_{1:N} \tag{289}$$

$$\leq N \int_{-\infty}^{\infty} \left(\prod_{j=2}^N Q(y_j)\right) \frac{P_x(y_1)}{\bar{Z}_x(y_{1:N},1)\Delta_x} \left(\log\left[\frac{\hat{Z}_x(y_{1:N})}{\Delta_x \bar{Z}_x(y_{1:N},1)}\right] + \log e \frac{\Delta_x \bar{Z}_x(y_{1:N},1)}{\hat{Z}_x(y_{1:N})}\right) dy_{1:N} \tag{290}$$

$$= N \int_{-\infty}^{\infty} \left(\prod_{j=2}^N Q(y_j)\right) \frac{P_x(y_1)}{\bar{Z}_x(y_{1:N},1)\Delta_x} \log\left[\frac{\hat{Z}_x(y_{1:N})}{\Delta_x \bar{Z}_x(y_{1:N},1)}\right] dy_{1:N} + \log(e) \tag{291}$$

$$= \frac{N}{\Delta_x} \mathbb{E}_{Y_{1:N} \sim Q}\left[\frac{P_x(Y_1)/Q(Y_1)}{\bar{Z}_x(Y_{1:N},1)} \log\left(\frac{\hat{Z}_x(Y_{1:N})}{\Delta_x \bar{Z}_x(Y_{1:N},1)}\right)\right] \tag{292}$$

We show the proof for (285) below.

**Sort Coding Bound.** To bound the expectation term, we first apply Jensen's inequality and conditioning on the accepted chunk of batches $L = \ell$:

$$\mathbb{E}[\log \hat{K}_1 | Y_{K_1,1:N} = y_{1:N}, K_2 = 1, U_{K_1} = u] \tag{293}$$

$$\leq \log(\mathbb{E}[\hat{K}_1 | Y_{K_1,1:N} = y_{1:N}, K_2 = 1, U_{K_1} = u]) \tag{294}$$

$$= \log(\mathbb{E}[\hat{K}_1 | Y_{K_1,1:N} = y_{1:N}, K_2 = 1, U_{K_1} = u]) \tag{295}$$

$$= \log(\mathbb{E}_L[\mathbb{E}[\hat{K}_1 | Y_{K_1,1:N} = y_{1:N}, K_2 = 1, U_{K_1} = u, L = \ell]]) \tag{296}$$

We now repeat the previous argument in standard RS. Specifically, given $\hat{K}_1$ is within the range $L = \ell$ and $U_{K_1} = u$, we can express $\hat{K}_1$ as follows:

$$\hat{K}_1 = |\{U_i < u, (\ell - 1)\lfloor \Delta^{-1}\rfloor + 1 \leq i \leq \ell\lfloor \Delta^{-1}\rfloor\}| + 1, \tag{297}$$

$$= \Omega(u, \ell) + 1 \tag{298}$$

i.e. the number of $U_i$ (plus 1 for the ranking) within the range $L$ that has value lesser than $u$.

We can see that the the index $i$ within the range $L$ satisfying $U_i < u$ are from the indices that are either (1) rejected, i.e. index $i < \hat{K}_1$ or (2) not examined by the algorithm, i.e. index $i > \hat{K}_1$. The rest of this proof will show the following bound:

$$\mathbb{E}[\Omega(u,\ell)|Y_{K_1,1:N} = y_{1:N}, K_2 = 1, U_{K_1} = u, L = \ell] \leq \Delta^{-1}u, \text{ for any } \ell \tag{299}$$

For readability, we split the proof into different proof steps.

**Proof Step 1:** We condition on the mapped index of $\pi(\hat{K})$ on the original array:

$$\mathbb{E}[\hat{K}_1|Y_{K_1,1:N} = y_{1:N}, K_2 = 1, U_{K_1} = u, L = \ell] \tag{300}$$

$$= \mathbb{E}_{\pi(\hat{K}_1)}\left[\mathbb{E}[\hat{K}_1 \mid Y_{K_1,1:N} = y_{1:N}, K_2 = 1, U_{K_1} = u, L = \ell, \pi(\hat{K}_1) = k_1]\right] \tag{301}$$

$$= \mathbb{E}_{\pi(\hat{K}_1)}\left[\mathbb{E}[\Omega(u,\ell) + 1 \mid Y_{K_1,1:N} = y_{1:N}, K_2 = 1, U_{K_1} = u, L = \ell, \pi(\hat{K}_1) = k_1]\right] \tag{302}$$

$$= \mathbb{E}_{\pi(\hat{K}_1)}\left[\mathbb{E}[\Omega(u,\ell) \mid Y_{K_1,1:N} = y_{1:N}, K_2 = 1, U_{K_1} = u, L = \ell, \pi(\hat{K}_1) = k_1]\right] + 1 \tag{303}$$

$$= \mathbb{E}_{\pi(\hat{K}_1)}\left[\mathbb{E}[\Omega_1(u,\ell,k_1) + \Omega_2(u,\ell,k_1) \mid Y_{K_1,1:N} = y_{1:N}, K_2 = 1, U_{K_1} = u, L = \ell, \pi(\hat{K}_1) = k_1]\right] + 1, \tag{304}$$

where $\Omega_1(u,\ell,k_1), \Omega_2(u,\ell,k_1)$ are the number of $U_i < u$ within the range $L = \ell$ that occurs before and after the selected index $k_1$ respectively. Specifically:

$$\Omega_1(u,\ell,k_1) = |\{U_i < u, (\ell-1)\lfloor\Delta^{-1}\rfloor + 1 \leq i < (\ell-1)\lfloor\Delta^{-1}\rfloor + k_1\}| \tag{305}$$

$$\Omega_2(u,\ell,k_1) = |\{U_i < u, (\ell-1)\lfloor\Delta^{-1}\rfloor + k_1 + 1 \leq i \leq \ell\lfloor\Delta^{-1}\rfloor\}|, \tag{306}$$

which also naturally gives $\Omega(u,\ell) = \Omega_1(u,\ell,k_1) + \Omega_2(u,\ell,k_1)$.

**Proof Step 2:** Consider $\Omega_2(u,\ell,k_1)$, since each proposal $(Y_{i,1:N}, U_i)$ is i.i.d distributed and the fact that $k_1$ is the index of the *first accepted batch*, for every $i > k_1$, we have:

$$\Pr(U_i < u \mid \bar{Y}_{1:N} = y_{1:N}, K_2 = 1, U_{K_1} = u, L = \ell, \pi(\hat{K}_1) = k_1) = \Pr(U_i < u)$$

This gives us:

$$\mathbb{E}[\Omega_2(u,\ell,k_1) \mid Y_{K_1,1:N} = y_{1:N}, K_2 = 1, U_{K_1} = u, L = \ell, \pi(\hat{K}_1) = k_1] \tag{307}$$

$$= (\lfloor\Delta^{-1}\rfloor - k_1)\Pr(U < u) \tag{308}$$

$$= (\lfloor\Delta^{-1}\rfloor - k_1)u \tag{309}$$

$$\leq \frac{(\lfloor\Delta^{-1}\rfloor - k_1)u}{\Pr(\text{Batch is rejected})} \tag{310}$$

$$\leq \frac{(\lfloor\Delta^{-1}\rfloor - k_1)u}{1 - \Delta} \tag{311}$$

**Proof Step 3:** For $\Omega_1(u,\ell,\hat{k}_1)$, we do not have such independent property since for every batch with index $i < k_1$, we know that they are rejected batches, and hence for $i < k_1$:

$$\Pr(U_i < u \mid Y_{K_1,1:N} = y_{1:N}, K_2 = k, U_{K_1} = u, L = \ell, \pi(\hat{K}_1) = k_1) \tag{312}$$

$$= \Pr(U_i < u|Y_{i,1:N} \text{ is rejected}) \tag{313}$$

$$= \frac{\Pr(U_i < u, Y_{i,1:N} \text{ is rejected})}{\Pr(Y_{i,1:N} \text{ is rejected})} \tag{314}$$

$$\leq \frac{\Pr(U_i < u)}{\Pr(Y_{i,1:N} \text{ is rejected})} \tag{315}$$

$$= \frac{u}{1 - \Delta}, \tag{316}$$

which gives us:

$$\mathbb{E}[\Omega_2(u,\ell,k_1) \mid Y_{K_1,1:N} = y_{1:N}, K_2 = 1, U_{K_1} = u, L = \ell, \pi(\hat{K}_1) = k_1] \leq \frac{(k_1 - 1)u}{1 - \Delta} \tag{317}$$

To prove Equation (313), note that the following events are equivalent:

$$\{Y_{K_1,1:N} = y_{1:N}, K_2 = 1, U_{K_1} = u, L = \ell, \pi(\hat{K}_1) = k_1\} \tag{318}$$

$$= \{Y_{k_1,1:N} = y_{1:N}, K_2 = 1, U_k = u, B_{1,\dots,k-1} \text{ are rejected}\} \tag{319}$$

$$\triangleq \Lambda(u, y, k_1) \tag{320}$$

Here, we note that $Y_{k_1}, U_{k_1}$ denote the value at batch index $k$ within $W$, which is different from $Y_{K_1}, U_{K_1}$, the value selected by the rejection sampler. Hence:

$$\Pr(U_i < u | \Lambda(u, y, k_1)) \tag{321}$$

$$= \frac{\Pr(U_i < u, B_{1\dots k_1 - 1} \text{ are rejected} | Y_{k,1:N} = y_{1:N}, U_k = u, K_2 = 1)}{\Pr(B_{1\dots k_1 - 1} \text{ are rejected} | Y_{k,1:N} = y_{1:N}, U_k = u, K_2 = 1)} \tag{322}$$

$$= \frac{\Pr(U_i < u, \ B_{1\dots k_1 - 1} \text{ are rejected})}{\Pr(B_{1\dots k_1 - 1} \text{ are rejected})} \quad \text{(Since } (Y_i, U_i) \text{ are i.i.d)} \tag{323}$$

$$= \Pr(U_i < u | B_i \text{ is rejected}), \tag{324}$$

**Proof Step 4:** From the above result from Step 2 and 3, we have $\Omega(u, \ell) = \Omega_1(u, \ell, k) + \Omega_2(u, \ell, k) \leq \frac{(\lfloor \Delta^{-1} \rfloor - 1)u}{1 - \Delta}$ and as a result:

$$\mathbb{E}[\log \hat{K}_1 | Y_{K_1,1:N} = y_{1:N}, K_2 = 1, U_{K_1} = u] \leq \frac{(\lfloor \Delta^{-1} \rfloor - 1)u}{1 - \Delta} + 1 \tag{325}$$

$$\leq \frac{(\Delta^{-1} - 1)u}{1 - \Delta} + 1 \quad (\text{Since} \lfloor \Delta^{-1} \rfloor \leq \Delta^{-1}) \tag{326}$$

$$= \Delta^{-1}u + 1 \tag{327}$$

which completes the proof.

# F ERS Matching Lemmas

## F.1 Preliminaries

We begin by providing the following bounds on inverse moments of averages.

**Proposition F.1.** *Let $Y_1, Y_2, ..., Y_N \sim Q_Y(.)$ and suppose the target distribution $P_Y$ satifies:*

$$d_2(Q_Y||P_Y) \triangleq \mathbb{E}_{Y \sim Q_Y(.)} \left[ \frac{Q_Y(Y)}{P_Y(Y)} \right] < \infty, \tag{328}$$

*then we have:*

$$\mathbb{E}_{Y_{1:N} \sim Q_Y(.)} \left[ \frac{N}{\sum_{i=1}^{N} \frac{P_Y(Y_i)}{Q_Y(Y_i)}} \right] \leq d_2(Q_Y||P_Y). \tag{329}$$

*Proof.* Applying the Cauchy-Schwarz inequality, we have:

$$\frac{N}{\sum_{i=1}^{N} \frac{P_Y(Y_i)}{Q_Y(Y_i)}} \leq \frac{1}{N} \sum_{j=1}^{N} \frac{Q_Y(Y_i)}{P_Y(Y_i)} \tag{330}$$

Taking the expectation of both sides yield the desired inequality. □

**Remark F.2.** *In general, stronger results on the inverse moments of averages exist under weaker moment assumptions, specifically:*

$$\mathbb{E}_{Y \sim Q_Y} \left[ \left( \frac{Q_Y(Y)}{P_Y(Y)} \right)^{\eta} \right]$$

*is finite for some $\eta < 0$. The resulting bound has a similar form (some power terms involved) to that of Proposition F.1 but requires a mild threshold on $N$. For further details, see Proposition A.1 in [10].*

We show an application of Proposition F.1, which we will use repeatly:

**Corollary F.3.** *Let $Y_1, Y_2, ..., Y_N \sim Q_Y(.)$ and suppose the target distributions $P_Y^A, P_Y^B$ satisfy:*

$$d_2(Q_Y||P_Y^A) \triangleq \mathbb{E}_{Y \sim Q_Y(.)} \left[ \frac{Q_Y(Y)}{P_Y^A(Y)} \right] < \infty, \quad and \quad \frac{P_Y^A(y)}{Q_Y(y)}, \frac{P_Y^B(y)}{Q_Y(y)} \leq \omega \text{ for all } y. \tag{331}$$

*Then, for any $N \geq 1$,*

$$\mathbb{E}_{Y_{1:N} \sim Q_Y(.)} \left[ \frac{\sum_{j=1}^{N} \frac{P_Y^B(Y_j)}{Q_Y(Y_j)}}{\sum_{i=1}^{N} \frac{P_Y^A(Y_i)}{Q_Y(Y_i)}} \right] \leq \mathbb{I}_N(\omega, 1) \cdot d_2(Q_Y||P_Y), \tag{332}$$

*where we define $\mathbb{I}_N(\omega, i) \triangleq (2\mathbb{1}_{N>i} + \omega\mathbb{1}_{N=i})$.*

*Proof.* For $N = 1$, applying the conditions for $P_Y^A$ and $P_Y^B$ gives us an upper-bound of $\omega d_2(Q_Y||P_Y^A)$. For $N > 1$, we have:

$$\mathbb{E}_{Y_{1:N} \sim Q_Y(.)} \left[ \frac{\sum_{j=1}^{N} \frac{P_Y^B(Y_i)}{Q_Y(Y_i)}}{\sum_{i=1}^{N} \frac{P_Y^A(Y_i)}{Q_Y(Y_i)}} \right] = N\mathbb{E}_{Y_{1:N} \sim Q_Y(.)} \left[ \frac{\frac{P_Y^B(Y_1)}{Q_Y(Y_1)}}{\sum_{i=1}^{N} \frac{P_Y^A(Y_i)}{Q_Y(Y_i)}} \right] \quad \text{(Due to symmetry)} \tag{333}$$

$$\leq N\mathbb{E}_{Y_{1:N} \sim Q_Y(.)} \left[ \frac{\frac{P_Y^B(Y_1)}{Q_Y(Y_1)}}{\sum_{i=2}^{N} \frac{P_Y^A(Y_i)}{Q_Y(Y_i)}} \right] \quad \left(\text{since } \frac{P_Y^A(Y_i)}{Q_Y(Y_i)} \geq 0\right) \tag{334}$$

$$= \frac{N}{N-1}\mathbb{E}_{Y_{1:N} \sim Q_Y(.)} \left[ \frac{N-1}{\sum_{i=2}^{N} \frac{P_Y^A(Y_i)}{Q_Y(Y_i)}} \right] \tag{335}$$

$$\leq 2d_2(Q_Y||P_Y^A) \quad \text{(Proposition F.1 and } N > 1), \tag{336}$$

which completes the proof. □

## F.2 Distributed Matching Without Batch Communication

Before the proof, we outline the details of each case in Section 3.2, covering scenarios without and with communication between the encoder and decoder.

**Without-Communication.** In this scenario, let $P_Y^A(.)$ and $P_Y^B(.)$ be the target distributions at the encoder and decoder respectively, where we use the same shared randomness $W$ as in Section 5.1 where we use the proposal distribution $Q_Y(.)$. Furthermore, we assume that:

$$\max_y \left( \frac{P_Y^A(y)}{Q_Y(y)} \right) = \omega_A, \quad \max_y \left( \frac{P_Y^B(y)}{Q_Y(y)} \right) = \omega_B, \quad \max_y \left( \frac{P_Y^A(y)}{Q_Y(y)}, \frac{P_Y^B(y)}{Q_Y(y)} \right) \le \omega, \quad (337)$$

Using the ERS procedure, the encoder and decoder select the indices $K_A$ and $K_B$ respectively.

$$K_A = \mathrm{ERS}(W; P_Y^A, Q_Y), \quad K_B = \mathrm{ERS}(W; P_Y^B, Q_Y), \quad (338)$$

The ERS($\cdot$) function follows Algorithm 1, without requiring any specific scaling factor. During the selection process, the calculation of $\bar{Z}$ in Step 3 of this algorithm, which determines the acceptance probability, uses the ratio upper bounds $\omega_A$ and $\omega_B$ for parties $A$ and $B$, respectively. Proposition F.4 establishes the bound on the probability that both parties produce the same output, conditioned on $Y_{K_A} = y$.

**Proposition F.4.** *Let $K_A, K_B$ and $P_Y^A, P_Y^B$ defined as above and $N \ge 2$, we have:*

$$\Pr(Y_{K_A} = Y_{K_B} | Y_{K_A} = y) \ge \left( 1 + \mu_1(N) + \frac{P_Y^A(y)}{P_Y^B(y)} (1 + \mu_2(N)) \right)^{-1}, \quad (339)$$

*where $\mu_1(N)$ and $\mu_2(N)$ are defined as in Appendix F.3 and we note that $\mu_1(N), \mu_2(N) \to 0$ as $N \to \infty$ under mild assumptions on the distributions $P_Y^A, P_Y^B$ and $Q_Y$.*

*Proof.* See Appendix F.4. □

**With Communication.** Following the setup described in Section 5.3, we define the ratio upperbounds in the communication case as below:

$$\max_z \left( \frac{P_{Y|Z}(y|x)}{Q_Y(y)} \right) = \omega_x, \quad \max_z \left( \frac{\tilde{P}_{Y|Z}(y|z)}{Q_Y(y)} \right) = \omega_z, \quad \max_{y,z} \left( \frac{P_{Y|X}(y|x)}{Q_Y(y)}, \frac{\tilde{P}_{Y|Z}(y|z)}{Q_Y(y)} \right) \le \omega,$$

and similar to the case without communication, the ERS(.) selection process at the encoder and decoder also follows Algorithm 1, with the calculation of $\bar{Z}$ in Step 3 uses the upperbound $\omega_x$ and $\omega_z$ respectively for the encoder and decoder. The bound for this case is shown below.

**Proposition F.5.** *For $N \ge 2$ and $X, Y, Z$ defined as above, we have:*

$$\Pr(Y_{K_A} = Y_{K_B} | Y_{K_A} = y, X = x, Z = z) \ge \left( 1 + \mu_1^{\mathrm{cond}}(N) + \frac{P_{Y|X}(y|x)}{\tilde{P}_{Y|Z}(y|z)} (1 + \mu_2^{\mathrm{cond}}(N)) \right)^{-1}, \quad (340)$$

*where $\mu_1^{\mathrm{cond}}(N)$ and $\mu_2^{\mathrm{cond}}(N)$ are defined as in Appendix F.5 and we note that $\mu_1^{\mathrm{cond}}(N), \mu_2^{\mathrm{cond}}(N) \to 0$ as $N \to \infty$ under mild assumptions on the distributions $P_{Y|X}(.|x), \tilde{P}_{Y|Z}(.|z)$ and $Q_Y(.)$.*

*Proof.* See Appendix F.6. □

## F.3 Coefficients in Proposition F.4

We first define the coefficient $\mu_1(N)$ and $\mu_2(N)$ in Proposition F.4.

$$\mu_1(N) = \frac{1}{N} \left[ \omega + \omega \mathbb{I}_N(\omega, 2) d_2(Q_Y || P_Y^B) + \frac{\omega^2}{N-1} d_2(Q_Y || P_Y^B) \right] \quad (341)$$

$$\mu_2(N) = \frac{1}{N} \left[ \omega + \omega \mathbb{I}_N(\omega, 2) d_2(Q_Y || P_Y^A) + \frac{\omega^2}{N-1} d_2(Q_Y || P_Y^A) \right] \quad (342)$$

where we define $\mathbb{I}_N(\omega, i) \triangleq (2\mathbb{1}_{N>i} + \omega \mathbb{1}_{N=i})$ as in Proposition F.3.

### F.4 Proof of Proposition F.4

We prove the matching probability for the case of ERS. We note that in this proof, we will use the global index for the proposals $Y_1, ... Y_N \sim Q(.)$ instead of $Y_{1,1}, ... Y_{1,N}$ unless otherwise stated. First, consider:

$$\Pr(Y_{K_A} = Y_{K_B} | Y_{K_A} = y_1) \tag{343}$$

$$\geq \Pr(K_A = K_B | Y_{K_A} = y_1) \tag{344}$$

$$= \sum_{k=1}^{\infty} \Pr(K_A = K_B = k | Y_{K_A} = y_1) \tag{345}$$

$$\geq \sum_{k=1}^{N} \Pr(K_A = K_B = k | Y_{K_A} = y_1) \tag{346}$$

$$= N \Pr(K_A = K_B = 1 | Y_{K_A} = y_1) \tag{347}$$

$$= \frac{N Q_Y(y_1)}{P_Y^A(y_1)} \Pr(K_{2,A} = K_{2,B} = 1, K_{1,A} = K_{1,B} = 1 | Y_1 = y_1) \tag{348}$$

$$= \frac{N Q_Y(y_1)}{P_Y^A(y_1)} \int \Pr(K_{2,A} = K_{2,B} = 1, K_{1,A} = K_{1,B} = 1, Y_{2:N} = y_{2:N} | Y_1 = y_1) dy_{2:N} \tag{349}$$

$$= \frac{N Q_Y(y_1)}{P_Y^A(y_1)} \int \Pr(K_{2,A} = K_{2,B} = 1, K_{1,A} = K_{1,B} = 1 | Y_{1:N} = y_{1:N}) Q_Y(y_{2:N}) dy_{2:N} \tag{350}$$

$$= \frac{N Q_Y(y_1)}{P_Y^A(y_1)} \int \Pr(K_{2,A} = K_{2,B} = 1 | Y_{1:N} = y_{1:N})$$
$$\times \Pr(K_{1,A} = K_{1,B} = 1 | K_{2,A} = K_{2,B} = 1, Y_{1:N} = y_{1:N}) Q_Y(y_{2:N}) dy_{2:N} \tag{351}$$

$$= \frac{N Q_Y(y_1)}{P_Y^A(y_1)} \mathbb{E}_{Y_{2:N} \sim Q_Y(.)} [\Pr(K_{2,A} = K_{2,B} = 1 | Y_{1:N} = y_{1:N})$$
$$\times \Pr(K_{1,A} = K_{1,B} = 1 | K_{2,A} = K_{2,B} = 1, Y_{1:N} = y_{1:N})] \tag{352}$$

where (348) is due to the following fact that:

$$\{K_A = K_B = 1, Y_{K_A} = y_1\} = \{K_A = K_B = 1, Y_1 = y_1\}, \tag{353}$$

and thus:

$$\Pr(K_A = K_B = 1 | Y_{K_A} = y_1) = \frac{\Pr(K_A = K_B = 1 | Y_1 = y_1) Q_Y(y_1)}{P(Y_{K_A} = y_1)} \tag{354}$$

$$= \frac{\Pr(K_A = K_B = 1 | Y_1 = y_1) Q_Y(y_1)}{P_Y^A(y_1)} \tag{355}$$

$$\tag{356}$$

Define:

$$\hat{Z}(P_Y^A, y_{1:N}) = \sum_{i=1}^{N} \frac{P_Y^A(y_i)}{Q_Y(y_i)}, \quad \hat{Z}(P_Y^B, y_{1:N}) = \sum_{i=1}^{N} \frac{P_Y^B(y_i)}{Q_Y(y_i)} \tag{357}$$

Now, we note that:

$$\Pr(K_{2,A} = K_{2,B} = 1 | Y_{1:N} = y_{1:N}) \tag{358}$$

$$= \Pr(K_{2,A} = 1 | Y_{1:N} = y_{1:N}) \Pr(K_{2,B} = 1 | Y_{1:N} = y_{1:N}, K_{2,A} = 1) \tag{359}$$

$$= \frac{P_Y^A(y_1)/Q_Y(y_1)}{\sum_{i=1}^{N} P_Y^A(y_i)/Q_Y(y_i)} \Pr(K_{2,B} = 1 | Y_{1:N} = y_{1:N}, K_{2,A} = 1) \tag{360}$$

$$\geq \frac{P_Y^A(y_1)/Q_Y(y_1)}{\hat{Z}(P_Y^A, y_{1:N})} \left(1 + \frac{P_Y^A(y_1)}{P_Y^B(y_1)} \cdot \frac{\hat{Z}(P_Y^B, y_{1:N})}{\hat{Z}(P_Y^A, y_{1:N})}\right)^{-1}, \tag{361}$$

where we denote $\hat{Z}(P_Y^A, y_{1:N}) = \sum_{i=1}^N P_Y^A(y_i)/Q_Y(y_i)$ and the last inequality is due to Proposition 1 in [35]. Also:

$$\Pr(K_{1,A}(1)=K_{1,B}(1)=1 | K_{2,A}=K_{2,B}=1, Y_{1:N}=y_{1:N}) \tag{362}$$

$$\geq \min\left( \frac{\hat{Z}(P_Y^A, y_{1:N})}{\hat{Z}(P_Y^A, y_{2:N})+\omega}, \frac{\hat{Z}(P_Y^B, y_{1:N})}{\hat{Z}(P_Y^B, y_{2:N})+\omega} \right) \left( \text{ Since } \omega \geq \max_y \left( \frac{P_Y^A(y)}{Q_Y(y)}, \frac{P_Y^B(y)}{Q_Y(y)} \right) \right) \tag{363}$$

$$\geq \left( \frac{\hat{Z}(P_Y^A, y_{1:N})}{\hat{Z}(P_Y^A, y_{2:N})+\omega} \right) \left( \frac{\hat{Z}(P_Y^B, y_{1:N})}{\hat{Z}(P_Y^B, y_{2:N})+\omega} \right), \tag{364}$$

where we use the inequality $\min(a,b) \geq ab$ for $0 \leq a, b \leq 1$. Plug both in (352), we have:

$$\Pr(K_A=K_B | Y_{K_A}=y_1) \tag{365}$$

$$\geq \mathbb{E}_{Y_{2:N} \sim Q_Y(.)} \left[ \frac{1}{\left(1 + \frac{P_Y^A(y_1)}{P_Y^B(y_1)} \cdot \frac{\hat{Z}(P_Y^B,y_{1:N})}{\hat{Z}(P_Y^A,y_{1:N})}\right)} \left( \frac{N}{\hat{Z}(P_Y^A, y_{2:N})+\omega} \right) \left( \frac{\hat{Z}(P_Y^B, y_{1:N})}{\sum_{i=2}^N \hat{Z}(P_Y^B, y_{2:N})+\omega} \right) \right]$$

$$= \mathbb{E}_{Y_{2:N} \sim Q_Y(.)} \left[ \frac{1}{\left(1 + \frac{P_Y^A(y_1)}{P_Y^B(y_1)} \cdot \frac{\hat{Z}(P_Y^B,y_{1:N})}{\hat{Z}(P_Y^A,y_{1:N})}\right) \left( \frac{\hat{Z}(P_Y^A,y_{2:N})+\omega}{N} \right) \left( \frac{\hat{Z}(P_Y^B,y_{2:N})+\omega}{\hat{Z}(P_Y^B,y_{1:N})} \right)} \right] \tag{366}$$

$$\geq \left( \mathbb{E}_{Y_{2:N} \sim Q_Y(.)} \left[ \left(1 + \frac{P_Y^A(y_1)}{P_Y^B(y_1)} \cdot \frac{\hat{Z}(P_Y^B,y_{1:N})}{\hat{Z}(P_Y^A,y_{1:N})}\right) \left( \frac{\hat{Z}(P_Y^A,y_{2:N})+\omega}{N} \right) \left( \frac{\hat{Z}(P_Y^B,y_{2:N})+\omega}{\hat{Z}(P_Y^B,y_{1:N})} \right) \right] \right)^{-1} \tag{367}$$

$$= \left( \mathbb{E}_{Y_{2:N} \sim Q_Y(.)} \left[ \zeta_1 + \frac{P_Y^A(y_1)}{P_Y^B(y_1)} \zeta_2 \right] \right)^{-1}, \tag{368}$$

where we use Jensen's inequality for the convex function $1/x$ in line (367) and set:

$$\zeta_1 = \left( \frac{\hat{Z}(P_Y^A, y_{2:N})+\omega}{N} \right) \left( \frac{\hat{Z}(P_Y^B, y_{2:N})+\omega}{\hat{Z}(P_Y^B, y_{1:N})} \right) \tag{369}$$

$$= \frac{\hat{Z}(P_Y^A, y_{2:N}) \cdot \hat{Z}(P_Y^B, y_{2:N})}{N \hat{Z}(P_Y^B, y_{1:N})} + \frac{\omega}{N} \cdot \frac{\hat{Z}(P_Y^A, y_{2:N})}{\hat{Z}(P_Y^B, y_{1:N})} + \frac{\omega}{N} \cdot \frac{\hat{Z}(P_Y^B, y_{2:N})}{\hat{Z}(P_Y^B, y_{1:N})} + \frac{\omega^2}{N \cdot \hat{Z}(P_Y^B, y_{1:N})}$$

$$\leq \frac{1}{N} \hat{Z}(P_Y^A, y_{2:N}) + \frac{\omega}{N} \cdot \frac{\hat{Z}(P_Y^A, y_{2:N})}{\hat{Z}(P_Y^B, y_{2:N})} + \frac{\omega}{N} + \frac{\omega^2}{N \hat{Z}(P_Y^B, y_{2:N})}, \tag{370}$$

with the last inequality due to $\sum_{i=1}^N z_i \geq \sum_{i=2}^N z_i$ for any positive $z$. We then have:

$$\mathbb{E}_{y_{2:N} \sim Q_Y(.)}[\zeta_1] \tag{371}$$

$$\leq \mathbb{E}_{y_{2:N} \sim Q_Y(.)} \left[ \frac{\hat{Z}(P_Y^A, y_{2:N})}{N} + \frac{\omega}{N} \cdot \frac{\hat{Z}(P_Y^A, y_{2:N})}{\hat{Z}(P_Y^B, y_{2:N})} + \frac{\omega}{N} + \frac{\omega^2}{N \hat{Z}(P_Y^B, y_{2:N})} \right] \tag{372}$$

$$= \frac{N-1}{N} + \frac{\omega}{N} + \frac{\omega}{N} \mathbb{E}_{y_{2:N} \sim Q_Y(.)} \left[ \frac{\hat{Z}(P_Y^A, y_{2:N})}{\hat{Z}(P_Y^B, y_{2:N})} \right] + \frac{\omega^2}{N} \mathbb{E}_{y_{2:N} \sim Q_Y(.)} \left[ \frac{1}{\hat{Z}(P_Y^B, y_{2:N})} \right] \tag{373}$$

$$\leq 1 + \frac{1}{N} \left( \omega + \omega \mathbb{E}_{y_{2:N} \sim Q_Y(.)} \left[ \frac{\hat{Z}(P_Y^A, y_{2:N})}{\hat{Z}(P_Y^B, y_{2:N})} \right] + \omega^2 \mathbb{E}_{y_{2:N} \sim Q_Y(.)} \left[ \frac{1}{\hat{Z}(P_Y^B, y_{2:N})} \right] \right) \tag{374}$$

$$\leq 1 + \frac{1}{N} \left[ \omega + \omega \mathbb{I}_N(\omega, 2) d_2(Q_Y || P_Y^B) + \frac{\omega^2}{N-1} d_2(Q_Y || P_Y^B) \right] \tag{375}$$

$$= 1 + \mu_1(N), \tag{376}$$

where the last inequality is due to Proposition F.1 and Corollary F.3.. For the other term, we have:

$$\zeta_2 = \left( \frac{\hat{Z}(P_Y^B, y_{1:N})}{\hat{Z}(P_Y^A, y_{1:N})} \right) \left( \frac{\hat{Z}(P_Y^A, y_{2:N}) + \omega}{N} \right) \left( \frac{\hat{Z}(P_Y^B, y_{2:N}) + \omega}{\hat{Z}(P_Y^B, y_{1:N})} \right) \tag{377}$$

$$= \frac{1}{N} \left( \frac{\hat{Z}(P_Y^A, y_{2:N})}{\hat{Z}(P_Y^A, y_{1:N})} + \frac{\omega}{\hat{Z}(P_Y^A, y_{1:N})} \right) \left( \hat{Z}(P_Y^B, y_{2:N}) + \omega \right) \tag{378}$$

$$\leq \frac{1}{N} \left( 1 + \frac{\omega}{\hat{Z}(P_Y^A, y_{2:N})} \right) \left( \hat{Z}(P_Y^B, y_{2:N}) + \omega \right) \tag{379}$$

$$= \frac{\hat{Z}(P_Y^B, y_{2:N})}{N} + \frac{1}{N} \left( \omega + \frac{\omega \hat{Z}(P_Y^B, y_{2:N})}{\hat{Z}(P_Y^A, y_{2:N})} + \frac{\omega^2}{\hat{Z}(P_Y^A, y_{2:N})} \right). \tag{380}$$

where we again repeatedly use the inequality $\sum_{i=1}^{N} z_i \geq \sum_{i=2}^{N} z_i$ for any positive $z$. This gives us:

$$\mathbb{E}_{y_{2:N} \sim Q_Y(.)}[\zeta_2] \tag{381}$$

$$\leq \frac{1}{N} \left( \omega + \omega \mathbb{E}_{y_{2:N} \sim Q_Y()} \left[ \frac{\hat{Z}(P_Y^B, y_{2:N})}{\hat{Z}(P_Y^A, y_{2:N})} \right] + \omega^2 \mathbb{E}_{y_{2:N} \sim Q_Y()} \left[ \frac{1}{\hat{Z}(P_Y^A, y_{2:N})} \right] \right) \tag{382}$$

$$\leq \frac{1}{N} \left[ \omega + \omega \mathbb{I}_N(\omega, 2) d_2(Q_Y || P_Y^A) + \frac{\omega^2}{N-1} d_2(Q_Y || P_Y^A) \right] \tag{383}$$

$$= \mu_2(N), \tag{384}$$

where the last inequality is due to Proposition F.1 and Corollary F.3. This completes the proof.

### F.5 Coefficients in Proposition F.5

We define the coefficient $\mu_1^{\text{cond}}(N)$ and $\mu_2^{\text{cond}}(N)$ in Proposition F.5.

$$\mu_1^{\text{cond}}(N) = \frac{1}{N} \left[ \omega + \omega \mathbb{I}_N(\omega, 2) d_2(Q_Y || \tilde{P}_{Y|Z}(.|z)) + \frac{\omega^2}{N-1} d_2(Q_Y || \tilde{P}_{Y|Z}(.|z)) \right] \tag{385}$$

$$\mu_2^{\text{cond}}(N) = \frac{1}{N} \left[ \omega + \omega \mathbb{I}_N(\omega, 2) d_2(Q_Y || P_{Y|X}(.|x)) + \frac{\omega^2}{N-1} d_2(Q_Y || P_{Y|X}(.|x)) \right] \tag{386}$$

where we define $\mathbb{I}_N(\omega, i) \triangleq (2\mathbb{1}_{N>i} + \omega \mathbb{1}_{N=i})$ as in Proposition F.3.

### F.6 Proof of Proposition F.5

We will use the global index for the proposals $Y_1, ... Y_N \sim Q(.)$ instead of $Y_{1,1}, ... Y_{1,N}$ unless otherwise stated. For the communication version, we have:

$$\Pr(Y_{K_A} = Y_{K_B} | Y_{K_A} = y_1, X = x, Z = z) \tag{387}$$

$$\geq \Pr(K_A = K_B | Y_{K_A} = y_1, X = x, Z = z) \tag{388}$$

$$= \sum_{k=1}^{\infty} \Pr(K_A = K_B = k | Y_{K_A} = y_1, X = x, Z = z) \tag{389}$$

$$\geq \sum_{k=1}^{N} \Pr(K_A = K_B = k | Y_{K_A} = y_1, X = x, Z = z) \tag{390}$$

$$= N \Pr(K_A = K_B = 1 | Y_{K_A} = y_1, X = x, Z = z) \tag{391}$$

$$= N \Pr(K_{1,A} = K_{1,B} = 1, K_{2,A} = K_{2_B} = 1 | Y_{K_A} = y_1, X = x, Z = z) \tag{392}$$

Define:

$$\hat{Z}(P_{Y|X=x}, y_{1:N}) = \sum_{i=1}^{N} \frac{P_{Y|X}(y_i|x)}{Q_Y(y_i)}, \quad \hat{Z}(\tilde{P}_{Y|Z=z}, y_{1:N}) = \sum_{i=1}^{N} \frac{\tilde{P}_{Y|Z}(y_i|z)}{Q_Y(y_i)} \tag{393}$$

Now consider the following terms:

$$E_1 = \Pr(K_{1,A} = 1, K_{2,A} = 1 | Y_{K_A} = y_1, Y_{2:N} = y_{2:N}, X = x, Z = z)$$
$$\times P(Y_{2:N} = y_{2:N} | Y_{K_A} = y_1, X = x, Z = z) \tag{394}$$

$$= \frac{1}{P_{X,Y,Z}(x, y_1, z)} Q_Y(y_{1:N}) P_X(x) \Pr(K_{2,A} = 1 | Y_{1:N} = y_{1:N}, X = x)$$
$$\times \Pr(K_{1,A} = 1 | Y_{1:N} = y_{1:N}, X = x, K_{2,A} = 1) P_Z(z | Y_{1:N} = y_{1:N}, X = x, K_A = 1) \tag{395}$$

$$= \frac{1}{P_{X,Y,Z}(x, y_1, z)} Q_Y(y_{1:N}) P_X(x) \Pr(K_{2,A} = 1 | Y_{1:N} = y_{1:N}, X = x)$$
$$\times \Pr(K_{1,A} = 1 | Y_{1:N} = y_{1:N}, X = x, K_{2,A} = 1) P_{Z|X,Y}(z | X = x, Y = y_1) \tag{396}$$

$$= \frac{Q_Y(y_{1:N})}{P_{Y|X}(y_1 | x)} \Pr(K_{2,A} = 1 | Y_{1:N} = y_{1:N}, X = x)$$
$$\times \Pr(K_{1,A} = 1 | Y_{1:N} = y_{1:N}, X = x, K_{2,A} = 1) \tag{397}$$

$$= \frac{Q_Y(y_{1:N})}{P_{Y|X}(y_1 | x)} \frac{P_{Y|X}(y_1 | x) / Q_Y(y_1)}{\hat{Z}(P_{Y|X=x}, y_{2:N}) + \omega_x} \tag{398}$$

$$= \frac{Q_Y(y_{2:N})}{\hat{Z}(P_{Y|X=x}, y_{2:N}) + \omega_x} \tag{399}$$

and:

$$E_2 \tag{400}$$
$$= \Pr(K_{2,B} = 1 | K_A = 1, Y_{1:N} = y_{1:N}, X = x, Z = z) \tag{401}$$
$$= 1 - \Pr(K_{2,B} \neq 1 | K_A = 1, Y_{1:N} = y_{1:N}, X = x, Z = z) \tag{402}$$

$$= 1 - \Pr\left( \min_{j \neq 1} \frac{S_j}{\frac{\tilde{P}_{Y|Z}(y_j | z)}{\hat{Z}(\tilde{P}_{Y|Z=z}, y_{1:N})}} \leq \frac{S_1}{\frac{\tilde{P}_{Y|Z}(y_1 | z)}{\hat{Z}(\tilde{P}_{Y|Z=z}, y_{1:N})}} \middle| K_A = 1, Y_{1:N} = y_{1:N}, X = x, Z = z \right) \tag{403}$$

$$= 1 - \Pr\left( \min_{j \neq 1} \frac{S_j}{\frac{\tilde{P}_{Y|Z}(y_j | z)}{\hat{Z}(\tilde{P}_{Y|Z=z}, y_{1:N})}} \leq \frac{S_1}{\frac{\tilde{P}_{Y|Z}(y_1 | z)}{\hat{Z}(\tilde{P}_{Y|Z=z}, y_{1:N})}} \middle| K_A = 1, Y_{1:N} = y_{1:N}, X = x \right) \tag{404}$$

$$= 1 - \Pr\left( \min_{j \neq 1} \frac{S_j}{\frac{\tilde{P}_{Y|Z}(y_j | z)}{\hat{Z}(\tilde{P}_{Y|Z=z}, y_{1:N})}} \leq \frac{S_1}{\frac{\tilde{P}_{Y|Z}(y_1 | z)}{\hat{Z}(\tilde{P}_{Y|Z=z}, y_{1:N})}} \middle| K_{2,A} = 1, Y_{1:N} = y_{1:N}, X = x \right) \tag{405}$$

$$\geq \left( 1 + \frac{P_{Y|X}(y_1 | x)}{\tilde{P}_{Y|Z}(y_1 | z)} \frac{\hat{Z}(\tilde{P}_{Y|Z=z}, y_{1:N})}{\hat{Z}(P_{Y|X=x}, y_{1:N})} \right)^{-1}, \tag{406}$$

where (404) is due to the Markov condtion $Z - (X, Y) - W$, (405) is due to the fact that the uniform random variable $U$ is independent of $S_1^N$ and (406) is due to the conditional importance matching lemma [35]. We note the following events are equivalent:

$$\{K_A = 1, Y_{1:N} = y_{1:N}, X = x, Z = z, K_{2,B} = 1\} \tag{407}$$

$$\triangleq \left\{ U \leq \frac{\hat{Z}(P_{Y|X=x}, y_{1:N})}{\hat{Z}(P_{Y|X=x}, y_{2:N}) + \omega_x}, Y_{1:N} = y_{1:N}, X = x, Y_{K_A} = y_1, Z = z \right\} \tag{408}$$

$$\triangleq \mathcal{E} \cap \{Z = z\} \tag{409}$$

where $\mathcal{E} = \left\{ U \leq \frac{\hat{Z}(P_{Y|X=x}, y_{1:N})}{\hat{Z}(P_{Y|X=x}, y_{2:N}) + \omega_x}, Y_{1:N} = y_{1:N}, X = x, Y_{K_A} = y_1 \right\}$. Then, we have:

$$E_3 = \Pr(K_{1,B} = 1 | K_A = 1, Y_{1:N} = y_{1:N}, X = x, Z = z, K_{2,B} = 1) \tag{410}$$

$$= \Pr\left( U \leq \frac{\hat{Z}(\tilde{P}_{Y|Z=z}, Y_{1:N})}{\hat{Z}(\tilde{P}_{Y|Z=z}, Y_{2:N}) + \omega_z} \middle| \mathcal{E}, Z = z \right) \tag{411}$$

$$= \Pr\left( U \leq \frac{\hat{Z}(\tilde{P}_{Y|Z=z}, Y_{1:N})}{\hat{Z}(\tilde{P}_{Y|Z=z}, Y_{2:N}) + \omega_z} \middle| \mathcal{E} \right) \tag{412}$$

$$= \min\left( 1, \frac{\hat{Z}(\tilde{P}_{Y|Z=z}, Y_{1:N})}{\hat{Z}(\tilde{P}_{Y|Z=z}, Y_{2:N}) + \omega_z} \cdot \frac{\hat{Z}(P_{Y|X=x}, y_{2:N}) + \omega_x}{\hat{Z}(P_{Y|X=x}, y_{1:N})} \right), \tag{413}$$

where the second to last equality is due to the Markov condition $Z - (X, Y) - W$.

Combining all three terms $E_1, E_2, E_3$ and continue from step (392), we have:

$$\Pr(Y_{K_A} = Y_{K_B} | Y_{K_A} = y_1, X = x, Z = z) \tag{414}$$

$$\geq N \int \frac{Q_Y(y_{2:N})}{\hat{Z}(P_{Y|X=x}, y_{2:N}) + \omega_x} \left( 1 + \frac{P_{Y|X}(y_1|x)}{\tilde{P}_{Y|Z}(y_1|z)} \frac{\hat{Z}(\tilde{P}_{Y|Z=z}, y_{1:N})}{\hat{Z}(P_{Y|X=x}, y_{1:N})} \right)^{-1}$$

$$\times \min\left( 1, \frac{\hat{Z}(\tilde{P}_{Y|Z=z}, Y_{1:N})}{\hat{Z}(\tilde{P}_{Y|Z=z}, Y_{2:N}) + \omega_z} \cdot \frac{\hat{Z}(P_{Y|X=x}, y_{2:N}) + \omega_x}{\hat{Z}(P_{Y|X=x}, y_{1:N})} \right) dy_{2:N} \tag{415}$$

$$= N \int \frac{Q_Y(y_{2:N})}{\hat{Z}(P_{Y|X=x}, y_{1:N})} \left( 1 + \frac{P_{Y|X}(y_1|x)}{\tilde{P}_{Y|Z}(y_1|z)} \frac{\hat{Z}(\tilde{P}_{Y|Z=z}, y_{1:N})}{\hat{Z}(P_{Y|X=x}, y_{1:N})} \right)^{-1}$$

$$\times \min\left( \frac{\hat{Z}(P_{Y|X=x}, y_{1:N})}{\hat{Z}(P_{Y|X=x}, y_{2:N}) + \omega_x}, \frac{\hat{Z}(\tilde{P}_{Y|Z=z}, Y_{1:N})}{\hat{Z}(\tilde{P}_{Y|Z=z}, Y_{2:N}) + \omega_z} \right) dy_{2:N} \tag{416}$$

$$\geq \int \frac{N Q_Y(y_{2:N})}{\hat{Z}(P_{Y|X=x}, y_{1:N})} \left( 1 + \frac{P_{Y|X}(y_1|x)}{\tilde{P}_{Y|Z}(y_1|z)} \frac{\hat{Z}(\tilde{P}_{Y|Z=z}, y_{1:N})}{\hat{Z}(P_{Y|X=x}, y_{1:N})} \right)^{-1}$$

$$\times \left( \frac{\hat{Z}(P_{Y|X=x}, y_{1:N})}{\hat{Z}(P_{Y|X=x}, y_{2:N}) + \omega} \cdot \frac{\hat{Z}(\tilde{P}_{Y|Z=z}, Y_{1:N})}{\hat{Z}(\tilde{P}_{Y|Z=z}, Y_{2:N}) + \omega} \right) dy_{2:N} \tag{417}$$

with the last inequality follows the fact that $\omega > \max(\omega_x, \omega_z)$. The rest of the proof follows similar steps as in the proof of Proposition F.4. This completes the proof.

# G ERS Matching with Batch Communication

**Setup.** We first describe the setup in the case where the selected batch index is communicated from the encoder to the decoder. The main difference between this and the setup in Section 5.3 is that the decoder (party $B$) will use the Gumbel-Max selection method instead of the ERS one, since it knows which batch the encoder index belongs to. Furthermore, we note this scheme requires a noiseless channel between the encoder and decoder, which is available in the distributed compression scenario. Similarly to Section 3.2.2, let $(X, Y, Z) \in \mathcal{X} \times \mathcal{Y} \times \mathcal{Z}$ with a joint distribution $P_{X,Y,Z}$. We use the same common randomness $W$ as in Section 5.3, with the proposal distribution $Q_Y$ requiring that the bounding condition hold for the tuple $(P_{Y|X=x}, Q_Y)$. The protocol is as follows:

1. The encoder receives the input $X = x \sim P_X$ and selects its value using ERS procedure:

$$K_A = \text{ERS}(W; P_{Y|X=x}, Q_Y), \tag{418}$$

   and sends the batch index $K_{1,A}$ to the decoder. It then sets $Y_A = Y_{K_A}$

2. Given $X = x, Y_A = y$, we generate $Z = z \sim P_{Z|X,Y}(.|x,y)$ and note that the Markov chain $Z - (X, Y_A) - W$ holds.

3. The decoder receives the batch index $K_{1,A}$ and $Z = z$ will use the Gumbel Max process to queries a sample from the common randomness $W$:

$$K_{1,B} = K_{1,A} \quad K_{2,B} = \text{Gumbel}(B_{K_{1,A}}; \tilde{P}_{Y|Z=z}, Q_Y) \quad K_B = (K_{1,B} - 1)N + K_{2,B},$$

   and output $Y_B = Y_{K_B}$. The procedure $\text{Gumbel}(.)$ corresponds to Step 1,2 in Algoirhm 1.

Given the above setup, we have the following bound on the matching event $\{Y_A = Y_B\}$:

**Proposition G.1.** *Let $K_A, K_B, P_{Y|X}(.|X = x)$ and $\tilde{P}_{Y|Z}(.|z)$ defined above and set $P_Y^A = P_{Y|X=x}, P_Y^B = \tilde{P}_{Y|Z=z}$. For $N \geq 2$, we have:*

$$\Pr(Y_A = Y_B | Y_A = y, X = x, Z = z) \geq \left( 1 + \mu_1'(N) + \frac{P_Y^A(y)}{P_Y^B(y)} \left( 1 + \mu_2'(N) \right) \right)^{-1}, \tag{419}$$

*where $\mu_1'(N)$ and $\mu_2'(N)$ are defined as in Appendix G.1 and we note that $\mu_1'(N), \mu_2'(N) \to 0$ as $N \to \infty$ with rate $N^{-1}$ under mild assumptions on the distributions $P_{Y|X}(y|x)$ and $Q_Y(.)$.*

*Proof.* See Appendix G.2. $\qquad\qquad\qquad\qquad\qquad\qquad\qquad\qquad\qquad\qquad\qquad\qquad\qquad\qquad\square$

## G.1 Coefficients in Proposition G.1

We define the coefficient $\mu_1'(N)$ and $\mu_2'(N)$ in Proposition G.1.

$$\mu_1'(N) = \frac{3\omega}{N} \tag{420}$$

$$\mu_2'(N) = \frac{\omega}{N} \mathbb{I}_N(\omega, 2) d_2(Q_Y || P_{Y|X=x}) \tag{421}$$

where we define $\mathbb{I}_N(\omega, i) \triangleq (2\mathbb{1}_{N>i} + \omega\mathbb{1}_{N=i})$ as in Proposition F.3 and $\omega = \max_y \frac{P_{Y|X}(y|x)}{Q_Y(y)}$.

## G.2 Proof of Proposition G.1

We now formally prove the bound Proposition G.1. First, we define:

$$\hat{Z}(P_{Y|X=x}, y_{1:N}) = \sum_{i=1}^{N} \frac{P_{Y|X}(y_i|x)}{Q_Y(y_i)}, \quad \hat{Z}(\tilde{P}_{Y|Z=z}, y_{1:N}) = \sum_{i=1}^{N} \frac{\tilde{P}_{Y|Z}(y_i|z)}{Q_Y(y_i)} \tag{422}$$

Recall that $K_{2,A}$ is the local index within the selected batch by party $A$ and $Y_{K_{1,A},1:N}$ are the samples within the selected batch, we have:

$$\Pr(Y_A = Y_B | Y_A = y_1, X = x, Z = z) \tag{423}$$

$$= \Pr(Y_{K_A} = Y_{K_B} | Y_{K_A} = y_1, X = x, Z = z) \tag{424}$$

$$\geq \Pr(K_{2,A} = K_{2,B} | Y_{K_A} = y_1, X = x, Z = z) \tag{425}$$

$$= \sum_{i=1}^{N} \Pr(K_{2,A} = K_{2,B} = i | Y_{K_A} = y_1, X = x, Z = z) \tag{426}$$

$$= N \Pr(K_{2,A} = K_{2,B} = 1 | Y_{K_A} = y_1, X = x, Z = z) \quad \text{(Due to Symmetry)} \tag{427}$$

$$= N \Pr(K_{2,A} = K_{2,B} | Y_{K_A} = y_1, K_{2,A} = 1, X = x, Z = z)$$
$$\times \Pr(K_{2,A} = 1 | Y_{K_A} = y_1, X = x, Z = z) \tag{428}$$

$$= \Pr(K_{2,A} = K_{2,B} | Y_{K_A} = y_1, K_{2,A} = 1, X = x, Z = z) \tag{429}$$

$$= \int_{-\infty}^{\infty} P(Y_{K_{1,A},2:N} = y_{2:N} | Y_{K_A} = y_1, K_{2,A} = 1, X = x, Z = z)$$
$$\times \Pr(K_{2,A} = K_{2,B} | Y_{K_A} = y_1, K_{2,A} = 1, Y_{K_{1,A},2:N} = y_{2:N}, X = x, Z = z) dy_{2:N}, \tag{430}$$

where (429) is due to $\Pr(K_{2,A} = 1 | Y_{K_A} = y_1, X = x, Z = z) = N^{-1}$. Let $Y_{1:N} \sim Q$ are $N$ i.i.d. proposal samples, then $\{Y_{K_A,1:N} = y_{1:N}\} = \{Y_{1:N} = y_{1:N}, A \text{ accepts } Y_{1:N}\}$ and we have:

$$\Pr(K_{2,A} = K_{2,B} | Y_{K_A} = y_1, K_{2,A} = 1, Y_{K_{1,A},2:N} = y_{2:N}, X = x, Z = z) \tag{431}$$

$$= 1 - \Pr(K_{2,B} \neq 1 | Y_{K_{1,A},1:N} = y_{1:N}, K_{2,A} = 1, Y_{K_A} = y_1, X = x, Z = z)$$

$$= 1 - \Pr(\min_{j \neq 1} \frac{S_j}{\frac{\tilde{P}_{Y|Z}(y_j|z)}{\hat{Z}(\tilde{P}_{Y|Z=z}, y_{1:N})}} \leq \frac{S_1}{\frac{\tilde{P}_{Y|Z}(y_1|z)}{\hat{Z}(\tilde{P}_{Y|Z=z}, y_{1:N})}} | Y_{1:N} = y_{1:N}, A \text{ selects 1st index},$$
$$A \text{ accepts } Y_{1:N}, Y_{K_A} = y_1, X = x, Z = z) \tag{432}$$

$$= 1 - \Pr(\min_{j \neq 1} \frac{S_j}{\frac{\tilde{P}_{Y|Z}(y_j|z)}{\hat{Z}(\tilde{P}_{Y|Z=z}, y_{1:N})}} \leq \frac{S_1}{\frac{\tilde{P}_{Y|Z}(y_1|z)}{\hat{Z}(\tilde{P}_{Y|Z=z}, y_{1:N})}} | Y_{1:N} = y_{1:N}, A \text{ selects 1st index},$$
$$A \text{ accepts } Y_{1:N}, Y_{K_A} = y_1, X = x) \tag{433}$$

$$= 1 - \Pr\left( \min_{j \neq 1} \frac{S_j}{\frac{\tilde{P}_{Y|Z}(y_j|z)}{\hat{Z}(\tilde{P}_{Y|Z=z}, y_{1:N})}} \leq \frac{S_1}{\frac{\tilde{P}_{Y|Z}(y_1|z)}{\hat{Z}(\tilde{P}_{Y|Z=z}, y_{1:N})}} | Y_{1:N} = y_{1:N}, A \text{ selects 1st index}, X = x \right) \tag{434}$$

$$\geq \left( 1 + \frac{P_{Y|X}(y_1|x)}{\tilde{P}_{Y|Z}(y_1|z)} \frac{\hat{Z}(\tilde{P}_{Y|Z=z}, y_{1:N})}{\hat{Z}(P_{Y|X=x}, y_{1:N})} \right), \tag{435}$$

where (433) is due to Markov property $Z - (X, Y) - W$, i.e. $Z$ has no effects on the statistics of the exponential random variables. Line (434) is due to the fact that conditioning on $A$ selected the 1st index, whether $A$ selects $Y_{1:N}$ or not depends only on $U$. The final inequality is due to conditional matching lemma from [35].

Recall that $\omega = \max_y \frac{P_{Y|X}(y|x)}{Q_Y(y)}$, we have:

$$P(Y_{K_{1,A},2:N} = y_{2:N} | Y_{K_A} = y_1, K_{2,A} = 1, X = x, Z = z) \tag{436}$$

$$= P(Y_{K_{1,A},2:N} = y_{2:N} | Y_{K_A} = y_1, K_{2,A} = 1, X = x) \tag{437}$$

$$= \frac{\bar{P}_{Y,K_{2,A}|X}(y_{1:N}, 1|x)}{P_{Y|X}(y_1|x) N^{-1}} \tag{438}$$

$$= \frac{N Q_Y(y_{2:N})}{\Delta_{P_{Y|X=x}}(\hat{Z}(P_{Y|X=x}, y_{2:N}) + \omega)} \tag{439}$$

where $\bar{P}_{Y,K_{2,A}|X}(y_{1:N}, 1|x)$ is the ERS target distribution (151) where we use $P_{Y|X}(.|x)$ as the target distribution and $\Delta_{P_{Y|X=x}} < 1$ is the normalized constant. We now shorthand $P_Y^A \triangleq P_{Y|X=x}$,

$P_Y^B \triangleq \tilde{P}(Y|Z=z)$ and $\Delta_{P_Y^A} \triangleq \Delta_{P_{Y|X=x}}$, and combining the two expressions, we have:

$$\Pr(Y_A = Y_B | Y_A = y_1, X = x, Z = z) \tag{440}$$

$$\geq \mathbb{E}_{Y_{2:N} \sim Q_Y} \left[ \frac{N}{(\hat{Z}(P_Y^A, y_{2:N}) + \omega)\Delta_{P_Y^A} \left(1 + \frac{P_Y^A(y_1)}{P_Y^B(y1)} \frac{\hat{Z}(P_Y^B, y_{1:N})}{\hat{Z}(P_Y^A, y_{1:N})}\right)} \right] \tag{441}$$

$$\geq \mathbb{E}_{Y_{2:N} \sim Q_Y} \left[ \frac{N}{(\hat{Z}(P_Y^A, y_{2:N}) + \omega) \left(1 + \frac{P_Y^A(y_1)}{P_Y^B(y1)} \frac{\hat{Z}(P_Y^B, y_{1:N})}{\hat{Z}(P_Y^A, y_{1:N})}\right)} \right] \quad (\text{Since } \Delta_{P_Y^A} \leq 1) \tag{442}$$

$$\geq \left( \mathbb{E}_{Y_{2:N} \sim Q_Y} \left[ \frac{(\hat{Z}(P_Y^A, y_{2:N}) + \omega)}{N} \left(1 + \frac{P_Y^A(y_1)}{P_Y^B(y1)} \frac{\hat{Z}(P_Y^B, y_{1:N})}{\hat{Z}(P_Y^A, y_{1:N})}\right) \right] \right)^{-1} \quad (\text{By Jensen's Inequality}) \tag{443}$$

Since:

$$\mathbb{E}_{Y_{2:N} \sim Q_Y} \left[ \frac{\hat{Z}(P_Y^A, y_{2:N}) + \omega}{N} \right] \leq \frac{N-1}{N} + \frac{\omega}{N} \tag{444}$$

and:

$$\mathbb{E}_{Y_{2:N} \sim Q_Y} \left[ \left( \frac{\hat{Z}(P_Y^A, y_{2:N}) + \omega}{N} \right) \frac{\hat{Z}(P_Y^B, y_{1:N})}{\hat{Z}(P_Y^A, y_{1:N})} \right] \tag{445}$$

$$= \mathbb{E}_{Y_{2:N} \sim Q_Y} \left[ \frac{\hat{Z}(P_Y^A, y_{2:N})}{N} \frac{\hat{Z}(P_Y^B, y_{1:N})}{\hat{Z}(P_Y^A, y_{1:N})} + \frac{\omega}{N} \frac{\hat{Z}(P_Y^B, y_{1:N})}{\hat{Z}(P_Y^A, y_{1:N})} \right] \tag{446}$$

$$\leq \frac{N-1}{N} + \frac{P_Y^B(y_1)/Q_Y(y_1)}{N} + \frac{\omega}{N} \mathbb{E}_{Y_{2:N} \sim Q_Y} \left[ \frac{\hat{Z}(P_Y^B, y_{1:N})}{\hat{Z}(P_Y^A, y_{1:N})} \right] \tag{447}$$

where we have:

$$\mathbb{E}_{Y_{2:N} \sim Q_Y} \left[ \frac{\hat{Z}(P_Y^B, y_{1:N})}{\hat{Z}(P_Y^A, y_{1:N})} \right] = \mathbb{E}_{Y_{2:N} \sim Q_Y} \left[ \frac{P_Y^B(y_1)/Q_Y(y_1)}{\hat{Z}(P_Y^A, y_{1:N})} + \frac{\hat{Z}(P_Y^B, y_{2:N})}{\hat{Z}(P_Y^A, y_{1:N})} \right] \tag{448}$$

$$\leq \mathbb{E}_{Y_{2:N} \sim Q_Y} \left[ \frac{P_Y^B(y_1)/Q_Y(y_1)}{P_Y^A(y_1)/Q_Y(y_1)} \right] + \mathbb{E}_{Y_{2:N} \sim Q_Y} \left[ \frac{\hat{Z}(P_Y^B, y_{2:N})}{\hat{Z}(P_Y^A, y_{2:N})} \right] \tag{449}$$

$$\leq \frac{P_Y^B(y_1)}{P_Y^A(y_1)} + \mathbb{I}_N(\omega, 2) d_2(Q_Y || P_Y^A) \tag{450}$$

Then, combining (450) into (447), then combine with (444) into the term (443), we have:

$$\Pr(Y_A = Y_B | Y_A = y_1, X = x, Z = z) \tag{451}$$

$$\geq \left(1 + \frac{\omega}{N} + \frac{P_Y^A(y_1)}{P_Y^B(y_1)} \left(\frac{N-1}{N} + \frac{P_Y^B(y_1)/Q_Y(y_1)}{N} + \frac{\omega}{N} \left(\frac{P_Y^B(y_1)}{P_Y^A(y_1)} + \mathbb{I}_N(\omega, 2) d_2(Q_Y || P_Y^A)\right)\right)\right)^{-1} \tag{452}$$

$$= \left(1 + \frac{\omega}{N} + \frac{P_Y^A(y_1)}{P_Y^B(y_1)} \left(\frac{N-1}{N} + \frac{P_Y^B(y_1)/Q_Y(y_1)}{N} + \frac{\omega}{N} \left(\frac{P_Y^B(y_1)}{P_Y^A(y_1)} + \mathbb{I}_N(\omega, 2) d_2(Q_Y || P_Y^A)\right)\right)\right)^{-1} \tag{453}$$

$$\geq \left(1 + \frac{3\omega}{N} + \frac{P_Y^A(y_1)}{P_Y^B(y_1)} \left(1 + \frac{\omega}{N} \mathbb{I}_N(\omega, 2) d_2(Q_Y || P_Y^A)\right)\right)^{-1} \tag{454}$$

$$= \left(1 + \mu_1'(N) + \frac{P_Y^A(y_1)}{P_Y^B(y_1)} (1 + \mu_2'(N))\right)^{-1}, \tag{455}$$

where we repeatedly use the fact that $P_Y^A(y)/Q_Y(y) \leq \omega$. This completes the proof.

# H Proof of Proposition 5.6

---
**Algorithm 2:** Wyner-Ziv Distributed Compression Protocol

---
**Encoder:** Receives $X = x$ and $W$, performs:

1. Select $K_A = \mathrm{ERS}(W; P_{Y'|X=x}, Q_{Y'})$;    2. Sends $(K_{1,A}, V_{K_A})$ to the decoder.

**1**   **Decoder:** Receives $Z = (V_{K_A}, K_{1,A}, X')$ and $W$, performs:

1. Keep batch $K_{1,A}$;    2. Remove all $j$ where $V_{K_{1,A},j} \neq V_{K_A}$;    3. Select $K_B$ with $P_{Y'|X'=x'}$.

---

**Main Proof.** We remind the protocol in Algorithm 2. The encoder and decoder's target distribution for this case are:

$$P_Y^A(y, v) = P_{Y|X}(y|x) P_V(v) \quad P_Y^B(y, v) = P_{Y|X'}(y|x) \mathbb{I}_V(v) \tag{456}$$

For a sufficient large batch size $N$ and apply Proposition G.1, we have:

$$\Pr(Y'_{K_A} \neq Y'_{K_B} | (Y'_{K_A}, V_{K_A}) = (y', v), X = x, Z = (x', v)) \tag{457}$$

$$= \Pr((Y'_{K_A}, V_{K_A}) \neq (Y'_{K_B}, V_{K_B}) | (Y'_{K_A}, V_{K_A}) = (y', v), X = x, Z = (x', v)) \tag{458}$$

$$\leq 1 - \left(1 + \epsilon + \frac{P_{Y'|X}(y'|x) P_V(v)}{P_{Y'|X'}(y'|x') \mathbb{I}_v(v)} (1 + \epsilon)\right)^{-1} \tag{459}$$

$$\leq 1 - \left(1 + \epsilon + \mathcal{V}^{-1}(1 + \epsilon) \frac{P_{Y'|X}(y'|x)}{P_{Y'|X'}(y'|x')}\right)^{-1} \tag{460}$$

$$= 1 - \left(1 + \epsilon + \mathcal{V}^{-1}(1 + \epsilon) \frac{P_{Y'|X}(y'|x)}{P_Y'(y')} \frac{P_Y'(y')}{P_{Y'|X'}(y'|x')}\right)^{-1} \tag{461}$$

$$= 1 - \left(1 + \epsilon + \mathcal{V}^{-1}(1 + \epsilon) 2^{i_{Y';X}(y';x) - i_{Y';X'}(y';x')}\right)^{-1} \tag{462}$$

Finally, taking the expectation of both sides yields the final result.

**Coding Cost.** In terms of the bound on $r$, recall the following bound on batch acceptance probability:

$$\Delta = \mathbb{E}_{Y_{1:N} \sim P_Y(.)} \left[\frac{N}{\bar{Z}(1, Y'_{1:N})}\right] \geq \frac{N}{\mathbb{E}_{Y'_{1:N} \sim P_Y(.)}[\bar{Z}(1)]} = \frac{N}{N - 1 + \omega} \tag{463}$$

Here for $N = \omega$, we have $\Delta > \frac{1}{2}$ and thus the chunk size $L = \lfloor \Delta^{-1} \rfloor$ in the ERS coding scheme is 1 and thus do not need to send $\hat{K}_1$. Using the fact that $\mathbb{E}[\log L] \leq 1$, we have $r \leq H[L] + 1 = 4$bits by entropy coding with Zipf distribution [28].

**Compressing Multiple Samples.** When compressing $n$ samples jointly, let the rate per sample (without the overhead for batch communication) be $r'$ where $\log(V) = nr'$ consider the following approximation:

$$\sum_{i=1}^{n} i(y'_i; x_i) - i(y'_i; x'_i) \approx nI(X; Y'|X'),$$

Then we have:

$$2^{\sum_{i=1}^{n} [i(y'_i; x_i) - i(y'_i; x'_i)] - \log(V)} \approx 2^{nI(X; Y'|X') - \log(V)} \tag{464}$$

$$= 2^{n(I(X; Y'|X') - r')}, \tag{465}$$

and thus, if $r' > I(X; Y'|X')$, by increasing $n$ we reduces the mismatching probability while maintaining the compression rate per sample. We visualize this in the experimental results with $N = 2^{19}$ in Figure 9.

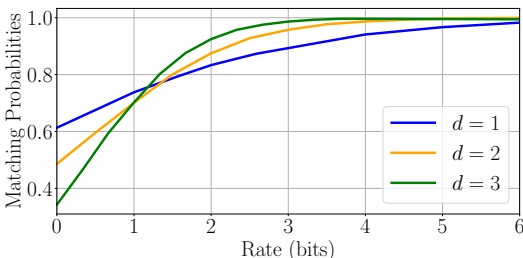

Figure 9: Matching Probabilities with $N = 2^{19}$ and jointly compressing $1, 2, 3$ i.i.d. samples respectively. Target distortion $\sigma^2_{Y'|X} = 0.008$ for every samples.

# I  Feedback Scheme

In distributed compression, decoding errors can lead to significant average reconstruction distortion. To address this, feedback communication from the decoder can be employed to correct errors and enhance rate-distortion performance, as proposed in [35]. The feedback mechanism is identical for both ERS and IML, except that ERS additionally transmits the batch index to the decoder.

We recall that $K_{1,A}$ and $K_{2,A}$ denote the batch index and local index, respectively, of samples selected by party $A$ through the ERS sample selection. On the other hand, party $B$ uses Gumbel-Max selection process to output its selected local index $K_{2,B}$ within the $K_{1,A}$ batch, then the ERS process can be described as follows:

1. *Index Selection.* After transmitting the batch index $K_{1,A}$, the encoder sends the $\log_2(\mathcal{V})$ least significant bits (LSB) of the selected index $K_{2,A}$ to the decoder.

2. *Decoding and Feedback.* The decoder outputs $K_{2,B}$ and sends the $\log_2(N/\mathcal{V})$ most significant bits (MSB) of $K_B$ to the encoder.

3. *Re-transmission.* Based on the received MSB feedback, if the index is correct, the encoder responds with an acknowledgment bit, say $1$. Otherwise, it sends $0$ along with the MSB of its selection to the decoder.

We note that, in this context, using LSB instead of random bits in step 1 does not yield a noticeable difference in performance. For the rate-distortion analysis, the rate is computed based on the total length of messages transmitted during index selection and re-transmission, including any acknowledgment messages. However, the rate of the feedback message is excluded from this calculation, which can be justified in scenarios with asymmetric communication costs in the forward and reverse directions, such as in wireless channels.

# J  Neural Contrastive Estimator

In our ERS scheme, the selection rule requires estimating the following ratio at the decoder side:

$$\tilde{K}_B = \underset{1 \leq k \leq N}{\operatorname{argmin}} \frac{S_{ik}}{\frac{P_{Y|X'}(y|x')\mathbb{I}_V(v)}{Q_Y(Y_{ik})\mathrm{V}^{-1}}}, \quad \text{where } i = K_{1,A}, \tag{466}$$

where the normalization term can be ignored as it is the same for every sample in the batch $K_{1,A}$. Our goal is to learn the ratio $P_{Y|X'}(Y_{ik}|x')/Q_Y(Y_{ik})$ from data. In particular, we can access the data samples from the joint distribution $P_{X,Y,X'}$.

To this end, we construct a binary neural classifier $h'(y, x') = \frac{1}{1+\exp[-h(y,x')]}$ which classifies if the input $(y, x')$ is distributed according to the marginal distribution $Q_Y(.) \times P_{X'}(.)$ (positive samples) or the joint $P_{Y,X'}$ (negative samples). Once converged, we can use the logits value $h(y, x')$ to compute the log of the ratio of interest [22]. In particular:

$$h(y, x') \approx -\log \frac{P_{Y|X'}(Y_{ik}|x')}{Q_Y(y)} \tag{467}$$

This allows us to estimate the ratio without needing to obtain the exact ratio's value. Finally, to generate the positive samples, we simply generate $Y \sim Q_Y(.)$ and get a random $X'$ from the training data. For negative samples, we generate the data according to the Markov sequence $X' - X - Y$. The ratio between the two labels should be the same.

## K   Distributed Compression with MNIST

### K.1   Training Details

$\beta$-**VAE Architecture.** We adopt a setup similar to [35]. Our neural encoder-decoder model comprises an encoder network $y = \mathrm{enc}(x)$, a projection network $\mathrm{proj}(x')$, and a decoder network $\hat{x} = \mathrm{dec}(y, \mathrm{proj}(x'))$, as detailed in Table 1. The encoder network converts an image into two vectors of size 3 (total 6D output), with the first vector representing the output mean $\mu(x)$ and the second representing the output variance $\sigma^2(x)$. Here, we define $p_{Y|X}(.|x) = \mathcal{N}(\mu(x), \sigma^2(x))$ and use the prior distribution $p_Y(.) = \mathcal{N}(0,1)$. At the decoder side (party $B$), the projection network first maps the side information image $X'$ to a 128-dimensional vector, which is then combined with a 3-dimensional vector from the encoder. This concatenated vector is input to the decoder network, producing a reconstructed output of size $28 \times 28$.

Table 1: Architecture of the encoder, projection network, and decoder for distributed MNIST image compression. Convolutional and transposed convolutional layers are denoted as "conv" and "upconv," respectively, with specifications for the number of filters, kernel size, stride, and padding. For "upconv," an additional output padding parameter is included.

| **(a)**Encoder | **(b)**Projection Network | **(c)**Decoder Network |
|---|---|---|
| Input $28 \times 28 \times 1$ | Input $14 \times 14 \times 1$ | Input-(3+128) |
| conv (128:3:1:1), ReLU | conv (32:3:1:1), ReLU | Linear-$(132, 512)$, ReLU |
| conv (128:3:2:1), ReLU | conv (64:3:2:1), ReLU | upconv (64:3:2:1:1), ReLU |
| conv (128:3:2:1), ReLU | conv (128:3:2:1), ReLU | upconv (32:3:2:1:1), ReLU |
| Flatten | Flatten | upconv (1:3:1:1), Tanh |
| Linear (6272, 512), ReLU | Linear (2048, 512), ReLU | |
| Linear (512, 6) | Linear (512, 128) | |

**Loss Function** We train our $\beta$-VAE network by optimizing the following rate-distortion loss function:

$$\mathcal{L} = \beta(X - \hat{X})^2 + E_X[D_{\mathrm{KL}}(p_{Y|X}(.|v)||p_Y(.))] \tag{468}$$

where we vary $\beta$ for different rate-distortion tradeoff.Each model is trained for 30 epochs on an NVIDIA RTX A4500, requiring approximately 30 minutes per model. We use random horizontal flipping and random rotation within the range $\pm 15^{\circ}$. We use the following values of $\beta \in \{0.225, 0.28, 0.31, 0.4\}$ that corresponds to the target distortions $\{6.6, 6.3, 6.1, 5.8\} \times 10^{-2}$ in Figure 6.

**Neural Contrastive Estimator Network.** The neural estimator network comprises two subnetworks. The first subnetwork projects the side information into a 128-dimensional embedding. The second subnetwork combines this 128D embedding with a 4D embedding, derived from either $p_{Y|X}$ or the prior $p_Y$, and outputs the probability that $X', Y$ originate from the joint or marginal distributions. The model is trained for 100 epochs.

Table 2: Neural Estimator Networks for Distributed Image Compression.

| **(a)**Projection Network | **(b)** Combine and Classify |
|---|---|
| Input $14 \times 14 \times 1$ | Input $128 + 3$ |
| conv (32:3:1:1), ReLU | Linear (132, 128), l-ReLU |
| conv (64:3:2:1), ReLU | Linear (128,128), l-ReLU |
| conv (128:3:2:1), ReLU | Linear (128,128), l-ReLU |
| Flatten | Linear (128, 1) |
| Linear (2048, 512), ReLU | |
| Linear (512, 128) | |

| Rate (bits/image) | Model | Embedding MSE | Pixel MSE |
|---|---|---|---|
| 8.75 | Gaussian Regressor | 0.7300 | 0.0696 |
| | ERS (NCE) | **0.6024** | **0.0683** |
| 9.60 | Gaussian Regressor | 0.5260 | 0.0660 |
| | ERS (NCE) | **0.4807** | **0.0647** |
| 10.10 | Gaussian Regressor | 0.4310 | 0.0638 |
| | ERS (NCE) | **0.3616** | **0.0623** |
| 10.60 | Gaussian Regressor | 0.3600 | 0.0626 |
| | ERS (NCE) | **0.2930** | **0.0606** |

Table 3: Comparison of Gaussian regressor vs. ERS (with NCE) under different rates. Distortion is reported as MSE (lower is better).

| $\log \mathcal{V}$ | $N$ | $N^*$ | Target dB |
|---|---|---|---|
| 9.6 | 0.6e6 | 1.0e6 | $-21.5$dB |
| 10.6 | 0.7e6 | 1.1e6 | $-22$dB |
| 11.6 | 0.8e6 | 1.5e6 | $-22.5$dB |
| 12.6 | 1.04 | 1.6e6 | $-23$dB |

Table 4: Details for ERS Gaussian Experiment in Figure 5 (right)

## K.2 Decoder Estimation with Neural Contrastive Estimator: Gaussian Assumption for PML

In our experiment, the decoder uses a NCE to directly estimate the likelihood ratio without assuming a specific form—such as Gaussian—for the posterior. Consequently, computing the local constant needed for PML becomes intractable. If we instead assume a Gaussian form, PML becomes feasible; however, our experimental results show that this assumption introduces a mismatch that worsens the rate–distortion tradeoff.

Table 3 reports the distortion results across different rates. Here, *Embedding MSE* refers to the error measured with respect to the encoder's neural network output, while *Pixel MSE* captures the distortion at the image level. Lower values indicate better performance.

## L    Wyner-Ziv Gaussian Experiment

In Figure 5 (left), the batch size of ERS are $N \in \{2^{19}, 2^{20}\}$ respectively for the average number of proposals $N^* \in \{1.1, 1.6\} \times 10^6$. For Figure 5 (right), details for ERS are shown in Table 4.

## M    Additional Experiment on CIFAR-10 Dataset

We conduct experiments on the CIFAR-10 dataset and compare our method with implicit neural representations [20], the quantization approach [2], and the IML [35]. We use Mean Squared Error (MSE) as the distortion metric across all schemes, where lower values indicate better performance. Our approach consistently achieves lower distortion by leveraging side information within the encoding scheme.

Table 5: Comparison of distortion (MSE) on CIFAR-10 at different rates (bits/image). Lower is better.

| Rate (bits/image) | Ballé et al. [2] | RECOMBINER [20] | IML [35] | ERS Ours |
|---|---|---|---|---|
| $\sim 9$ | 0.0972 | 0.0968 | 0.0711 | **0.0703** |
| $\sim 10$ | 0.0915 | 0.0912 | 0.0668 | **0.0659** |
| $\sim 11$ | 0.0802 | 0.0810 | 0.0621 | **0.0606** |

