# OpenReview forum: "Channel Simulation and Distributed Compression with Ensemble Rejection Sampling"
_NeurIPS.cc/2025/Conference — NeurIPS 2025 poster_

### Official Review · Reviewer_2a3z · 2025-06-06

**Clarity:** 2
**Significance:** 2
**Originality:** 3
**Rating:** 4
**Confidence:** 2

**Summary:**

This paper proposes Ensemble Rejection Sampling (ERS) for channel simulation and distributed compression. Theoretically, ERS achieves near-optimal coding rates matching for channel simulation and distributed matching probabilities competitive with the Poisson Matching Lemma (PML) as batch size increases. Experiments on Gaussian sources and MNIST validate ERS’s advantages over importance sampling (IML) in bias reduction and distortion control.

**Questions:**

refer to Strengths And Weaknesses.

**Ethical Concerns:**

["NO or VERY MINOR ethics concerns only"]

**Final Justification:**

My concerns have been addressed. I will maintain my initial score.

**Limitations:**

limitations and potential negative societal impact are not discussed.

**Quality:**

3

**Strengths And Weaknesses:**

**Strengths:**

1. The paper provides significant theoretical advancements, including a new coding scheme for RS that achieves near-optimal rates and extends prior discrete results to continuous settings.

2. Synthetic experiments show ERS outperforms IML in low-distortion regimes, and MNIST confirms lower MSE with unbiased samples.

**Weaknesses:**

1. Only simple datasets are tested and no high-dimensional or real-world benchmarks are adopted. The scaliability of ERS is unclear. Moreover, comparision to SOTA compression methods is missed.

2. While the ERS matching lemma is new, core mechanisms, such as batching and Gumbel tricks, build heavily on prior IS/RS work.

---

> ### Author Rebuttal · Authors · 2025-07-30
>
> Thank you for your review and encouraging comments. We're glad you appreciated the theoretical advancements of our ERS and RS-based coding scheme and the empirical benefits of ERS in both synthetic and real-data experiments. Please find your concerns answered below.
>
> **(1) Extra Experiments and Comparison to other Compression Method**  We conduct experiments on the CIFAR-10 dataset, comparing our ERS approach against the quantization baseline [A1], the implicit neural representation method [A2] (which is one of the recent state-of-the-art image compression method), and the Importance Matching Lemma (IML) [33]. We use Mean Squared Error (MSE) as the distortion metric across all schemes, where lower values indicate better performance. By effectively leveraging side information in the encoding process, ERS  consistently achieve lower distortion across all settings.
>
> | Rate (bits/image) | Balle (2018) [A1] | RECOMBINER (He 2023) [A2]| IML (Phan 2024) [33] | ERS (Ours) |
> |-------------------|--------------|------------------------|------------------|------------|
> | ~9                | 0.0972       | 0.0968                 | 0.0711           | **0.0703** |
> | ~10               | 0.0915       | 0.0912                 | 0.0668           | **0.0659** |
> | ~11               | 0.0802       | 0.0810                 | 0.0621           | **0.0606** |
>
> **(2) Technical Contributions**
>
> The core technical contribution of our work is the development of new encoding schemes within the RS/ERS framework, which offer several key advantages over existing methods:
> - Compared to the Importance Matching Lemma (IML): ERS produces exact output samples rather than approximate ones, resulting in better performance, particularly in low-distortion regimes (see Section 6).
> - Compared to Greedy Rejection Sampling (GRS): ERS achieves substantially higher distributed matching probabilities.
>
> - Compared to the original Poisson Matching Lemma (PML): Our ERS-based framework, integrated with neural networks, overcomes the decoder-side termination issue that remains a challenge in the original PML design for high-dimensional data.
>
>
> Besides the development of new coding schemes, our contribution also includes substantial technical novelty in the analysis, with each core proof spanning over 10 pages. In particular, the achievability proof for channel simulation with ERS at an arbitrary batch size $N$ involves bounding a complex probabilistic integral arising from self-normalized importance sampling, without assuming any specific form for the target or proposal distributions [A3]. This derivation notably relies on a symmetry-based argument, rather than standard concentration tools such as Hoeffding’s or log-Sobolev inequalities, allowing us to establish an upper bound that holds uniformly over all values of $N$. The challenges of handling self-normalized importance sampling terms also arise in our analysis of the matching lemmas. Specifically, proving the bounds in Propositions 5.4 and 5.6 requires substantial analytical effort, involving intricate combinations of probabilistic reasoning as well as set-theoretic arguments.
>
>
> **(3) Limitations and Impact** We will include a more detailed discussion of the limitations and potential impact of our work in the main paper. Channel simulation techniques are commonly used in privacy-preserving applications, such as secure data sharing and communication, which contributes positively to societal impact.
>
>
> Let us know if you have additional concerns and we will address them.
>
> **References**
>
> [A1] Ballé, Johannes, et al. "Variational image compression with a scale hyperprior." International Conference on Learning Representations. 2018.
>
> [A2] He, Jiajun, et al. "RECOMBINER: Robust and Enhanced Compression with Bayesian Implicit Neural Representations." The Twelfth International Conference on Learning Representations.
>
> [A3] Agapiou, Sergios, et al. "Importance sampling: Intrinsic dimension and computational cost." Statistical Science (2017): 405-431.

---

> > ### Comment · Reviewer_2a3z · 2025-08-06
> >
> > Thank you for your response. My concerns have been addressed. I will think carefully and give my final rating.

---

> > > ### Author Response · Authors · 2025-08-06
> > >
> > > Thank you again for your feedback and assessment of our paper. We are glad that our additional results were able to address your questions and concerns.

---

### Official Review · Reviewer_MpW9 · 2025-07-01

**Clarity:** 2
**Significance:** 3
**Originality:** 4
**Rating:** 5
**Confidence:** 4

**Summary:**

The authors propose a new coding scheme for channel simulation based on standard Rejection Sampling (RS) and extend this analysis to Ensemble Rejection Sampling (ERS). They derive the coding cost for both methods, showing that they are first-order optimal. The paper then applies ERS to the problem of distributed matching, for which the authors derive a new bound on the matching probability, establishing a counterpart to the Poisson Matching Lemma (PML) and Importance Matching Lemma (IML). The practical utility of the proposed ERS-based method is verified in a distributed compression setting on both synthetic Gaussian sources and the MNIST image dataset.

**Questions:**

1.  **Practicality of PML as a Baseline in the MNIST Experiment:** In the distributed image compression experiment, you argue that PML is inapplicable because the upper bound of the density ratio $p_{Y'|X'}(y'|x')/q_{Y'}(y')$ is unknown when learned by a neural network. However, this seems to warrant more discussion:

    **1a.** PML, like PFR or GRS, only requires the *local* constant $\omega_{B,x'} = \sup_{y'} p_{Y'|X'}(y'|x')/q_{Y'}(y')$ at the decoder for a given side information $x'$. Since the decoder has access to the neural network representing these distributions (in this case a Gaussian), why can it not compute or tightly bound this local constant on the fly for each sample and use a PML-based scheme?

    **1b.** Furthermore, your ERS scheme operates with a chosen group size, which is directly related to the density ratio. Could you clarify why the information implicit in the choice of group size cannot be used to parameterize a PML-based competitor for a direct comparison?

2.  **Computational Cost of the Sorting-Based Coding:** The rejection sampling coding scheme has some similarities to PFR, where uniform random variables serve as arrival times (though it's not quite a Poisson process since $N$ is fixed). Can you think of a more computationally efficient method for coding the sample--ideally on the order of $\omega_x$ ? Or must all the uniforms in the group be drawn every time? Finally, is there any downside, in terms of rate, to making groups larger than $\omega$?


### Minor Suggestions for Presentation:

*   **Clarity of Indices:** The use of same letters for different indices (e.g., group index, batch index, local index) can be slightly confusing to follow.
*   **Figure Readability:** The font size in the figures is smaller than the main text and can is difficult to read. Font size should be at least as large as the main text, ideally larger, to ensure clarity.
*   **Context in RD Plots:** In Figure 5 (Right), adding a curve for the theoretical Wyner-Ziv rate-distortion bound would provide valuable context. It would help to assess the gap between all the practical methods and the theoretical limit.

**Ethical Concerns:**

["NO or VERY MINOR ethics concerns only"]

**Final Justification:**

I maintain my 'Accept' rating.

Discussion with the authors clarified my misunderstanding about the different distributions required for distributed compression.

The limitations of the proposed scheme do not negate its valuable contribution. These include:
1) an upper bound on excess functional information that is worse than PFR, and
2) a computational complexity dependent on a global bounding constant, versus a local one for PFR and GRS--for channel simulation. For distributed matching, we do not know of a good alternative when utilising neural networks at decoder side.

**Limitations:**

Yes, the authors briefly mention improving runtime efficiency as a future direction. However, the paper would benefit from a more direct discussion of its two key limitations and their practical context:
1.  The computational complexity can be significantly higher than competing methods since it depends on the global density ratio bound not a local one.
2.  The requirement for this global bound to be known by both parties is a stricter assumption than that of several existing channel simulation schemes.

**Paper Formatting Concerns:**

No major formatting issues.

**Quality:**

4

**Strengths And Weaknesses:**

## Strengths:

- **New Rejection Sampling Channel Simulation:** The paper introduces a novel coding scheme for standard Rejection Sampling (RS) based on grouping and sorting proposals. It bypasses the suboptimality of naive runtime-based coding for RS, achieving a near-optimal rate of $I(X;Y) + O(\log I(X;Y))$.
- **New Ensemble Rejection Sampling Channel Simulation** The authors extend the RS coding technique to Ensemble Rejection Sampling (ERS) and also show it to be near-optimal.
- **New Distributed Matching Lemma:** A major theoretical strength is the derivation of a distributed matching lemma for ERS. This result serves as a new tool for one-shot distributed information theory, alongside existing Poisson Matching Lemma (PML) and the Importance Matching Lemma (IML). Its existence expands the toolkit available for analyzing distributed compression and communication problems.
- **ERS Distributed Compression:** Authors show how ERS matching can be used for distributed compression with decoder-only side information, mirroring IML scheme. Unlike in IML, which produces biased samples, the chosen sample $Y$ comes from exact distribution.

## Weaknesses:

- **Dependence on Global Bounding Constant:** A practical limitation of the proposed RS and ERS schemes is the requirement that both the encoder and decoder know the global worst-case density ratio $\omega = \sup_{x,y} \frac{p_{Y|X}(y|x)}{q_Y(y)}$ beforehand. This contrasts with schemes like Poisson Functional Representation (PFR) or Greedy Rejection Sampling (GRS) which do not require such a global constant to be known to the decoder. For PFR and GRS only the local constant is needed $\omega_x = \sup_y \frac{p_{Y|X}(y|x)}{q_Y(y)}$ and exclusively at the encoder.
- **Computational Complexity:** The average computational complexity of the proposed methods scales with the global constant $\omega$, rather than the instance-specific ratio $\omega_x = \sup_y \frac{p_{Y|X}(y|x)}{q_Y(y)}$. This can lead to significant inefficiency. Consider the following example: $p(y|x=0)=Unif[0,1), p(y|x=1)=Unif[0,c), p(x=1)=c$; average PFR/GRS computation $<2$, while average RS and ERS computation $>1/(2c)$ can be arbitrarily large.
- **Looser Rate Bounds:** While the coding rates for both RS and ERS are shown to be asymptotically optimal (i.e., achieving the leading term $I(X;Y)$), the upper bounds contain larger logarithmic and constant terms compared to the current channel simulation methods, such as those shown by PFR ($I(X;Y) + \log(I(X;Y)+1) + 4$).

---

> ### Author Rebuttal · Authors · 2025-07-30
>
> Thank you for your thoughtful and detailed review. We appreciate your recognition of our theoretical contributions—including the new RS and ERS coding schemes and the distributed matching lemma—as well as the practical value of ERS in distributed compression tasks. Please find your concerns addressed below.
>
> **(1) Dependence on Global Bounding Constant/Computational Complexity**
>
> Your observation is indeed correct for the current scheme based on standard rejection sampling, where both parties must share knowledge of $\omega$. In practice, the encoder can transmit this information prior to sending the actual samples, allowing the coding cost of $\mathcal{O}(\log(\omega))$ to be amortized over time.
>
> Moreover, ERS provides several benefits compared to existing methods:
> - Compared to the Importance Matching Lemma, ERS delivers exact output samples, resulting in improved performance in low-distortion scenarios for distributed compression, as shown in Section 6.
> - Compared to the Greedy Rejection Sampling scheme, ERS also achieves better distributed matching probabilities, where GRS falls short.
> - Compared to the original PML scheme for distributed compression with high-dimensional data, our ERS-based coding scheme with neural networks effectively addresses the decoder-side termination issue, which the original PML scheme does not fully resolve (also see the answer for Question 1 below).
>
> We further note that, in the context of distributed compression using ERS, choosing the batch size as $N = 2^{\mathbb{E}[D_{\mathrm{KL}}(p_{Y|X}(\cdot|x) \,\|\, q_Y(\cdot))] + 4}$ is sufficient to ensure a low coding cost for the batch index. Empirically, the algorithm tends to terminate within 2 to 4 batches. This leads to a constant $\mathcal{O}(1)$ coding cost, allowing the batch index to be efficiently encoded via unary coding. This is practically more efficient than using the global constant $\omega$ described in Proposition 5.6, which serves as a conservative bound for the general achievability result. Importantly, the validity of the output samples is maintained regardless of the specific value of $N$.
>
> **(2) Looser Rate Bounds**
>
> Our experiments in Figures 2 and 3, conducted with actual distributions, indicate that the coding cost bounds stated in Proposition 4.1 and Proposition 5.1 tend to overestimate the cost observed in practice. This suggests that the theoretical bounds—particularly the one for ERS—can likely be tightened and the higher constants could be the consequences of our bounding techniques.
>
>
> **(3) Practicality of PML: Compute Local Constant On The Fly**
>
> In our scheme using ERS, the encoder requires an upper bound $\omega_x$, which depends on the target distribution $p_{Y|X}$ and the proposal $q_Y$, to guarantee termination. Importantly, this constant is local to the encoder, and the scheme does not rely on $\omega_{x'}$, the corresponding constant on the decoder’s side. In contrast, PML requires the decoder to compute $\omega_{x'}$ to ensure termination, which demands further assumptions on $p_{Y|X'}$ and can degrade performance. This asymmetry highlights a key practical advantage of ERS over PML.
>
> In our experiment, the decoder uses a neural contrastive estimator to directly estimate the likelihood ratio $\frac{p_{Y'|X'}(y'|x')}{q_{Y'}(y')}$ without assuming a specific form---such as Gaussian---for the posterior $p_{Y'|X'}$. Consequently, computing the local constant $\omega_{B,x'}$ needed for PML becomes intractable. If we instead assume a Gaussian form, $\mathcal{N}(\mu(x'), \sigma(x'))$, PML becomes feasible; however, our experimental results show that this assumption introduces a mismatch that worsens the rate-distortion tradeoff. See the table below for further details, note that we are presenting MSE distortion for different schemes  and lower is better. Embedding MSE refers to the error measured with respect to the encoder's neural network output, while pixel MSE captures the distortion at the image level.
>
>
> | Rate  | Model              | Embedding MSE | Pixel MSE |
> |-------|--------------------|------------|-------------|
> | 8.75  | Gaussian Regressor | 0.7300     | 0.0696      |
> |       | ERS (NCE)          | **0.6024**     | **0.0683**      |
> | 9.60  | Gaussian Regressor | 0.5260     | 0.0660      |
> |       | ERS (NCE)          | **0.4807**     | **0.0647**      |
> | 10.10 | Gaussian Regressor | 0.4310     | 0.0638      |
> |       | ERS (NCE)          | **0.3616**     | **0.0623**      |
> | 10.60 | Gaussian Regressor | 0.3600     | 0.0626      |
> |       | ERS (NCE)          | **0.2930**     | **0.0606**      |
>
> **(4) Implicit Information in the choice of group size and PML**
>
> We understand you are referring to the batch size $N$, chosen such that $N > \omega$, where $\omega$ denotes the global upper bound of the encoder’s density ratio. Please let us know if this interpretation is incorrect. Since $\omega$ serves as a global constant for the encoder’s target distributions, we are not aware of any existing methods for inferring the decoder’s local constant $\omega_{B,x’}$ based solely on $\omega$. This is because $\omega_{B,x’}$ requires an explicit knowledge of $p_{Y|X’}$, which is complex/intractable in general.  We will clarify this difference and add the presented experimental results above to clarify the advantage of our scheme over PML.
>
> **(5) Computational Cost of the Sorting-Based Coding**
>
> A key challenge is that the decoder does not know the value of $x$, and therefore the group size $\omega_x$ is also unknown. As a result, encoding the sample using its runtime complexity $\omega_x$ remains an open problem. While explicitly transmitting $\omega_x$ would incur a high communication cost, exploring strategies where the sampling complexity adapts to $\omega_x$ is a promising direction for future work, which we will highlight in the paper.
>
> **(6) Downside of making groups larger than $\omega$**
>
> In the proposed protocol, where each sample has an acceptance probability of $\frac{1}{\omega}$, the group size is set to $\omega $ (or its floor value). If the group size exceeds $\omega$, for instance, approaching infinity, the coding cost for $\hat{K}$ will increase since there will be more uniform random variables whose value is smaller than the one selected by the rejection sampler (in terms of expectation). Technically, this is evident in equations (82-84) of the proof in the appendix, where replacing $\lfloor \omega \rfloor$ (the group size) with extreme values will show the effect.
>
> **(7) Minor Suggestions for Presentation**
>
> Thank you for your comment. We will incorporate these in the revised version.
>
> We hope this clarify your concerns. Let us know if you have further questions.

---

> > ### Comment · Reviewer_MpW9 · 2025-08-05
> >
> > Thank you for responding to the points I’ve raised.
> >
> > Regarding question (4), I was confused. I had assumed there was a single global bounding constant, $\omega$, but I now see that the encoder and decoder can have different ones.
> >
> > As a final question, is the proposed ERS distributed matching unbiased only if the learned network correctly estimates the ratio $P_{Y'|X'}(y') / Q_{Y'}(y')$? Even if so, I think that ERS is a valuable contribution to distributed matching, but this would be an important limitation to highlight.

---

> ### Author Response · Authors · 2025-08-06
>
> We thank the reviewer for the question and would like to provide some clarification. We first note that in **channel simulation**, the output sample generated by ERS is guaranteed to be **unbiased**. Our scheme ---alongside Poisson Monte Carlo (PMC) and Greedy Rejection Sampling (GRS)--- satisfies this property.
>
> In contrast, for **distributed matching with communication** (please see Section 3.2.2) ---  the setting that applies to distributed compression — the scenario is a bit different. Note that the encoder and decoder use two different target distributions. The encoder uses a target distribution $P^A(y)$ while the decoder with side information $Z$ uses a target distribution $P^B_{Y|Z}(y|z)$. As we have noted in our paper, while the sample $Y_A$ generated at the encoder remains unbiased i.e.,  $Y_A$ follows the target distribution $P^A(y)$ this is not the case with the sample $Y_B$ generated at the decoder i.e., $Y_B$ may not follow the target distribution  $P^B_{Y|Z}(y|z)$ even if the density ratios $P^B_{Y|Z}(y|z)/Q_Y(y)$ is perfectly estimated. We kindly refer you to Section 3.2.2 ( lines 132-138 on page 4) for further clarification of this issue. The reason for this is that side information $Z$ in general can be dependent on the common randomness $W$  as it is only required to satisfy the Markov chain $ Z - (X, Y_A) - W $ (see Section~3.2.2). We note that this has also been clarified in the [1] (see discussion after Lemma 2) which considers a similar setting using the Poission Monte Carlo sampling method and is not a limitation of our proposed method.
>
>
> We would like to however emphasize that our proposed scheme does guarantee that the samples generated the encoder are unbiased and follow the target distribution at the encoder. Indeed any reference to unbiasedness in our experimental section refers solely to the encoder’s sample quality.  This property enables our scheme to outperform the Importance Matching Lemma baseline, where even the encoder's outputs are biased.
>
>
> We also note that if the density ratio is incorrectly estimated at the decoder, it could decrease the matching probability and in turn introduce more errors at the decoder. We will add a clarification on this in the paper if it is accepted.
>
> [1] Li, Cheuk Ting, and Venkat Anantharam. "A unified framework for one shot achievability via the Poisson matching lemma." IEEE Transactions on Information Theory 67.5 (2021): 2624-2651.

---

### Official Review · Reviewer_VjgP · 2025-07-02

**Clarity:** 3
**Significance:** 3
**Originality:** 2
**Rating:** 4
**Confidence:** 4

**Summary:**

This paper studies problems of channel simulations and (distributed) matching probabilities. Although PFRL and PML-based coding schemes can achieve near-optimal coding rates for these problems, they are impractical as they require an infinite number of proposals (unbounded termination time). To address this issue, the authors have developed a new coding technique for RS (for distributed matching) and ERS (for both channel simulation and distributed matching), demonstrating that it can achieve near-optimal performance for the considered problems. The practicality of the proposed approach is demonstrated through experiments.

**Questions:**

-	As mentioned by the authors, the main pitfall of PMC-based approach is their termination time. However, when using RS or ERS in practice, it is also necessary to choose a maximum termination time.

To better explain this; consider Proposition 4.1. that establishes a bound that is independent of $\omega$. This means that letting $\omega \to \infty$ does not induce any (considerable) penalty for the coding rate. However, this comes at the cost of a larger $K$; hence more common randomness and larger delays. For example, even when the intended $K$ is found, in this scheme to send $\hat{K}$, one must wait until end of the group (of size $\lceil \omega \rceil$) in which $K$ is located, in order to sort them. Thus, in practice, for large values of $\omega$, an upper limit must be set. The question then is what effect such a limit has on the quality of the simulated sample.

-	It is unclear what benefit ERS has over RS for channel simulation problems. In other words, what benefit does the analysis offer for general $N$? As mentioned by authors, the analysis for $N=1$ and $N=\infty$ is similar to that in previous works.

-	In [39], the issue of unbounded termination time in PFRL for channel simulation is addressed using a simple Gumbel-max trick. Could a similar approach be used for PML-based coding schemes and the distributed matching problem?

-	The definition of $\mu_1'$ is deferred to Appendix F.5. Is it referred to as $\mu_1^{cond}$ therein? If so; they do not only depend on $N$; but also on $\omega$ and distributions. It would be better to at mention this.

-	As he authors mentioned, the main contribution of the paper is the analysis of ERS for matching probability. However, if I understand correctly, it has not been used or verified in the experimental part. Moreover, I do not fully understand why, in the context of 'lossy compression', one would be interested in matching probability in Proposition 5.6. Instead, one should be interested in the incurred distortion, as considered in the experiments.

-	I am also concerned about the practicality of these approaches in real life, for example in image compression, where the distributions are unknown and the data is high-dimensional. In the experimental section of the paper, an example is given of a neural contrastive estimator being used to approximate the ratio. However, such approaches perform poorly in the high-dimensional regime. This limitation is not specific to this paper, but to similar papers as well. Do the authors have any suggestions for how such approaches could be used in real-life applications?

Some of the typos: some mispelled words such as `Simmulation`, missing space at line 318, many places $\ldot$ should be replaced by $\cdot$, e.g. $\mu(\cdot)$ in line 323.

=======================

POST REBUTTAL:

This is a rather interesting paper. There are still some minor concerns on the computability of the log density ratio in high dimension; but given the clear nature of this paper, which is theoretic, and the technical novelty, I vote for acceptance.

**Ethical Concerns:**

["NO or VERY MINOR ethics concerns only"]

**Final Justification:**

This is a rather interesting paper. There are some concerns on the computability of the log density ratio in high dimension; but given the clear nature of this paper, which is theoretic, and the technical novelty, I vote for acceptance if room.

**Limitations:**

yes

**Paper Formatting Concerns:**

No concerns.

**Quality:**

3

**Strengths And Weaknesses:**

On the strong points:

-	The new coding technique and its analysis for RS are novel and interesting.

-	Similarly, the analysis for ERS, particularly for distributed matching problem is involved and novel.

-	The proofs appear correct and rigorous to me. (I have not gone through all the proofs; but even those over which I could not go in details I have glanced over them.)

On the weak points:

For details, please refer to the below `Questions' section.

-	The benefits of ERS and the proposed coding schemes require further elaboration.

-	The application of the proposed approach to machine learning problems is not discussed in detail and remains unclear. For a paper submitted to a top-tier ML conference I feel that this aspect should be largely improved. For the moment the paper appears to be more suitable for a conference such as ISIT or the Info. Theory Transactions.

-	The boundedness of the ratio can be restrictive in practice; as one needs to ensure this by using a very large $\omega$; which results in longer delays.

---

> ### Author Rebuttal · Authors · 2025-07-30
>
> Thank you for your thoughtful review and for highlighting the novelty and rigor of our RS and ERS coding techniques, as well as the strength of our theoretical analysis. Please find the answer to your questions below.
>
> **(1) Sample Quality for Truncated $\omega$**
>
> When the constant $\omega$ is truncated, we believe it is possible to characterize the quality of the generated samples. In particular, it suffices to consider inputs $x$ where the local constant $\omega_x = \max_y P_{Y|X}(y|x)/Q_Y(y)\geq \omega$, as sample quality remains unaffected otherwise. To the best of our knowledge, the effect of truncating the upper bound $\omega$ has been studied in the contexts of rejection sampling (RS) and importance sampling (IS), but not for PMC. For $N = 1$, which corresponds to standard RS, existing early stopping techniques [A6] can be applied to derive bounds on sample quality. When $N > 1$, ERS incorporates an IS step, allowing similar analyses from [6,39] to be adapted for bounding the sample quality under ERS. We note that PMC remains the primary competitor for generating unbiased samples with high matching probability and this point showcase another advantage of our scheme over PMC that we will put a discussion in the revision.
>
> Finally, in terms of terminating condition in the distributed compression application there is an important practical advantage that ERS has over PMC as discussed in reponse to Question 1 for Reviewer MpW9 (answered in points (3) and (4) in our rebuttal to their).
>
> **(2) Benefits of ERS over prior methods and RS**
> The motivation for developing ERS is to improve the matching probability of RS and its greedy variant, i.e. Greedy Rejection Sampling (GRS), which alone are not competitive, also demonstrated in the examples of Figure 4 and Appendix D.2. To address this, ERS integrates RS with IS—an approach known for achieving high matching probabilities, albeit with biased output samples. In this sense, ERS combines the best of both worlds: high matching probability and unbiased output samples. We highlight several advantages of the ERS scheme over existing approaches:
> - Compared to the Importance Matching Lemma [33], ERS generates exact output samples, leading to improved performance in low-distortion regimes for distributed compression, as shown in Section 6.
>  - Compared to GRS  and standard RS scheme [12], ERS consistently achieves higher distributed matching probabilities.
> - Compared to the PML scheme for distributed compression with high-dimensional data, our ERS-based coding scheme with neural networks effectively addresses the decoder-side termination issue, which the original PML scheme does not fully resolve.
>
> In channel simulation, ERS has the potential to offer additional advantages over standard RS. There exists a substantial body of research aimed at improving the sample efficiency of IS, such as multiple IS and quasi-Monte Carlo. As ERS incorporates an IS within its procedure, we believe these advances could be effectively integrated to further enhance the efficiency of the ERS framework.
>
> **(3) The choice of $N$**
>
> For the distributed compression, the choice of $N$ is crucial for the performance. If $N=1$, we will have a low matching probability. On the other hand, setting $N$ very large will lead to high computational overheads. For any $N$, the scheme is feasible and the right choice of $N$ is to obtain the balance between matching probability and computational complexity. Our experiments suggest that setting the batch size as $N=2^{\mathbb{E}[D_{\mathrm{KL}}(p_{Y|X}(\cdot|x) \,\|\, q_Y(\cdot))]+4} $ is sufficient to achieve low coding costs for the batch index. This choice is not arbitrary—it aligns with the number of samples typically required to ensure low bias in IS [6]. With this, ERS usually terminates within just 2--4 batches. Thanks to this $\mathcal{O}(1)$ coding cost, the batch index can be efficiently encoded using unary coding and is more practical than relying on the global constant $\omega$ introduced in the theoretical analysis of Proposition 5.6, which primarily serves to make the rate analysis tractable. Importantly, the correctness of the generated samples is maintained for any choice of $N$.
>
> **(4) Gumbel-Max trick and Matching Lemmas**
>
> In  [39], the technique known as Ordered Random Coding utilizes the Gumbel-max trick to address termination time. However, this method inherently produces biased samples due to its reliance on IS (see Corollary 3.2 in [39]). In contrast PFRL is an exact sampling scheme that generates unbiased samples (unlike the approach in [39]). An extension of this approach to matching scenarios as you described was introduced by Phan et al. [33], known as the Importance Matching Lemma (IML), which also produces biased samples. Our work directly extends the work by Phan et al. [33], which serves as a baseline we compared against in Section 6. The primary distinction of our approach compared to those in [33, 39] lies in our method of generating unbiased samples by combining IS with RS, and achieving better empirical performance in the Section 6. In particular, the results in Section 6 demonstrate that ERS outperforms IML in distributed compression scenarios.
>
> **(5) Definitions of $\mu$** Thank you for pointing this out. We will revise and adjust this.
>
> **(6) Verification of Matching Probability and Implication in Lossy Compression.** We verify of the general matching probability in Proposition 5.4 with empirical comparisons of different methods presented in Figure 4. For Proposition 5.6, we report empirical matching probabilities under joint compression of multiple samples in Figure 9 (Appendix), which also illustrates the phenomenon described in Remark 5.7. We note that the matching probability is a central quantity in distributed compression, directly influencing the rate-distortion tradeoff. Our experiments involving distributed compression are designed to validate the performance gains attributed to improved matching probabilities. Schemes that fail to achieve high matching probability—despite being efficient for channel simulation such as RS and GRS—do not yield competitive rate-distortion performance in the distributed setting.
>
> Finally, result in Proposition 5.6 can be translated into related quantities, such as the probability of excess distortion, as discussed in [25]. This can be used to derive bounds on the expected distortion for bounded inputs, which we will incorporate in the revision. Since matching probability is a central quantity in the distributed compression setting, we characterize it in Proposition 5.6. This is consistent with the IML/PML baseline  and Proposition 5.6 serves as a counterpart result within our framework.
>
> **(7) Scaling Channel Simulation-Based Approaches**
>
> Channel simulation is increasingly applied to high-dimensional datasets. In image compression, [A1, A2] use it within diffusion models to achieve competitive performance on 512×512 images with < 10 second encoding, outperforming traditional quantization in perceptual quality. It is also effectively combined with implicit neural representations, which have shown superior results over quantization-based methods [A8].
>
> Furthermore, the application of channel simulation extends beyond lossy image compression. It has been explored in areas such as differential privacy [A3] and quantum communication [A9]. Recent works  [A4, A5] also apply similar distributed-matching techniques to accelerate inference in large language models, showcasing the growing interest of using such techniques for machine learning problems.
>
> Regarding the scaling of the neural contrastive estimator to higher dimensional data, a practical workaround is to partition the input into smaller chunks and train a separate neural contrastive estimator on each subset. Furthermore, state-of-the-art methods for density estimation are still rapidly evolving, as demonstrated by recent advances in multimedia generative models. Nonetheless, in applications such as vertical federated learning—where feature dimensionality does not increase as quickly as in image-based tasks—channel simulation-based approaches remain both practical and directly applicable. Overall, scaling channel simulation to high-dimensional data is an active and promising research direction (see [37], [A7]).
>
> We hope the reviewer is convinced about the strength of our scheme over prior methods.
>
> **References**
>
> [A1] Vonderfecht, J., & Liu, F. Lossy Compression with Pretrained Diffusion Models. ICLR 2025.
>
> [A2] Yang, Yibo, et al. "Progressive Compression with Universally Quantized Diffusion Models.". ICLR 2025
>
> [A3] Liu, Yanxiao, et al. "Universal exact compression of differentially private mechanisms.". NeurIPS 2024
>
> [A4] Kobus, Szymon, and Deniz Gündüz. "Speculative sampling via exponential races." arXiv preprint arXiv:2504.15475 (2025).
>
> [A5] Rowan, Joseph, Buu Phan, and Ashish Khisti. "List-Level Distribution Coupling with Applications to Speculative Decoding and Lossy Compression." arXiv preprint arXiv:2506.05632 (2025).
>
> [A6] Block, Adam, and Yury Polyanskiy. "The sample complexity of approximate rejection sampling with applications to smoothed online learning." PMLR, 2023.
>
> [A7] Jiajun He, et al.  "Accelerating relative entropy coding with space partitioning." NeurIPS 2024
>
> [A8] He, Jiajun, et al. "RECOMBINER: Robust and Enhanced Compression with Bayesian Implicit Neural Representations." ICLR 2024
>
> [A9] Steiner, Michael. "Towards quantifying non-local information transfer: finite-bit non-locality." Physics Letters 2000.

---

> > ### Comment · Reviewer_VjgP · 2025-08-05
> >
> > Thank you for your detailed response, which have addressed most of my points.
> >
> > - On your response to the question about the estimation of the log density of KL divergence term in practice using neural networks, it is not clear to me yet how you get around this step, known to be challenging (e.g., in the estimation of MI, CMI and other related problems).

---

> ### Author Response · Authors · 2025-08-06
>
> Thank you for your question. In our work, rather than estimating the mutual information $I(X’;Y)$, we focus on estimating the density ratio $P_{Y|X’}(y|x’)/Q(y)$ using the scheme in [1] (see Appendix C) --- an approach that is well-established and widely used in the literature. In high dimensional settings more advanced schemes developed in [2] can also be used.  In our experiments, since we project the input images (MNIST/CIFAR) into an embedding vector of size 3 and perform distributed compression in this lower dimensional space it turns out that vanilla estimator sufficiently gives stable and acceptable performance.
>
>
> We demonstrate this through the following experiment. In **Table 1**, we revisit the 4D Gaussian setup by replacing the closed-form log-density ratio with a neural network scheme in [1]. We evaluate both approaches at four target distortion levels—corresponding to the encoder's ideal distortion tradeoffs, similar to the setting in Figure 5 (middle). For each distortion level, we fix the same rate as in the experiment in the paper and measure the actual distortion at the decoder’s output. The results show that the neural network scheme closely matches the performance of the closed-form expression across all settings.
>
> **Table 1.** Distortion (in dB) achieved at various target rates using closed-form vs. NCE-based estimation (repeat with 5 seeds).
>
>
> | Target Rate / Distortion     | 2.77 bit / -21.5 dB | 3.03 bit / -22 dB | 3.26 bit / -22.5 dB | 3.52 bit / -23 dB |
> |------------------------------|---------------------|-------------------|---------------------|-------------------|
> | Closed-form Expression       | -21.26 ± 0.03       | -21.85 ± 0.04     | -22.37 ± 0.03       | -22.86 ± 0.02     |
> | NCE                          | -21.21 ± 0.04       | -21.82 ± 0.05     | -22.28 ± 0.04       | -22.80 ± 0.03     |
>
>
> To illustrate the stability of our approach in the MNIST setting, we train three neural networks with different random seeds to estimate the likelihood ratio. We then evaluate the *agreement rate*---defined as the percentage of times the decoders select the same output given the *same setup* (i.e., the same input \( X \), common randomness \( W \), and side information \( X' \)), but using different neural network estimators. We conduct experiments across four different target distortion level (or equivalently the $\beta-VAE$ hyperparameter described in Appendix K). We observe consistently high agreement scores across all settings (>97%), indicating that the proposed approach is stable with respect to variations in the neural network estimator. The results is in Table 2.
>
>
> **Table.2** Agreement rate across different target distortion levels (measured by \beta).
>
>
> | Target Distortion /$ \beta$        | 0.066 – 0.225 | 0.063 – 0.28 | 0.061 – 0.31 | 0.058 – 0.40 |
> |------------------------------|---------------|--------------|--------------|--------------|
> | Agreement Score              | 98.32%        | 98.01%       | 97.76%       | 97.31%       |
>
>
> We would like to add that in the distributed compression  the density ratio is used as an intermediate quantity that is used to compute  a *score function* to guide sample selection at the decoder. Small inaccuracies in this score can be considered as small perturbations in the ranked code points and consequently will not affect the decoding.   We sincerely hope that the reviewer is convinced about the technical soundness of our approach of computing the density ratio using neural networks and are happy to provide any further clarifications.
>
>
> [1] Hermans, Joeri, Volodimir Begy, and Gilles Louppe. "Likelihood-free mcmc with amortized approximate ratio estimators." International conference on machine learning. PMLR, 2020.
>
>
> [2] Choi, Kristy, et al. "Density ratio estimation via infinitesimal classification." International Conference on Artificial Intelligence and Statistics. PMLR, 2022.

---

> > ### Comment · Reviewer_VjgP · 2025-08-07
> >
> > Thank you for the additional clarifications which have addressed my concerns.
> >
> > This is a rather interesting paper; I have raised up my rating; and I will vote for acceptance.

---

> > > ### Author Response · Authors · 2025-08-07
> > >
> > > Thank you very much for your thoughtful engagement with our work. We’re pleased to hear that your concerns have been addressed and appreciate your constructive feedback throughout the process!

---

### Official Review · Reviewer_AAcC · 2025-07-03

**Clarity:** 3
**Significance:** 3
**Originality:** 3
**Rating:** 5
**Confidence:** 3

**Summary:**

This paper introduces a new framework for channel simulation and distributed compression using Ensemble Rejection Sampling (ERS), a generalization of standard rejection sampling (RS). It addresses two central problems: сhannel simulation and distributed matching.
Key contributions include:
- A novel RS-based compression method achieving coding costs close to the mutual information bound.
- A distributed matching lemma for ERS, extending prior work like the Poisson Matching Lemma (PML) and Importance Matching Lemma (IML).
- Bounds and empirical validation on synthetic data (Gaussian sources) and MNIST image compression tasks.

**Questions:**

- Channel simulation. What is the main task? Communicate a sample $Y$ or generate it from $P_{Y|X}$?
- Distributed matching. Please mention why the probability (2) is of the most interest. Am I right that $P_A$ and $P_B$ are discrete distributions?
- How does the choice of batch size N in ERS affect practical runtime and memory usage in large-scale settings?
- Can ERS be directly extended to domains beyond images, such as video or NLP, where the proposal and target distributions may be more complex?
- How would ERS perform in scenarios requiring privacy-preserving sampling (e.g., differential privacy)?

**Ethical Concerns:**

["NO or VERY MINOR ethics concerns only"]

**Final Justification:**

The authors carried out more experiments and addressed all my questions

**Limitations:**

The experiments are limited to synthetic Gaussian sources and MNIST. Including more complex or real-world datasets (e.g., CIFAR, ImageNet) could strengthen the empirical claims.

**Quality:**

3

**Strengths And Weaknesses:**

Strengths
The paper is well-written and clearly outlines its contributions in a structured manner, supported by theory and experiments. Also I would emhasize the presence of theoretical justification and well-structured proofs.

Weaknesses
- While the theoretical complexity is discussed, practical runtime and memory overhead for ERS (especially in high-dimensional settings) could be elaborated further.
- The experiments are limited to synthetic Gaussian sources and MNIST. Including more complex or real-world datasets (e.g., CIFAR, ImageNet) could strengthen the empirical claims.
- Broader potential implications (e.g., in privacy-preserving communication or edge AI) could have been mentioned.
- There are several typos, e.g. simmulation

---

> ### Author Rebuttal · Authors · 2025-07-30
>
> Thank you for your thoughtful review and positive feedback. We appreciate your recognition of the theoretical justification, well-structured proofs, and clear presentation of our contributions. You can find your questions addressed below.
>
> **(1) Additional Experiment on CIFAR-10 Dataset**
>  We conduct the experiments on the CIFAR-10 dataset and compare our method with the implicit neural representations [A3] , quantization approach [A7] and Importance Matching Lemma (IML). We use Mean Squared Error (MSE) as the distortion metric across all schemes, where lower values indicate better performance. Our approach consistently achieves lower distortion by leveraging side information within the encoding scheme.
>
> | Rate (bits/image) | Balle (2018) [A7] | RECOMBINER (He 2023) [A3] | IML (Phan 2024) [33] | ERS (Ours) |
> |-------------------|--------------|------------------------|------------------|------------|
> | ~9                | 0.0972       | 0.0968                 | 0.0711           | **0.0703** |
> | ~10               | 0.0915       | 0.0912                 | 0.0668           | **0.0659** |
> | ~11               | 0.0802       | 0.0810                 | 0.0621           | **0.0606** |
>
> **(2) Details about Channel Simulation**
> In the context of our work, the main task of channel simulation is compressing a noisy sample $Y \sim P_{Y|X=x}$ of the input $X = x $ from the encoder , which will be then transmitted to the decoder. In the setting we consider, both parties share a source of common randomness $W $, which corresponds to a shared random seed in practice. The encoder employs a sampling algorithm—such as (ensemble) rejection sampling, importance sampling, or Poisson functional representation—to select the sample $ Y $ using $ W $. Once $Y$ is selected, the goal is to encode the index of $Y$ in $W$ to be close to the theoretical limit $I(X;Y)$ and communicate this message to the decoder. Since the decoder has access to $W$, it can retrieve the value $Y$ after decoding the index from the message.
>
> **(3) Details about Distributed Matching Quantity**
> The probability in (2) is of interest because it quantifies how likely both parties are to select the same index, given their respective target distributions $P_A$ and $P_B$. This quantity plays a central role in the distributed compression setting. Since many distributed compression and communication problems can be characterized as specific instances of distributed matching (Figure 1.c), this quantity can be used to derive one-shot achievability results for a range of compression and communication settings [25]. We further note that this setup is related to the coupling problems in statistics and computer science and have found many applications therein.
>
> **(4) The choice of batch size N in ERS**
>
> In the context of distributed compression using ERS, setting the batch size as $N = 2^{\mathbb{E}[D_{\mathrm{KL}}(p_{Y|X}(\cdot|x) \,\|\, q_Y(\cdot))] + 4} $ is sufficient to ensure a low $ \mathcal{O}(1) $ coding cost for the batch index. This choice is motivated by the analysis of the number of samples required in importance sampling [6]. Empirically, the algorithm typically terminates within 2 to 4 batches, resulting in a constant $\mathcal{O}(1)$ coding cost. This enables efficient encoding of the batch index using unary coding. Notably, this choice is more efficient than relying on the global constant $\omega$ from Proposition 5.6, which provides a conservative bound for the general achievability result. Importantly, the validity of the output samples is preserved regardless of the specific value of $N$.
>
> In our experiments, when jointly compressing a 4D Gaussian vector as described in Section 6, this configuration requires approximately $ 2^{24}$ proposals, which are parallelized and executed on a single A4500 GPU with 20GB of memory. The average encoding time is under one second per sample. Further improvements can be achieved by incorporating techniques such as space partitioning [19]. Since larger values of $N$ reduce the overhead in batch index encoding, it is beneficial in practice to increase $N$ as much as GPU capacity allows, in order to maximize parallelism.
>
> **(4) ERS on domains beyond images**
>
> There are several existing works showing promise in scaling channel simulation techniques to high-dimensional data, which we believe can benefit ERS. For example, [A1, A2] demonstrate how to compress high-fidelity images using channel simulation methods applied to diffusion models. In particular, [A2] introduces a custom CUDA kernel that exploits parallelism to encode images in under 10 seconds. These methods outperform traditional quantization techniques by better preserving perceptual realism. An additional advantage is their natural compatibility with diffusion models, a leading approach in image generation. Given the strong performance of diffusion models in video generation, we believe similar extensions of channel simulation techniques to video are feasible. Beyond diffusion models, channel simulation has also been applied to implicit neural representation methods for image compression, demonstrating promising results. For example, [A3] applied channel simulation techniques to video compression problems, and such approaches may potentially be extended to other modalities as well.
>
> **(5) ERS in privacy-preserving setting**
>
> Several existing works have applied channel simulation techniques to differential privacy problems [A4, A5], suggesting that our ERS framework can also be extended in this direction. Moreover, since our scheme is inherently designed for distributed compression with distribution preservation, it opens up the possibility of incorporating differential privacy into distributed compression—an exciting direction for future research, particularly in practical settings such as vertical or clustered federated learning [A6].
>
>
> We hope that our rebuttal address your concerns. Please let us know if you have any additional questions.
>
>
> **References**
>
>
> [A1] Vonderfecht, J., & Liu, F. Lossy Compression with Pretrained Diffusion Models. In The Thirteenth International Conference on Learning Representations.
>
> [A2] Yang, Yibo, Justus Will, and Stephan Mandt. "Progressive Compression with Universally Quantized Diffusion Models." The Thirteenth International Conference on Learning Representations.
>
> [A3] He, Jiajun, et al. "RECOMBINER: Robust and Enhanced Compression with Bayesian Implicit Neural Representations." The Twelfth International Conference on Learning Representations.
>
> [A4] Shah, Abhin, et al. "Optimal compression of locally differentially private mechanisms." International Conference on Artificial Intelligence and Statistics. PMLR, 2022.
>
> [A5] Liu, Yanxiao, et al. "Universal exact compression of differentially private mechanisms." Advances in Neural Information Processing Systems 37 (2024): 91492-91531.
>
> [A6] Kim, Heasung, Hyeji Kim, and Gustavo De Veciana. "Clustered federated learning via gradient-based partitioning." Forty-first international conference on machine learning. 2024.
>
> [A7] Ballé, Johannes, et al. "Variational image compression with a scale hyperprior." International Conference on Learning Representations. 2018.

---

> > ### Comment · Reviewer_AAcC · 2025-08-07
> >
> > I would like to thank the authors for the answers. All my questions are addressed, I will increase my rating to 5.

---

> > > ### Author Response · Authors · 2025-08-07
> > >
> > > We sincerely appreciate your interest in our work and are pleased that your concerns have been resolved!

---

### Note · Authors · 2025-08-14

We thank all reviewers for their constructive feedback, which has led to new results and expanded discussions that will be incorporated into the final version of the paper. In this closing statement, we outline the paper’s main strengths and summarize key clarifications and additional experiments provided during the rebuttal.

## Strengths
We highlight several advantages of the ERS scheme over existing approaches:

- **Over the Importance Matching Lemma [1]** – ERS produces *exact* output samples, yielding improved performance in low-distortion regimes for distributed compression (Section 6).
- **Over GRS [2] and standard RS** – ERS consistently achieves higher distributed matching probabilities.
- **Over the PML scheme [3] for high-dimensional data** – Our ERS-based coding scheme, combined with neural networks, effectively resolves the decoder-side termination issue that the original PML scheme does not fully address.

## Additional Results from Discussion
Following discussions with the reviewers, the final version of the paper will include the following additions to further demonstrate the strength of our method over existing approaches:

- Additional experimental results comparing ERS with other neural compression approaches.
- Results on the consistency of the neural contrastive estimator for decoding.
- Direct comparisons between PML and ERS in distributed compression on the MNIST image dataset.
- Expanded discussion on batch-size selection, runtime complexity, and potential practical applications.

---

**References**

[1] Phan, Buu, Ashish Khisti, and Christos Louizos. *Importance matching lemma for lossy compression with side information.* International Conference on Artificial Intelligence and Statistics. PMLR, 2024.

[2] Flamich, Gergely, and Lucas Theis. *Adaptive greedy rejection sampling.* 2023 IEEE International Symposium on Information Theory (ISIT). IEEE, 2023.

[3] Li, Cheuk Ting, and Venkat Anantharam. *A unified framework for one-shot achievability via the Poisson matching lemma.* IEEE Transactions on Information Theory 67.5 (2021): 2624–2651.

---

### Decision · Program_Chairs · 2025-09-17

**Decision:**

Accept (poster)

**Comment:**

This paper uses a generalization of standard rejection sampling, called ensemble rejection sampling, to improve the performance of "channel simulation" and "distributed compression", two communication problems.

The paper is well received by the reviewer who favor acceptance and are happy with the answers provided by the authors.  Overall this is an interesting paper. My main concern is that the problems are disconnected with machine learning - however there is potential that the technique may be useful, because it is rather basic. So I recommend acceptance.